# Autophagy deficiency promotes triple-negative breast cancer resistance to T cell-mediated cytotoxicity by blocking tenascin-C degradation

Zhi-Ling Li [1,5], Hai-Liang Zhang[1,5], Yun Huang[1,5], Jun-Hao Huang[1,2], Peng Sun[1,3], Ning-Ning Zhou[1,4], Yu-Hong Chen[1], Jia Mai[1], Yan Wang[1,2], Yan Yu[1], Li-Huan Zhou[1,2], Xuan Li[1], Dong Yang[1], Xiao-Dan Peng[1], Gong-Kan Feng[1], Jun Tang [1,2✉], Xiao-Feng Zhu [1✉] & Rong Deng [1✉]

Most triple-negative breast cancer (TNBC) patients fail to respond to T cell-mediated immunotherapies. Unfortunately, the molecular determinants are still poorly understood. Breast cancer is the disease genetically linked to a deficiency in autophagy. Here, we show that autophagy defects in TNBC cells inhibit T cell-mediated tumour killing in vitro and in vivo. Mechanistically, we identify Tenascin-C as a candidate for autophagy deficiency-mediated immunosuppression, in which Tenascin-C is Lys63-ubiquitinated by Skp2, particularly at Lys942 and Lys1882, thus promoting its recognition by p62 and leading to its selective autophagic degradation. High Tenascin-C expression is associated with poor prognosis and inversely correlated with LC3B expression and CD8[+] T cells in TNBC patients. More importantly, inhibition of Tenascin-C in autophagy-impaired TNBC cells sensitizes T cell-mediated tumour killing and improves antitumour effects of single anti-PD1/PDL1 therapy. Our results provide a potential strategy for targeting TNBC with the combination of Tenascin-C blockade and immune checkpoint inhibitors.

[1] State Key Laboratory of Oncology in South China, Collaborative Innovation Center for Cancer Medicine, Guangdong Key Laboratory of Nasopharyngeal Carcinoma Diagnosis and Therapy, Sun Yat-sen University Cancer Center, Guangzhou 510060, China. [2] Department of Breast Oncology, Sun Yat-sen University Cancer Center, Guangzhou 510060, China. [3] Department of Pathology, Sun Yat-sen University Cancer Center, Guangzhou 510060, China. [4] Department of Medical Oncology, Sun Yat-sen University Cancer Center, Guangzhou 510060, China. [5] These authors contributed equally: Zhi-Ling Li, Hai-Liang Zhang, Yun Huang. ✉email: tangjun@sysucc.org.cn; zhuxfeng@mail.sysu.edu.cn; dengrong@sysucc.org.cn

Triple-negative breast cancer (TNBC) is the most aggressive form of breast cancer. An improved understanding of the immunogenicity of TNBC has provided therapeutic options for patients with TNBC. Compared with hormone receptor-positive breast cancers, TNBCs show higher expression of PD-L1, levels of prognosis-related tumour-infiltrating lymphocytes (TILs), rates of genomic instability and rates of genetic mutation, highlighting the potential of immunotherapy in treating this malignancy[1,2]. Among the immunotherapeutic strategies are currently being tested in preclinical studies or clinical trials involving TNBC patients[3], the use of immune checkpoint inhibitors is particularly attractive, and atezolizumab plus nab-paclitaxel is approved in advanced TNBC[4]. However, the response rate to anti-PD-1 or anti-PD-L1 antibody alone remains low among TNBC patients[5]. The reason is at least partly due to the immunosuppressive factors existing in the tumour micro-environment restricting T-cell-mediated tumour cytotoxicity. Unfortunately, the underlying molecular mechanisms are still poorly understood.

Autophagy is an intracellular pathway responsible for bulk protein degradation and the removal of damaged organelles by lysosomes[6–8]. Alterations in autophagy are involved in carcinogenesis and cancer progression[9]. Breast cancer is one of the first diseases genetically linked to autophagy impairment. Beclin 1, which is identified as a haploinsufficient tumour suppressor, is monoallelically deleted in ~50% of sporadic breast carcinomas[10]. Moreover, the low expression of beclin1 is more common in basal-like and HER2-enriched breast cancers than in luminal A/B intrinsic tumour subtypes and is strongly associated with a poor prognosis and an independent predictor of survival[11]. Our previous study also demonstrates that low expression of ULK1 is associated with breast cancer progression and is an adverse prognostic marker of survival for patients[12]. Therefore, to a certain extent, the autophagic ability either at the basal level or following exposure to various stresses, including hypoxia, starvation and lack of growth factors, may be impaired in breast cancer cells, especially in TNBC tumours.

Tenascin-C (TNC) is an extracellular matrix glycoprotein that is highly expressed in malignant solid tumours, including breast cancer tumours[13]. TNC has been implicated in the modulation of cell migration, proliferation, invasion and angiogenesis[14–16]. TNC is also well known for its function in arresting T-cell proliferation and activation to overcome immune surveillance. Mechanistically, TNC can engage the integrinβ receptor of T cells through its fibronectin type III (FNIII) repeats and then block GTPase Rho activation to inhibit the reorganization of the actin/myosin and microtubule cytoskeleton[17–19]. TNC also inhibits α5β1-dependent T-lymphocyte adhesion to fibronectin through the binding of its fnIII 1–5 repeats to fibronectin[20]. In addition, soluble TNC inhibits the proliferation of human T cells by preventing the appearance of functional NF-AT1 transcription factor complexes in T-cell nuclear extracts[21]. Therefore, TNC is a very interesting target in cancer therapy, and the $^{131}$I-anti-tenascin monoclonal antibody 81c6 has already been in a clinical trail[22].

Accumulating evidence suggests that autophagy influences cellular immune responses[23,24]. However, the mechanisms by which that autophagy can stimulate or limit the immune attack of T cells on tumour cells remain unclear[25]. Here, we examine the complex role of autophagy in TNBC tumour immunity. We find that a failure of autophagy contributed to a limitation of the T-lymphocyte attack on TNBC tumours. In addition, TNC is identified as a strong candidate for autophagy deficiency-mediated immunosuppression. We further evaluate whether the inhibition of TNC could boost the efficacy of immunotherapy in TNBC.

## Results

**Autophagy inhibition reduces T-cell-mediated cytotoxicity.** To investigate the potential contribution of autophagy to the tumour immune response in breast cancer patients, we first evaluated the correlation between autophagy-related gene expression and the abundance of TILs using the online public website TIMER[26,27]. Only in basal tumours but not luminal and Her2+ tumours, Beclin 1 and Atg5 expression levels were positively related to the abundance of B cells, CD4+ T cells, CD8+ T cells, neutrophils and dendritic cells (DCs) (Supplementary Fig. 1a), indicating that the expression of the two autophagy-related genes might be involved in the tumour immune response, especially in TNBC. Since the effectiveness of many immune checkpoint inhibitors in the treatment of TNBC is currently under evaluation in ongoing clinical trials, we further elucidated the role of autophagy in the T-lymphocyte immune system in TNBC. We screened a panel of TNBC cell lines and found that only the MDA-MB-231 cell line expressed HLA-A2, which is important for tumour cell recognition by HLA-A2+ T cells (Supplementary Fig. 1b). We depleted the autophagy-related genes Atg5 and Atg7 using specific single-guide RNAs (sgRNAs) in MDA-MB-231 cells to construct autophagy-deficient cell lines, and then added the autophagy factors back through overexpression of Atg5 or Atg7 to rescue autophagy. As indicated in MDA-MB-231-Atg5KO and Atg7KO cells, the formation of the lipidated form of LC3 (LC3B-II) was completely blocked following treatment with Earle's balanced salt solution (EBSS) and hypoxic conditions in the presence of lysosomal-acidification inhibitor bafilomycin A1 (BafA1) (Supplementary Fig. 2a–f). The cytotoxic T-lymphocyte assay showed that deficiency of autophagy significantly reduced the percentage of lysed (cleaved caspase-3+ and LDH+) MDA-MB-231 cells when the tumour cells were co-cultured with CD3/CD28-activated human HLA-A2+ T lymphocytes under both normoxic or hypoxic conditions, and this decrease was markedly reversed by recovery of autophagy (Fig. 1a–d, Supplementary Fig. 2g–h). Similar results were observed in autophagy-deficient Atg5$^{-/-}$ mouse embryonic fibroblasts (MEFs) (Supplementary Fig. 2i, j). Atg5 knockout in MEFs also caused resistance to killing by CD3/CD28-activated mouse T cells (Supplementary Fig. 2k). In addition, we perturbed the expression of genes required for the activation of autophagy in MDA-MB-231 cells (Supplementary Fig. 2l). The results showed that silencing of the expression of almost all autophagy-related genes decreased the susceptibility of the tumour cells to the apoptosis induced by CD3/CD28-activated human T lymphocytes (Fig. 1e, Supplementary Fig. 2m).

Then we further measured antigen-specific T-cell-mediated cytotoxicity in autophagy-deficient MDA-MB-231 cells. Peptide 264–272 from naturally processed p53 has proven to be a potential T-cell epitope because of its strong affinity to HLA-A2, and MDA-MB-231 cells display high p53 concentrations in the nucleus due to a p53 gene mutation in codon 280[28,29]. Our results also showed high levels of p53 protein in autophagy-deficient MDA-MB-231 cell lines, similar to the levels in autophagy-competent MDA-MB-231 cell lines (Supplementary Fig. 2n). In the experiment, DCs loaded with the P53$_{264-272}$ antigen were co-cultured with autologous T lymphocytes from healthy HLA-A2+ donors to induce P53 peptide-specific T cells. T cells stimulated with no peptide-pulsed DCs were used as control T cells. The results showed that the frequency of P53$_{264-272}$ tetramer+ CD8+ T cells increased from 0.12 to 2.2% after stimulation with P53$_{264-272}$ peptide-pulsed DCs. As a control staining, NY-ESO-1$_{157-165}$ tetramer+ CD8+ T cells were assessed, and they did not change obviously (Supplementary Fig. 2o). The cytotoxicity of P53 peptide-pulsed DC-treated T cells targeting MDA-MB-231 cells was higher than that of control T cells (Fig. 1f). These data suggest that T cells stimulated with P53$_{264-272}$ peptide-pulsed DCs

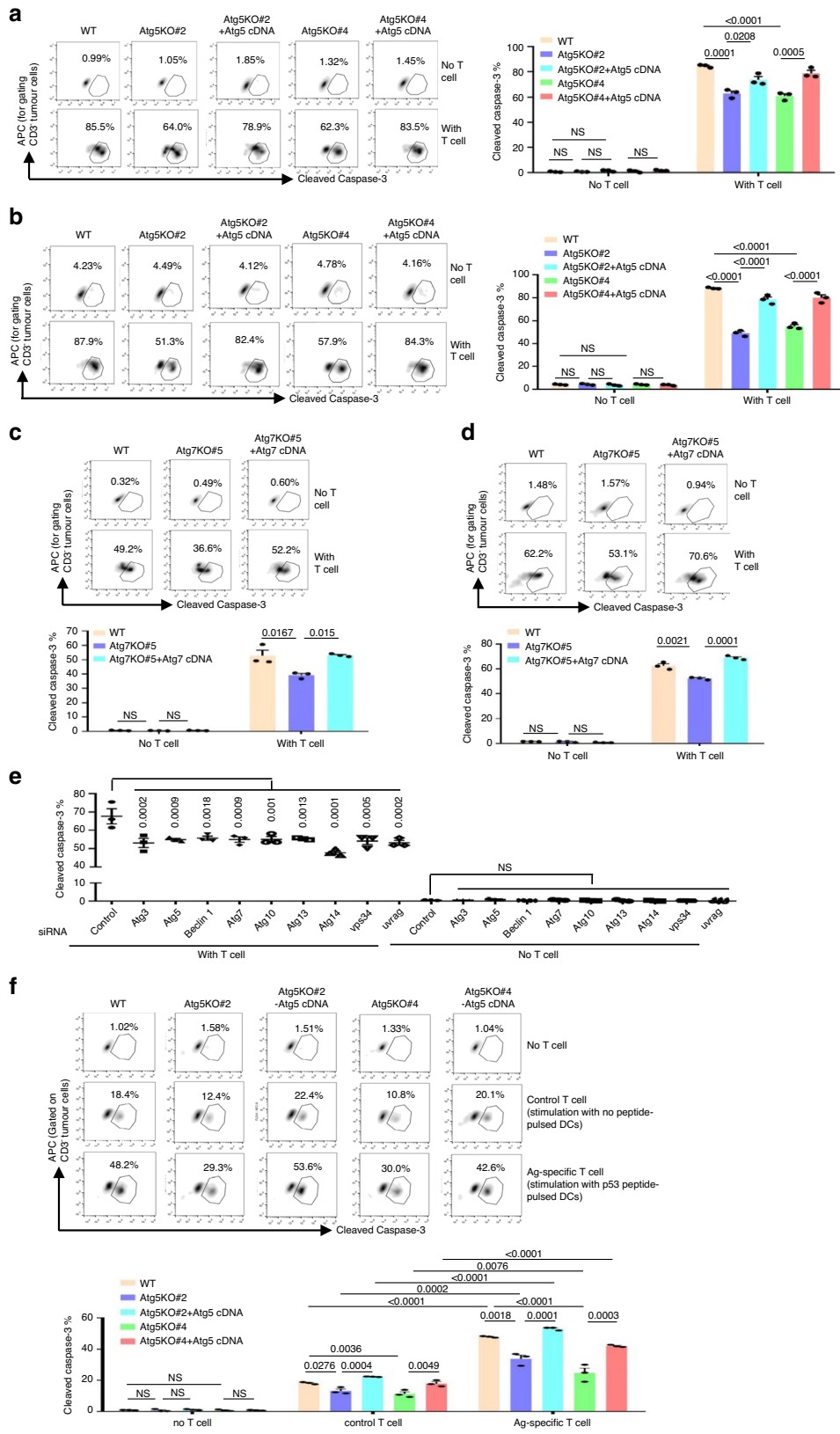

could kill MDA-MB-231 cells specifically by recognition of endogenous p53 epitope presented by tumour cells. As expected, we observed that the cytotoxicity of P53-specific T cells against MDA-MB-231-Atg5KO cells was reduced, but the cytotoxicity was recovered when Atg5 was restored (Fig. 1f). In addition, we depleted Atg7 in ovalbumin (OVA)-positive melanoma B16F10

cells (Supplementary Fig. 2p). Then the cells were co-cultured with activated CD8[+] T cells isolated from OT-1 TCR transgenic mice. The data also showed that compared to their autophagy-competent counterparts, autophagy-deficient B16F10-OVA-Atg7KO cells were more resistant to antigen-specific T-cell-mediated killing than the WT cells (Supplementary Fig. 2q).

**Fig. 1 Autophagy deficiency in TNBC cells reduces T-cell-mediated killing in vitro. a, b** The indicated MDA-MB-231-Atg5KO cells were co-cultured with CD3/CD28-activated human T-lymphocyte cells exposed to normoxic (**a**) or hypoxic conditions (1% $O_2$) (**b**). Left, representative dot plots of the cleavage of caspase-3 in tumour cells measured by flow cytometry. Right, percentage of cleaved caspase-3$^+$ tumour cells. **c, d** The indicated MDA-MB-231-Atg7KO cells were co-cultured with CD3/CD28-activated human T cells exposed to normoxic (**c**) or hypoxic conditions (**d**). Upper, representative dot plots of the cleavage of caspase-3 in tumour cells measured by flow cytometry. Bottom, percentage of cleaved caspase-3$^+$ tumour cells. **e** MDA-MB-231 cells were transiently transfected with siRNAs targeting autophagy-related genes for 48 h. Then, the tumour cells were co-cultured with CD3/CD28-activated human T cells. The percentage of cleaved caspase-3$^+$ tumour cells was determined using flow cytometry. **f** The indicated MDA-MB-231 cells were co-cultured with bulk populations of p53$_{264-272}$ peptide-specific human T cells. Upper, representative dot plots of the cleavage of caspase-3 in tumour cells measured by flow cytometry. Bottom, percentage of cleaved caspase-3$^+$ tumour cells. Error bars represent mean ± SEM, $n = 3$ biological independent samples. The $P$ value in (**a–d**, **f**) was determined by one-way ANOVA with Tukey's multiple comparisons test, the $P$ value in (**e**) was determined by one-way ANOVA with Dunnett's multiple comparisons test, no adjustments were made for multiple comparisons. NS no significance. All data are representative of three independent experiments.

Altogether, these data confirm that autophagy failure contributes to the limitation of T-lymphocyte attack on TNBC cells.

**Autophagy deficiency reduces T-cell antitumor response.** To evaluate the effect of autophagy on T-cell-mediated antitumour activity in vivo, we established autophagy-deficient murine models. Mouse mammary basal-like carcinoma 4T1 cells were used to establish the autophagy-incompetent model, which was generated by the depletion of Atg5 or Beclin1 with specific sgRNAs. Western blotting was used to confirm the blockage of the formation of LC3B-II in 4T1-Atg5KO cells and the decreased formation of LC3B-II in 4T1-Beclin1KO cells (Supplementary Fig. 3a). Consistent with the in vitro analysis, the autophagy-deficient 4T1-Atg5KO and 4T1-Beclin1KO tumours grew faster than the autophagy-competent 4T1 control cells in immunocompetent BALB/c mice, which were confirmed by the growth curves of the xenograft tumour volumes and the tumour weights (Fig. 2a, Supplementary Fig. 3b). Furthermore, the tumors induced by the autophagy-deficient 4T1 cells had not only decreased total CD4$^+$ and CD8$^+$ TIL populations but also fewer activated cytotoxic CD4$^+$ T and CD8$^+$ T cells in their TILs (IFNγ$^+$CD4$^+$ T and IFNγ$^+$CD8$^+$ T) than those induced by the autophagy-competent 4T1 control cells (Fig. 2b, Supplementary Fig. 3c). Then, we compared the antitumour effect of an anti-PD1 antibody between 4T1-WT and 4T1-Beclin1KO tumours (Fig. 2c). The results showed that systemic PD1 antibody-targeted treatment obviously limited the tumour volume of the 4T1-WT xenografts when such tumours were grown in normal, immunocompetent mice, and the tumour growth inhibition was nearly 65% according to the tumour weight (Fig. 2d). In addition, PD1 treatment resulted in an increased number of TUNEL-positive cells in the 4T1-WT tumours (Fig. 2e). However, PD1 antibody treatment slightly limited the growth of 4T1-Beclin1KO tumours and had produced no obvious significant difference in tumour volume (Fig. 2f). In addition, we obtained a similar result when comparing the antitumor effect of the anti-PD-L1 antibody between 4T1-WT and 4T1-Atg5KO tumours (Fig. 2g). The results showed that anti-PD-L1 treatment did control the growth of 4T1-WT tumours in normal, immunocompetent mice (Fig. 2h). Actually, the anti-PD-L1 treatment led to an increase in activated cytotoxic CD4$^+$ T and CD8$^+$ T cells in the TILs (IFNγ$^+$ CD4$^+$ T cells and IFNγ$^+$ CD8$^+$ T cells) (Fig. 2i) and resulted in an increased number of TUNEL-positive cells in 4T1-WT tumours (Fig. 2j). Double immunofluorescent staining showed that the apoptotic cells were mainly from EpCam$^+$ tumour cells, not from CD45$^-$ immune cells (Supplementary Fig. 3d–e). However, anti-PD-L1 treatment failed to control the growth of 4T1-Atg5KO tumours (Fig. 2k). Altogether, these data suggest that autophagy-competent 4T1 tumours are more sensitive to T-cell-mediated tumour killing than autophagy-incompetent 4T1 tumours and that autophagy failure may inhibit the capacity of

cancer cells to elicit a local immune response characterized by the recruitment of cytotoxic T cells.

**TNC is involved in autophagy defect-mediated immunosuppression.** Subsequently, we explored the mechanism underlying autophagy deficiency-mediated immunosuppression in TNBC tumours. There was no obvious change in the expression of MHC class I molecules between the autophagy-deficient and autophagy-competent MDA-MB-231 cells (Supplementary Fig. 4a). Since the most important function of autophagy is to degrade large aggregated proteins, we hypothesized that the ectopic accumulation and distribution of functional proteins may contribute to autophagy deficiency-mediated immunosuppression. To assess this possibility, SILAC and high-resolution MS procedures were implemented to quantify the protein variations between the autophagy-competent and autophagy-incompetent cells (Supplementary Fig. 4b). We identified a signature of 451 upregulated proteins in MDA-MB-231 Atg5KO cells and 994 upregulated proteins in Atg5$^{-/-}$ MEFs using around a normalized 1.2 ratio as the cutoff (Supplementary Data 1, 2). A query of Kyoto Encyclopaedia of Genes and Genomes (KEGG) databases showed that the proteins involved in focal adhesion were significantly upregulated in both the MDA-MB-231 and MEF autophagy-incompetent cell lines (Fig. 3a), which was similar to a previous study showing that innate anti-PD-1 therapy resistance in metastatic melanoma tumours displays a transcriptional signature involving the upregulation of genes related to cell adhesion[30]. Subsequently, we explored the 50 upregulated proteins that were identified in both autophagy-incompetent cell lines. Among these proteins, TNC, which belongs to the focal adhesion pathway, was one of the most upregulated proteins in both of the autophagy-deficient cell lines (Supplementary Fig. 4c). Furthermore, we confirmed that the protein level of TNC was substantially elevated in autophagy-deficient MDA-MB-231 cells and MEFs, and that this increase was blunted by restoration of autophagy (Fig. 3b).

TNC has been reported to protect prostate cancer stem-like cells and glioblastoma cells from immune surveillance by arresting T-cell activation[19,31]. Thus, we further elucidated whether TNC protected autophagy-incompetent TNBC cells from immune surveillance. Western blot analysis confirmed the specific knockdown of TNC in autophagy-deficient MDA-MB-231 cells (Supplementary Fig. 4d). Reduction in TNC expression in autophagy-deficient MDA-MB-231 cells did not influence their proliferation in vitro (Supplementary Fig. 4e). However, knockdown of TNC in autophagy-deficient MDA-MB-231 cells restored the autophagy-incompetent cells' susceptibility to killing by cytotoxic T cells, as assessed using the caspase-3 cleavage assay and LDH assay (Fig. 3c, Supplementary Fig. 4f). Carboxyfluorescein succinimidyl ester (CFSE) dilution as analyzed by flow cytometry showed that more CD4$^+$ T and CD8$^+$ T cells underwent proliferation in autophagy-incompetent TNBC cells

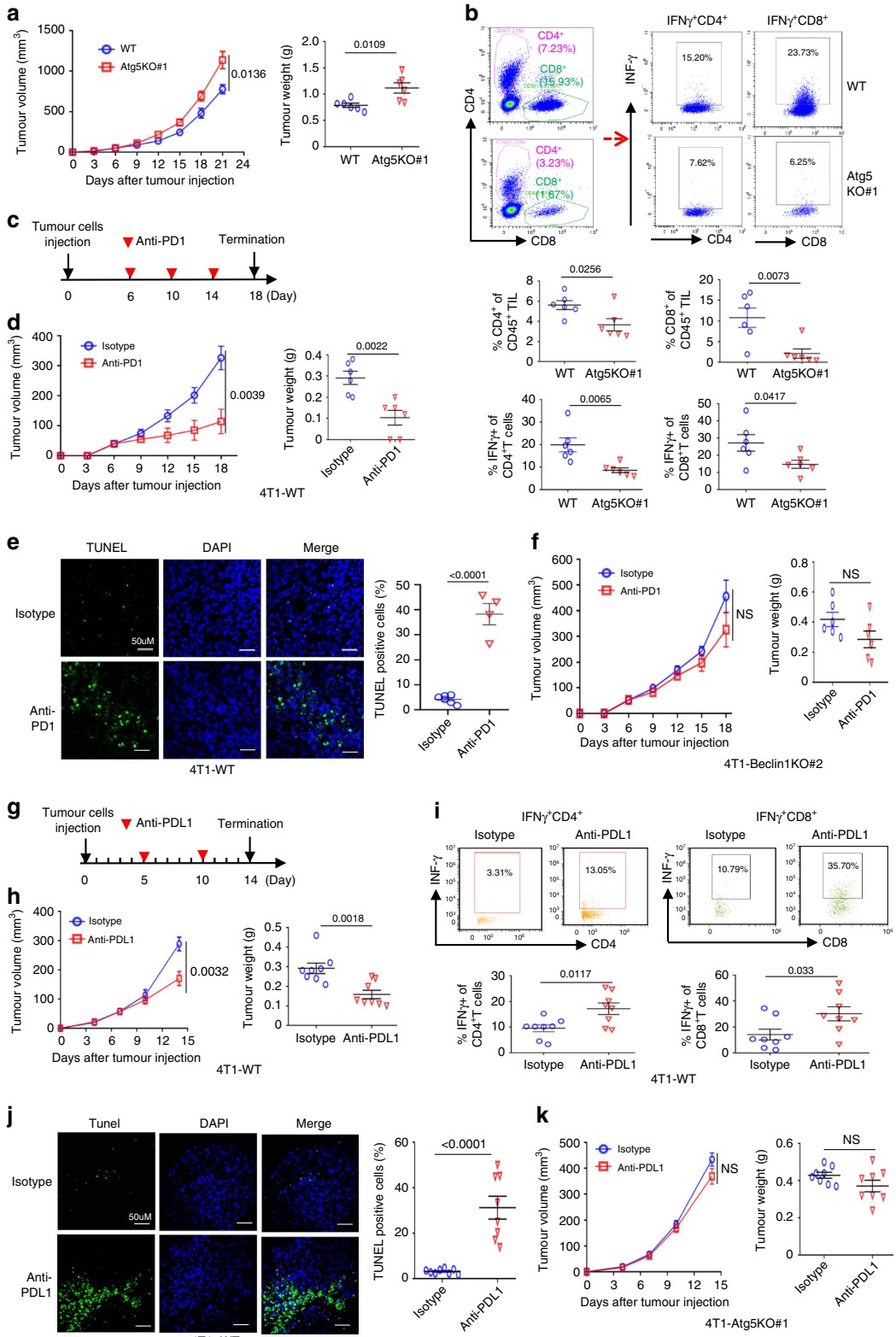

with TNC knockdown than in those cells with normal TNC expression (Supplementary Fig. 4g). We further examined the tumourigenesis of autophagy-deficient 4T1 cells after depleting TNC using specific sgRNA (Fig. 3d). The level of cas9 protein in the vector control cell line was similar to that in the TNC knockout cell line (Supplementary Fig. 4h), and the parental, vector control and TNC knockout cell lines showed similar

growth rate in vitro (Supplementary Fig. 4i). We observed that TNC knockout significantly reduced the tumour burden, as confirmed by the growth curves of the xenograft tumour volumes and the tumour weights in syngeneic BALB/c mice (Fig. 3e). However, we did not observe any significant changes in tumour growth in the severe combined immunodeficient (SCID) mice (Supplementary Fig. 4j), suggesting that the differential

**Fig. 2 The loss of autophagy promotes resistance to T-cell-mediated anti-tumour activity in vivo. a, b** Tumour growth of mouse 4T1-WT or 4T1-Atg5KO#1 cells in BALB/c mice (n = 6 mice per group). Tumour volumes (left, **a**) and tumour weights upon autopsy on day 21 (right, **a**) were calculated. Representative TILs dot plots from a representative mouse for each group were shown (upper, **b**). The percentage of TILs for each group were calculated (bottom,**b**). **c** Tumour growth of 4T1-WT and 4T1-Beclin1KO#2 tumour xenografts in BALB/c mice following treatment with mouse anti-PD1 antibody. The treatment protocol was summarized by the arrows. **d, e** Tumour growth of 4T1-WT cells in BALB/c mice with anti-PD-1 antibody treatment (n = 6 mice per group). Tumour volumes (left, **d**) and tumour weights upon autopsy on day 18 (right, **d**) were calculated. Representative images of TUNEL staining (green) and DAPI-stained nuclei (blue) in xenograft tumour sections were shown (left, **e**). Quantification of positive TUNEL cells (right, **e**). **f** Tumour growth of 4T1- Beclin1KO#2 cells in BALB/c mice with anti-PD-1 antibody treatment (n = 6 mice per group). Tumour volumes (left, **f**) and tumour weights upon autopsy on day 18 (right, **f**) were calculated. **g** Tumour growth of 4T1-WT and 4T1-Atg5KO#1 tumour xenografts in BALB/c mice following treatment with mouse anti-PD-L1 antibody. The treatment protocol was summarized by the arrows. **h–j** Tumour growth of 4T1-WT cells in BALB/c mice with mouse anti-PD-L1 antibody treatment (n = 8 mice per group). Tumour volumes (left, **h**) and tumour weights upon autopsy on day 14 (right, **h**) were calculated. Representative TILs dot plots from a representative mouse for each group were shown (upper, **i**). The percentage of TILs for each group were calculated (bottom, **i**). Representative images of TUNEL staining (green) and DAPI-stained nuclei (blue) in xenograft tumour sections were shown (left, **j**). Quantification of positive TUNEL cells (right, **j**). **k** Tumour growth of 4T1-Atg5KO#1 cells in BALB/c mice with mouse anti-PD-L1 antibody treatment (n = 8 mice per group). Tumour volumes (left, **k**) and tumour weights upon autopsy on day 14 (right, **k**) were calculated. Error bars represent mean ± SEM. The P value was determined by a two-tailed unpaired Student's t test. Data are representative of two independent experiments.

tumourigenicity was attributed to immune surveillance. Indeed, the TNC knockout tumours had more CD4$^+$ and CD8$^+$ TIL cells and more activated cytotoxic T cells (IFNγ$^+$CD4$^+$ and IFNγ$^+$CD8$^+$) in their TILs than the control tumours (Fig. 3f). Taken together, these results suggest that TNC can suppress T-cell activity in the tumour microenvironment and that TNC is a strong candidate for autophagy.

**Selective autophagic degradation of TNC via p62.** We then attempted to determine whether the increase in TNC protein levels in autophagy-deficient cells occurred at the transcription level or through effects on protein stability. The results showed that TNC mRNA levels were similar in autophagy-competent and autophagy-incompetent MDA-MB-231 cells (fold change was not more than two times) (Supplementary Fig. 5a). Therefore, we reasoned that TNC might be tightly regulated by the protein degradation pathway. Indeed, there was an obvious decrease in TNC protein levels in the presence of the protein synthesis inhibitor cycloheximide (CHX) (Supplementary Fig. 5b). To determine which degradative system dominantly regulates the degradation of TNC, we examined the protein stability of TNC using pharmacological approaches. Autophagy initiated by the mTOR inhibitor rapamycin or BMS 303141 remarkably degraded TNC (Supplementary Fig. 5c, d). We also observed that BafA1 and the autophagic-sequestration inhibitor 3-methyladenine (Supplementary Fig. 5e, f), but not the proteasome inhibitor MG132 (Supplementary Fig. 5g), enhanced the protein level of TNC. More importantly, we found that the TNC degradation induced by EBSS was mostly abrogated in autophagy-deficient Atg5KO cells (Fig. 4a). Furthermore, the puncta formation of fluorescence-tagged LC3 localized with TNC was also significantly increased by EBSS-induced autophagy, which indicated that TNC was incorporated in autophagosomes (Supplementary Fig. 5h). These results suggest that the protein stability of TNC is controlled by the autophagy-lysosome pathway.

To elucidate how TNC is degraded through selective autophagy, we further sought to identify the autophagy receptors for TNC. Our results showed that TNC mainly interacted with p62 rather than other receptors, including NDP52, NBR1, TAX1BP1, Tollip, OPTN and BNIP3L (Fig. 4b). On the basis of these data, we further confirmed the capacity of TNC to associate with p62. We observed that TNC could interact with p62. However, this interaction did not occur when p62 lacked the C-terminal UBA domain (Fig. 4c). Moreover, we found that EBSS-induced autophagy promoted the interaction between TNC and p62 (Fig. 4d). Confocal microscopy was used to further reveal that

the cytoplasmic formation of TNC-p62 puncta was significantly increased by EBSS- or hypoxia-induced autophagy (Fig. 4e). More importantly, the TNC degradation induced by EBSS was clearly abrogated when p62 was knocked down (Fig. 4f). To identify the domain(s) of TNC responsible for its interaction with p62, we constructed several deletion mutants according to the conserved domains of TNC (Fig. 4g). We found that the interaction between TNC and p62 depended on the FN3 domain of TNC (Fig. 4h). These data indicate that TNC is degraded by the selective autophagy-lysosome pathway through recognition by the autophagy receptor p62.

**Skp2 promotes TNC ubiquitination for autophagic degradation.** Since p62 dUBA mutants cannot interact with TNC, we hypothesized that the ubiquitin chains on TNC might serve as a recognition signal for subsequent p62-dependent degradation. As expected, we found that ubiquitinated TNC accumulated in the presence of EBSS and BafA1 (Fig. 5a), indicating that autophagy facilitated the degradation of ubiquitinated TNC. Our results also revealed that TNC was mainly ubiquitinated with K63 linkage but not with the other types of ubiquitin chains. The expression of a K63R ubiquitin mutant (unable to form K63-linked chains) impaired TNC polyubiquitination, whereas expression of the other mutants, i.e., K6R, K11R, K27R, K29R, K33R, and K48R, did not (Fig. 5b). In addition, the K63-linked ubiquitination of TNC was increased in the presence of EBSS and BafA1 (Fig. 5c). Subsequently, a panel of E3 ubiquitin ligases was screened by siRNAs to identify potential E3 ligase candidates for TNC. We found that only the depletion of Skp2 led to a significant accumulation of TNC (Fig. 5d). The TNC degradation induced by EBSS was nearly abrogated following the knockdown of Skp2 (Supplementary Fig. 6a). We further determined the functional role of Skp2 in TNC ubiquitination. We found that the downregulation of Skp2 by RNAi decreased TNC ubiquitination (Fig. 5e). In addition, TNC ubiquitination was considerably upregulated following the overexpression of Skp2-WT, whereas the inactive mutant Skp2-LRR (a Skp2 variant that is defective in Skp2 SCF E3 ligase activity) abolished this enhancement (Supplementary Fig. 6b). Based on the above observations, we sought to test whether Skp2 mediates the autophagic degradation of TNC. Knockdown of Skp2 significantly reduced the interaction between TNC and P62 (Fig. 5f). Confocal microscopy was used to reveal that the cytoplasmic formation of TNC-p62 puncta by EBSS-induced autophagy was reduced following Skp2 knockdown (Fig. 5g). Collectively, these results demonstrate that Skp2 might be a major E3 ligase for TNC and that the Lys63-linked polyubiquitin chains target TNC for proteasome degradation.

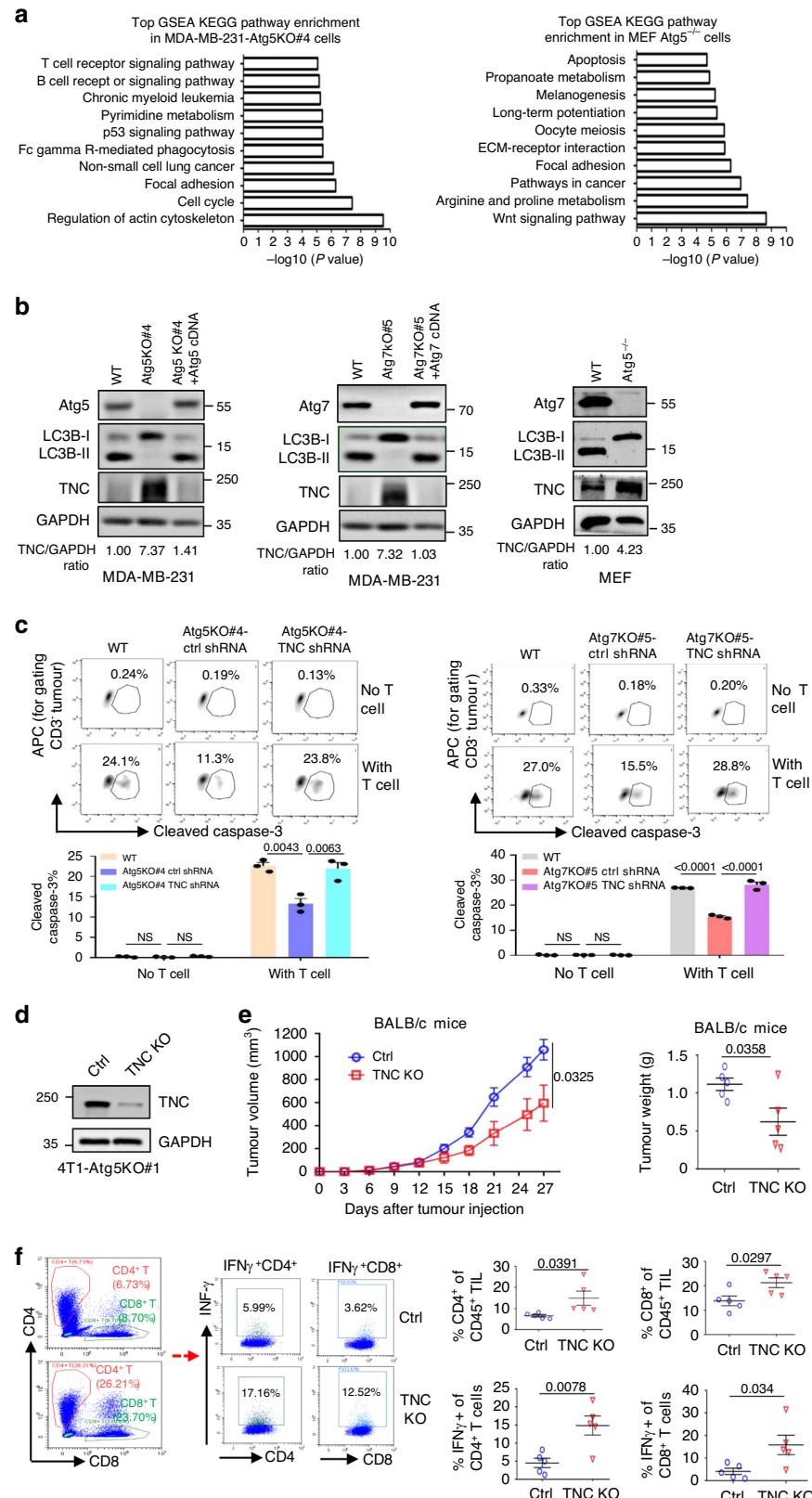

## Major sites for ubiquitination and autophagic removal of TNC.

Then we sought to determine which lysine residue in TNC might be a major ubiquitination site targeting TNC for autophagic degradation. We found that the ubiquitination level at the N-terminal domain of TNC was almost diminished in the presence of EBSS and BafA1, suggesting that the FN3 domain and the C-terminal domain of TNC might be responsible for its ubiquitination (Fig. 6a). According to the specificity and sensitivity of these loci, seven potential ubiquitination sites were predicted, K627, K886, K942, K969, K1837, K1882 and K1929 (Fig. 6b). Subsequently, we generated corresponding TNC-P2 mutants bearing a single lysine (K)-to-arginine (R) substitution in each

**Fig. 3 TNC is overexpressed in autophagy-deficient TNBC cells and inhibits T-cell priming. a** The Top10 KEGG pathways enriched for commonly upregulated proteins in MDA-MB-231-Atg5KO#4 cells and MEF-Atg5$^{-/-}$ cells compared to control cells. **b** Immunoassay of extracts of the indicated MDA-MB-231 cells and MEF cells. **c** The indicated MDA-MB-231 cells were co-cultured with CD3/CD28- activated human T-lymphocyte cells. Upper, representative dot plots of the cleavage of caspase-3 in tumour cells measured by flow cytometry. Bottom, percentage of the cleaved caspase-3 in tumour cells ($n = 3$ biological independent samples). **d** The effect of TNC knockout in 4T1-Atg5KO cells using CRISPR-Cas9 technology. **e** Tumour growth of indicated mouse 4T1-Atg5KO#1 cells in BALB/c mice ($n = 5$ mice per group). Tumour volumes were calculated (left), and tumour weights from experiment on autopsy on day 27 (right). **f** FACS analysis of CD45$^+$CD4$^+$, CD45$^+$CD8$^+$, and IFNγ$^+$ in CD45$^+$CD4$^+$T and CD45$^+$CD8$^+$T-cell populations from the isolated TILs in (**e**) ($n = 5$ mice per group, right). Representative dot plots from a representative mouse for each group (left). Error bars represent mean ± SEM. The P value in **c** was determined by one-way ANOVA with Tukey's multiple comparisons test, no adjustments were made for multiple comparisons. The P value in **e**, **f** was determined by a two-tailed unpaired Student's t test. NS no significance. All data are representative of three independent experiments.

potential ubiquitination site and found that the ubiquitination of the TNC-P2-K942R, TNC-P2-K969R, TNC-P2-K1837R and TNC-P2-K1882R mutants was abrogated (Fig. 6c). We further confirmed that TNC-P2-K942R and TNC-P2-K1882R mutants failed to interact with p62 (Fig. 6d). In addition, exogenously expressed TNC-P2-K942R and TNC-P2-K1882R were much more stable than exogenously expressed TNC-P2-WT after exposure to EBSS treatment (Fig. 6e). Altogether, these results indicated that ubiquitination at Lys942 and Lys1882 might be functionally important for the selective degradation of TNC.

**TNC is associated with poor prognosis in TNBC patients**. To assess the clinical significance of TNC deregulation in breast cancer, we performed Kaplan–Meier meta-analyses using the online Kaplan–Meier-Plotter database[32]. The results showed that high expression of TNC was consistently associated with poor relapse-free survival (RFS), poor overall survival (OS), early distant metastasis-free survival (DMFS) and post-progression survival (PPS) in ER-negative patients, while the opposite trends were observed in the patients with ER-positive tumours (Supplementary Fig. 7a–d). In addition, high TNC expression was significantly associated with poor RFS in patients with basal tumours, but the opposite trend was observed in patients with luminal A tumours (Fig. 7a). Further analysis showed that high TNC expression level was associated with poor RFS in TNBC patients with basal-like 1, basal-like 2, immunomodulatory, mesenchymal and mesenchymal stem-like subtypes, especially there was a statistically significant in the basal-like 1, basal-like 2, and immunomodulatory subtype (Fig. 7b). However, the opposite trend was observed in TNBC patients with the luminal androgen receptor subtype (Fig. 7b). Then, we further studied TNC expression at the protein level in TNBC primary tumours. TNC expression in breast tumour tissues was significantly higher than that in adjacent normal breast tissues (Fig. 7c). The Kaplan–Meier survival analysis further revealed that TNBC patients with high expression of TNC had a shorter OS than those with low expression of TNC (Fig. 7d). Moreover, TNC expression status and lymph node metastasis were independent prognostic factors for poor OS among the TNBC patients in multivariate Cox regression analyses (Supplementary Table 1).

To further validate our findings in TNBC patient samples, we analyzed the correlations among TNC, LC3B and CD8$^+$ T cells in these tumours. LC3B is a well-established marker for autophagy[33]. We observed a clearly negative correlation between the expression levels of TNC and LC3B. The frequency of cases with low expression of TNC among the patients with positive expression of LC3B (43/63 cases, 68.3%) was significantly higher than that among those patients with negative expression of LC3B (20/63, 31.7%) (Fig. 7e, Supplementary Fig. 7e). In addition, we found that TNBC tumors with low TNC expression had significantly higher CD8$^+$ T-cell tumour infiltration than those

with high TNC expression (Fig. 7f, Supplementary Fig. 7f). The specificity of the antibodies for IHC staining was also determined (Supplementary Fig. 8a–c). Altogether, these results suggest that the degradation of TNC by the autophagy-lysosome pathway could be physiologically significant and clinically relevant in TNBC patients.

**Targeting TNC sensitizes TNBC cells to checkpoint inhibitors**. As TNC is involved in autophagy deficiency-mediated immuno-suppression, we speculated that patients with high TNC expression in autophagy-impaired tumours might be resistant to PD-1 blockade immunotherapy. To determine the clinical relevance, we obtained publicly available data from 26 melanoma patients who underwent tumour biopsy before starting immunotherapy (GSE78220)[30]. In total, 60% of the patients in the "TNC-high and Beclin1-low" group were non-responders, and only 36% of the patients in the "Other" group were non-responders (Supplementary Fig. 9). In addition, our results indicated that autophagy-impaired TNBC cells were resistant to checkpoint inhibitors. Therefore, we tested the combination of checkpoint inhibitors and anti-TNC antibodies in autophagy-deficient tumour cells that expressed PD-L1 on the membrane (Supplementary Fig. 10a). The results showed that treatment with either anti-TNC or anti-PD-L1 antibodies only slightly improved T-cell-induced tumour killing, and a synergistic T-cell-induced tumour killing effect was observed when of anti-TNC and anti-PD-L1 antibodies were applied together in autophagy-impaired MDA-MB-231 cells (Fig. 8a). In addition, the combination of nivolumab and anti-TNC antibodies was more effective than each agent alone (Fig. 8b). In addition, we also used a P53 antigen-specific T-cell-mediated cytotoxicity assay to measure the synergistic effect. The results showed that the proportion of P53$_{264-272}$ tetramer-positive CD8$^+$ T cells increased from 0.17 to 1.14% after stimulation with P53$_{264-272}$-pulsed DCs (Supplementary Fig. 10b). As expected, blockade of TNC resensitized anti-PD-L1 antibody-treated MDA-MB-231-Atg7KO cells to antigen-specific T-cell-induced tumour killing (Fig. 8c). The effect of mouse OT-1-specific CD8$^+$ T-cell-mediated killing in B16F10-OVA-Atg7KO cells could also be enhanced when anti-TNC and anti-PD-L1 antibodies were used in combination (Supplementary Fig. 10c). Because the protein stability of TNC is stringently regulated by autophagy, we sought to determine whether the degradation of TNC by an autophagy inducer could enhance antitumour immunity. Therefore, we screened several autophagy inducers[34]. We found that Resveratrol, which is a natural phenol and a phytoalexin produced by several plants, could induce the formation of the lipidated form of LC3 (LC3B-II) but abolished the expression of TNC in a time- and concentration-dependent manner (Supplementary Fig. 11a). The results showed that Resveratrol treatment did not influence the growth of MDA-MB-231 cells (Supplementary Fig. 11b), but Resveratrol-pre-treated

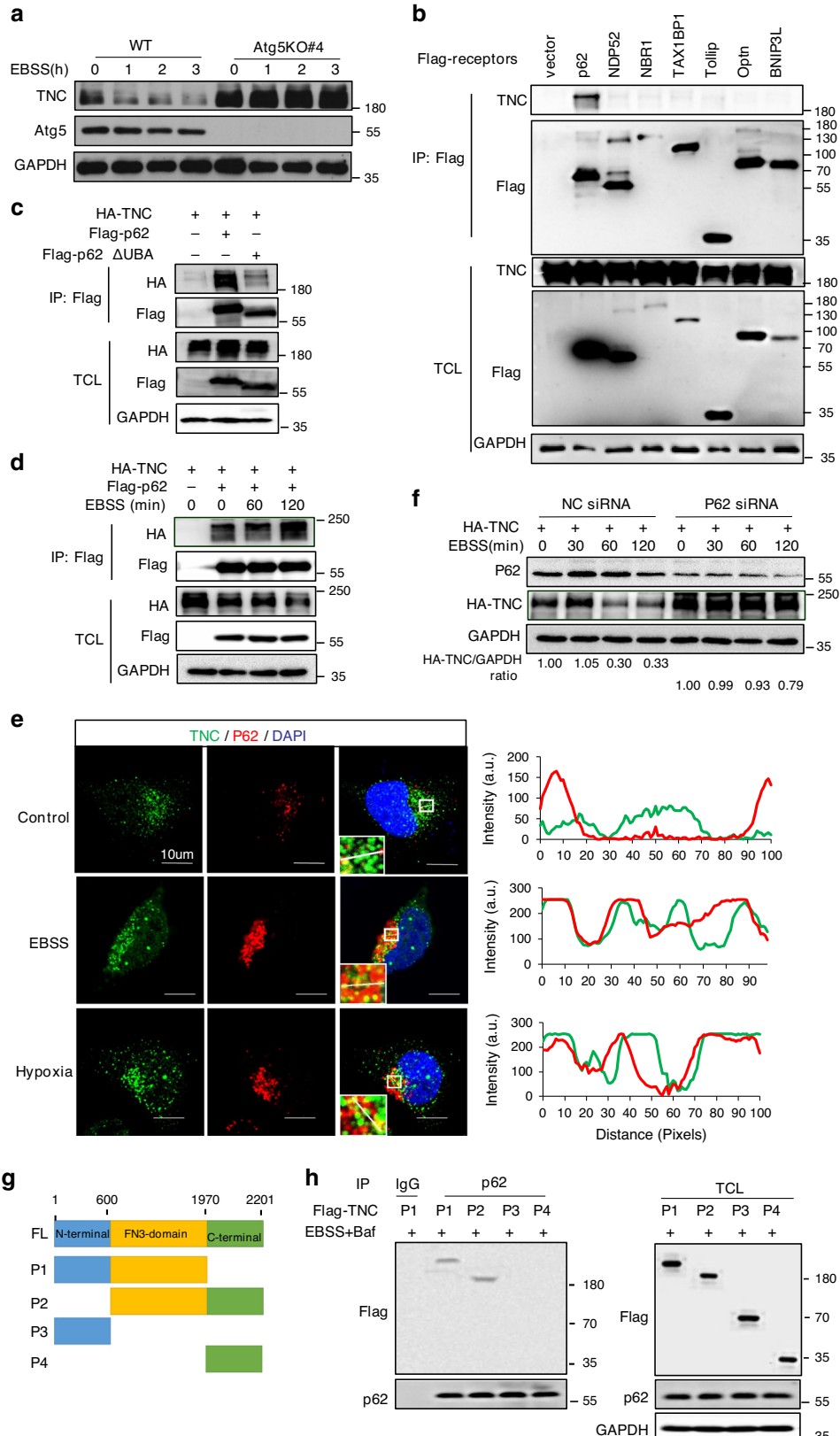

MDA-MB-231 cells were more sensitive to T-cell-mediated cytotoxicity (Fig. 8d). Collectively, these data indicate that targeting TNC with an anti-TNC antibody or autophagy inducer sensitizes T-cell-mediated killing and improves the antitumour effects of immune checkpoint inhibitors in autophagy-impaired TNBC cells in vitro.

**Knockdown of TNC sensitizes PD-1 blockade therapy in vivo.** To validate the combined effect of targeting TNC and checkpoint inhibitors in vivo, autophagy-deficient 4T1 cells stably expressing inducible short hairpin RNAs (shRNAs) against TNC were generated. Western blotting was performed to confirm the reduced levels of the TNC proteins after doxycycline (DOX) treatment

**Fig. 4 P62 mediates selective autophagic degradation of TNC. a** Immunoblot of MDA-MB-231-WT or MDA-MB-231-Atg5KO#4 cells treated with EBSS for the indicated time points. **b** HEK293T cells were transfected with Flag-tagged p62, NDP52, NBR1, TAX1BP1, Tollip, OPTN or BNIP3L, followed by immunoprecipitation with anti-Flag beads and immunoblot analysis with anti-TNC. **c** Coimmunoprecipitation and immunoassay of extracts of HEK293T cells transfected with FLAG-tagged wild-type p62 or its UBA domain deletion mutant, together with HA-tagged TNC. **d** Immunoprecipitation and immunoassay of extracts HEK293T cells transfected with Flag-p62, HA-TNC, and treated with EBSS for different hours. **e** Confocal microscopy of MDA-MB-231 cells treated with EBSS for 3 h or exposed to hypoxia for 12 h in the presence of BafA1. Scar bar, 10 μm. Line-scan analysis for each image is also shown. Green, TNC; Red, P62. **f** HEK293T cells were transiently transfected with p62 siRNA for 12 h, then co-transfected with HA-tagged TNC for another 48 h. Then the cells were treated with EBSS for different hours. **g** Construct deletion mutants of Flag-TNC according to the conserved domains of TNC. **h** HEK293T cells were transfected with Flag-tagged TNC or deletion mutants. Endogenous p62 was immunoprecipitated and the bound Flag-TNC proteins examined by immunoblot. All data are representative of three independent experiments.

(Fig. 9a). We selected PD1 blockade for combination therapy with TNC blockade in Atg5-knockout 4T1 cells implanted in immunocompetent BALB/c mice. Mice were treated with DOX and anti-PD1 antibodies as indicated (Fig. 9b). Consistent with our observations in vitro, the treatment with each single agent had a minimal effect, but the combined treatment with DOX induction and anti-PD-1 antibodies significantly improved the tumour growth inhibition, as confirmed by the growth curves of the xenograft tumour volumes and the tumour weights (Fig. 9c, d). Indeed, the combined treatment led to not only increased $CD4^+$ and $CD8^+$ TIL populations, but also a substantially increased granzyme B (GB) release and an increased number of TUNEL-positive cells compared to single-agent treatment (Fig. 9e, f). No obvious toxicity was observed in the mice receiving the treatment. Together, these data indicate that blockade of TNC has the potential to enhance the efficacy of PD-1 immunotherapy in autophagy-impaired TNBC tumours in vivo.

## Discussion

The present study provides experimental and clinical evidence supporting a potentially and interesting mechanism of tumour immune tolerance in TNBC. Our results showed that autophagy defects inhibited T-cell-mediated killing in TNBC tumours. We identified TNC as a key regulator of autophagy deficiency-mediated immunosuppression. The Lys63-linked ubiquitination of TNC catalyzed by the E3 ligase Skp2 was critical for the subsequent recognition of TNC by the autophagy receptor protein p62 in the process of selective autophagic degradation. High levels of TNC not only was an independent prognostic factor of poor OS, but also showed a significant negative correlation with LC3B expression and $CD8^+$ T-cell tumour infiltration among TNBC patients. Notably, targeting TNC sensitized cells to T-cell-mediated tumour killing and improved the antitumour effects of immune checkpoint inhibitors in autophagy-deficient TNBC tumours (Fig. 10). These observations suggest that TNC could be a candidate therapeutic target for developing combination immune checkpoint blockade (ICB) therapy strategies for TNBC.

The interactions between tumour cell autophagy and the immune system are complicated. Modulation of autophagy has been reported to induce potent immune responses. Chemotherapy-induced autophagy causes the release of ATP from tumour cells in mice, thereby leading to the recruitment of immune cells to stimulate antitumour immune responses[35]. PTEN loss in melanoma promotes resistance to T-cell-mediated killing by inhibiting autophagy[36]. In contrast, autophagy has also been reported to limit immune-mediated cytotoxicity. The hypoxia-induced resistance of lung tumours to CTL-mediated lysis is associated with autophagy induction in target cells[37]. Targeting autophagy inhibits melanoma growth by enhancing NK cell infiltration in a CCL5-dependent manner[38]. In this study, we validated that defective autophagy was a key immunosuppressive factor regulating the infiltration and activity of cytotoxic T cells in TNBC. We developed defective autophagy models in vitro and in vivo. The in vitro co-culture model revealed that blocking autophagy decreased tumour susceptibility to CTL-mediated immune attack. In addition, autophagy-deficient tumour cells inhibited the trafficking of tumour-reactive T cells to tumours in vivo. More importantly, anti-PD1 and anti-PD-L1 treatment obviously limited the growth autophagy-competent tumours grown in normal immune-competent mice but not autophagy-deficient tumours grown in the same setting. All of these data support that autophagy-deficient TNBC cells are more resistant to T-cell-mediated cytotoxicity than autophagy-competent TNBC cells. As the autophagic potential may be impaired in TNBC tumours, this finding may at least partially explain why most TNBC patients are resistant to immune checkpoint inhibitors.

Abnormal autophagic protein degradation has been associated with cancer[39]. For example, impaired autophagy leads to abnormalities of the selective autophagic degradation of HK2, which contributes to the substantial concomitant enhancement of glycolysis in liver cancer[40]. In this study, we identified that TNC was able to be degraded via the selective autophagy-lysosome pathway. The E3 ligase Skp2 was able to catalyze robust K63-linked ubiquitination of TNC at Lys942 and Lys1882, which facilitated recognition by p62, followed by further degradation by the autophagy-lysosome system. We also demonstrated that TNC was involved in autophagy deficiency-mediated immunosuppression. Knockdown of TNC in autophagy-deficient TNBC cells restored the cells' susceptibility to killing by cytotoxic T cells in vitro and in vivo. All of these findings indicate that targeting the abnormal autophagic protein degradation pathway may be a promising approach in the treatment of cancer.

As most TNBC patients are still refractory or not sensitive to immune checkpoint therapies, identifying novel immunotherapies and combination strategies is a major priority[41,42]. TNC has been subjected to extensive investigation due to its selective expression in the breast cancer[43]. In our analysis, TNC expression was positively associated with poor prognosis and inversely correlated with LC3B and $CD8^+$T cells in TNBC patients. As we demonstrated that autophagy-deficient TNBC cells used TNC to block T-cell activation, we proposed that the inhibition of TNC may be a good strategy to compensate for defective immunogenic signalling in autophagy-impaired TNBC tumours. We showed that inhibition of TNC in autophagy-impaired TNBC cells improved the antitumour effects of immune checkpoint inhibitors in vitro and in vivo. Therefore, the prevalent upregulation of TNC provides a rationale target in TNBC patients. According to Kaplan-Meier analysis of RFS based on TNC levels, TNBC patients with the basal-like 1, basal-like 2, immunomodulatory, and mesenchymal, and mesenchymal stem-like subtypes, especially those with the basal-like and immunomodulatory subtypes, may benefit from a combination strategy including inhibition of TNC and immune checkpoint inhibitors.

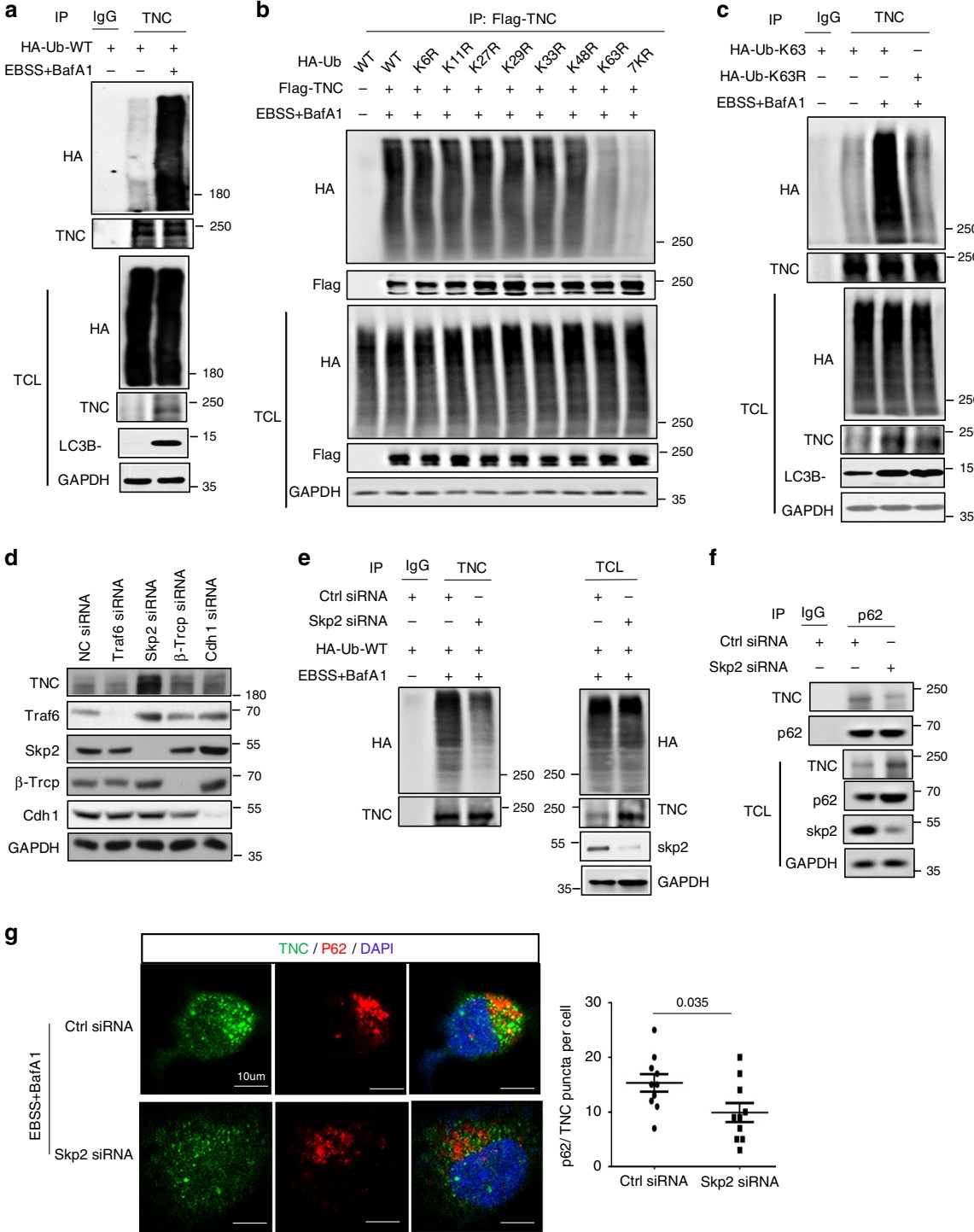

**Fig. 5 Skp2 mediates autophagic degradation of TNC. a** Coimmunoprecipitation and immunoassay of extracts of HEK293T cells treated with EBSS in the presence of BafA1 for 2 h. **b** HEK293T cells were transfected with Flag-tagged TNC and HA-tagged wild-type Ub (WT) or ubiquitin mutants (K6R, K11R, K27R, K29R, K33R, K48R, K63R, 7KR) and treated with EBSS in the presence of BafA1 for 2 h, followed by immunoprecipitation with anti-Flag beads and immunoblot analysis with anti-HA. **c** Coimmunoprecipitation and immunoassay of extracts of HEK293T cells transfected with various combinations of plasmid, then treated with or without EBSS in the presence of BafA1 for 2 h. **d** Immunoblot analysis of MDA-MB-231 cells transfected with the indicated siRNA oligonucleotides for 48 h. **e** HEK293T cells were transiently transfected with Skp2 siRNA for 12 h, then co-transfected with HA-Ub-WT for another 48 h. Then the cells were treated with EBSS treatment in the presence of BafA1 for 2 h. **f** HEK293T cells were transiently transfected with Skp2 siRNA for 48 h, p62 proteins were immunoprecipitated and the bound TNC proteins examined by immunoblot. **g** Confocal microscopy of MDA-MB-231 cells treated with Skp2 siRNA in the presence of EBSS for 2 h along with BafA1 (left). Scar bar, 10 μm. Statistics of the puncta formation by TNC-p62 were determined in the indicated samples ($n = 10$ cells per group, right). Error bars represent mean ± SEM. The P value in **g** was determined by a two-tailed unpaired Student's t test. All data are representative of three independent experiments.

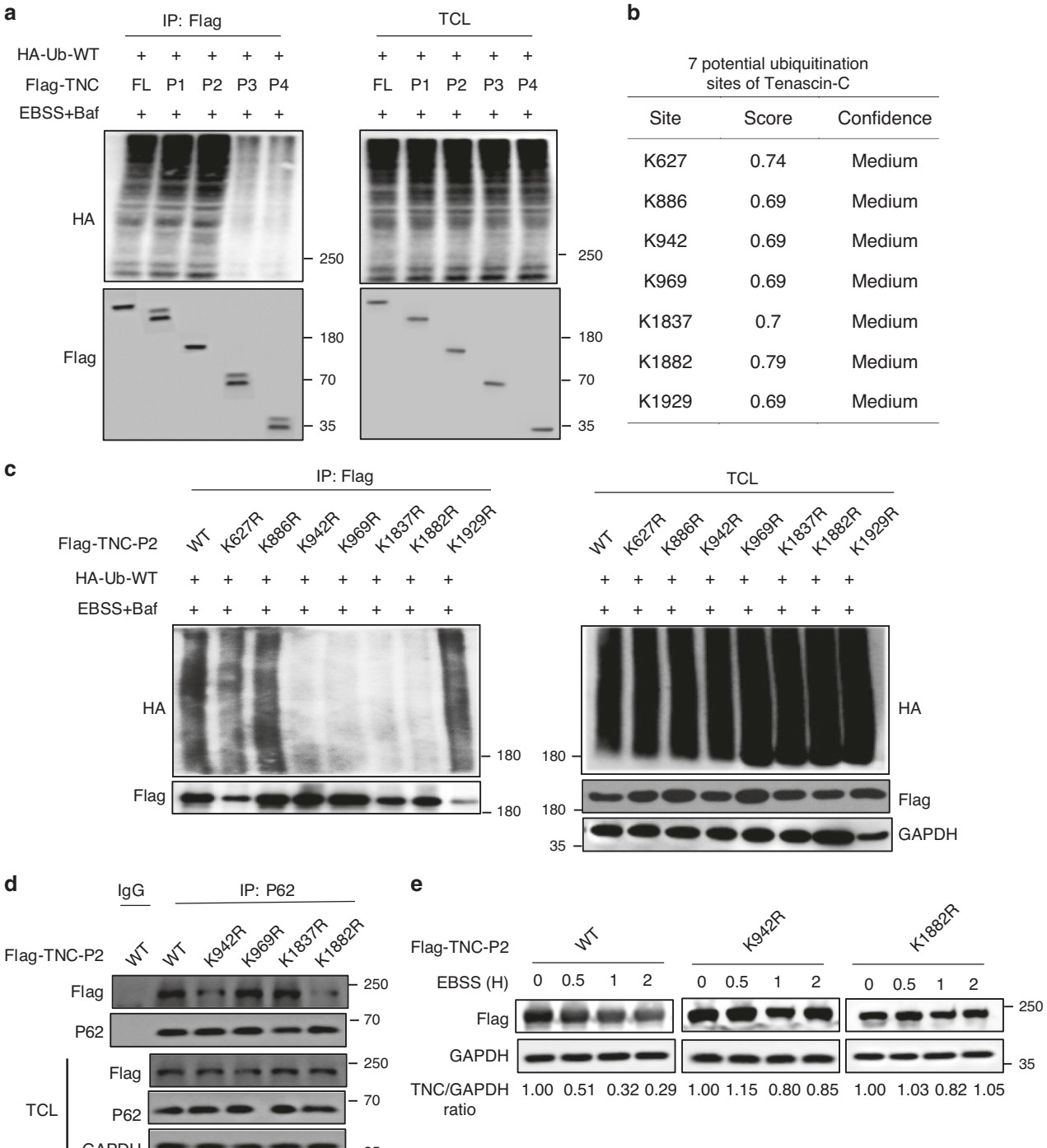

**Fig. 6 Identification of the ubiquitination site of TNC. a** Full-length Flag-TNC and TNC truncation mutants were co-transfected with HA-Ub-WT into HEK293T cells, then followed by EBSS treatment in the presence of BafA1 for 2 h. Immunoprecipitation analysis of exogenous TNC ubiquitination with the indicated antibodies. **b** Predict the ubiquitination lysine sites of TNC using Ubpred website (www.ubpred.org). **c** HEK293T cells transfected with Flag-TNC-P2 and indicated TNC mutants along with HA-Ub, then followed by EBSS treatment in the presence of BafA1 for 2 h. Immunoprecipitation analysis of exogenous TNC ubiquitination. **d** Coimmunoprecipitation and immunoassay of extracts of HEK293T cells transfected with the indicated plasmids. P62 proteins were immunoprecipitated and the bound exogenous TNC proteins examined by immunoblot. **e** HEK293T cells were transfected with the indicated plasmids, then treated with EBSS at indicated intervals and analyzed by immunoblot. All data are representative of three independent experiments.

In summary, our studies confirm that defective autophagy is a key immunosuppressive factor in TNBC tumours. Our findings also identify the crosstalk between autophagy and immune compartments as targetable component for potential therapeutic strategies for TNBC patients with poor ICB therapy response.

## Methods

**Cell culture and compounds**. Human MDA-MB-231, MDA-MB-436, HS578T, MDA-MB-468, mouse MEFs, mouse melanoma B16F10-OVA cells (kindly provided by Professor Penghui Zhou from Sun Yat-sen University Cancer Center) were maintained in Dulbecco's modified Eagle's medium supplemented with 10% foetal bovine serum (FBS, Gibico) at 37 °C under 5% $CO_2$. Human BT549, HCC1806, mouse mammary carcinoma 4T1 cells were maintained in RPMI 1640

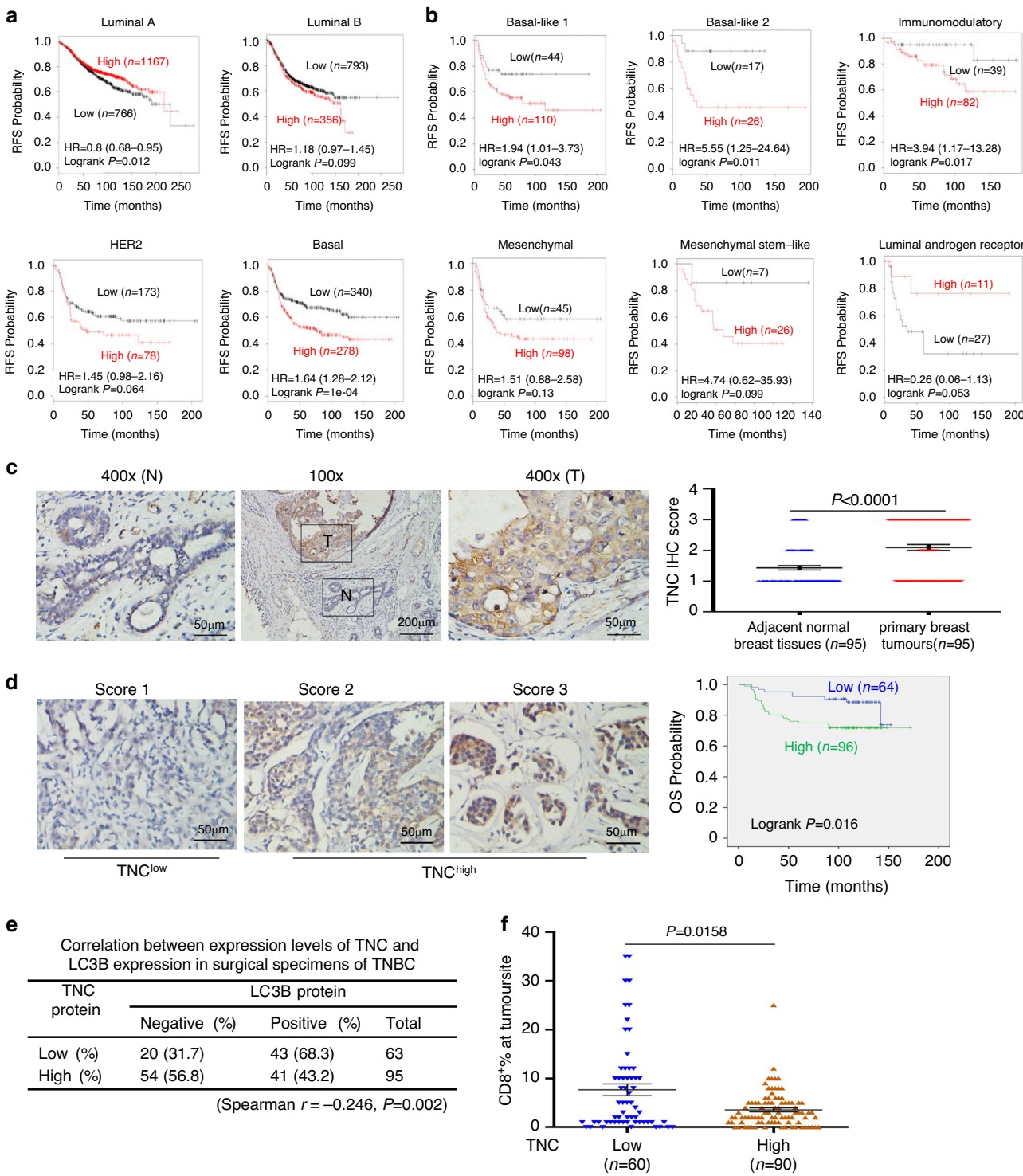

**Fig. 7 TNC is correlated with LC3B and CD8+ T cells in clinical TNBC samples. a, b** Kaplan–Meier analysis of RFS based on TNC (201645_at) mRNA levels using the KM-plotter breast cancer database (http://kmplot.com/analysis). Auto select best cutoff was chosen in the analysis. All patients were stratified according intrinsic subtype as indicated (**a**). Basal patients were stratified according the known six Pietenpol subtypes (**b**). **c** The representative images of strong TNC staining in tumour cells (T) and weak staining in the matched adjacent normal cells (N) (Left). Quantitative IHC analysis of TNC staining of primary breast tumours and adjacent normal breast tissues was shown (n = 95). **d** The representative images for each IHC score of TNC staining in 160 primary TNBC tumours were shown. Score 1 represented low TNC expression, Scores 2 and 3 represented high TNC expression (left). Kaplan–Meier analysis of OS for patients. All patients were stratified by expression of TNC (right). **e** The correlation of LC3B with TNC expression status in 158 primary TNBC tumours. Spearman's correlation coefficient r and P values were given at the bottom. **f** The correlation of CD8+ T-cell infiltration with TNC expression status in 150 primary TNBC tumours. Error bars represent mean ± SEM. The P value in **c, f** was determined by Wilcoxon matched-pairs signed rank test (two-sided). Survival curves in **a, b, d** were plotted by the Kaplan–Meier method and assessed using the log-rank test, and univariate Cox proportional hazards regression was carried out to identify HR and 95% CI.

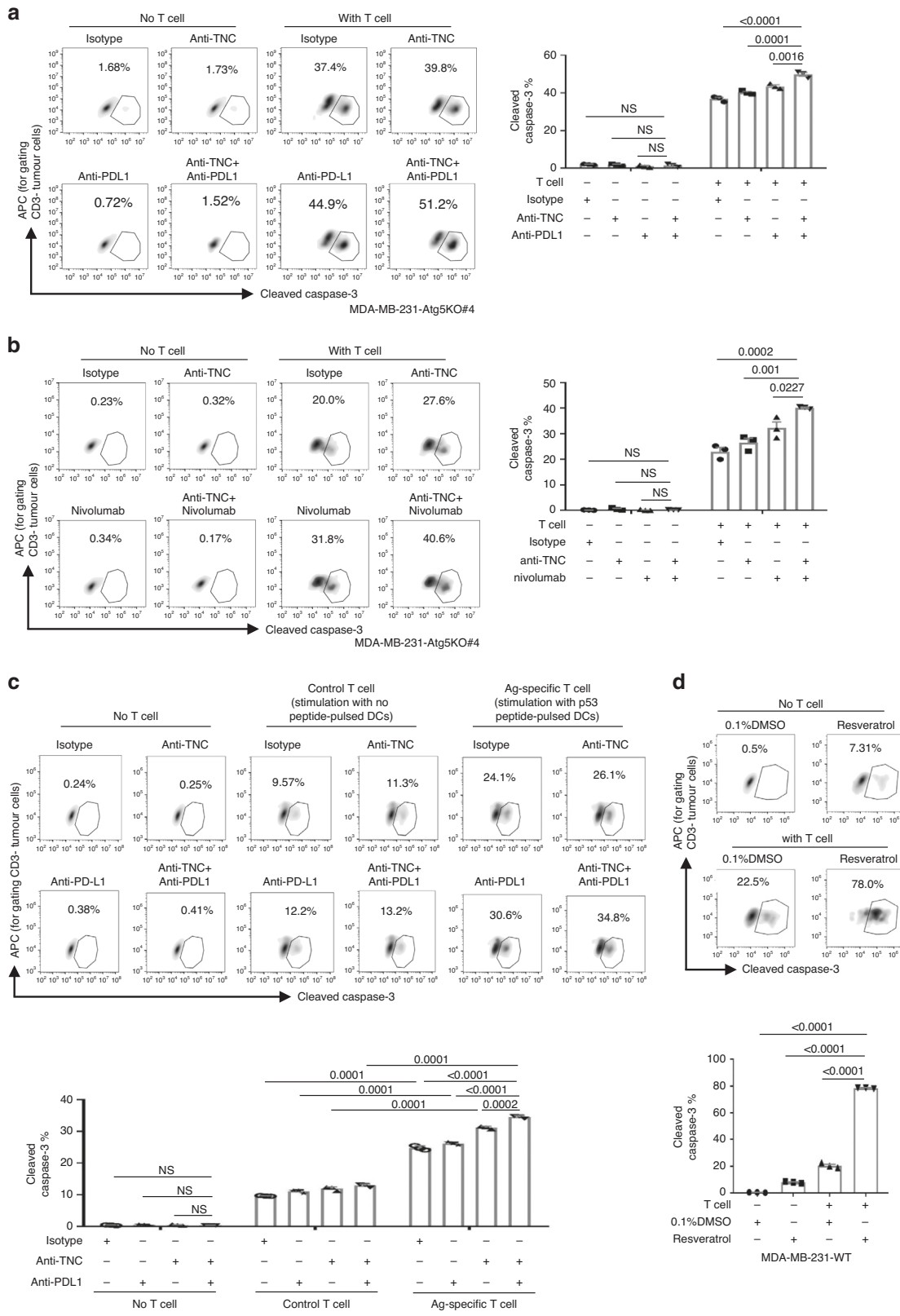

medium supplemented with 10% FBS at 37 °C under 5% $CO_2$. All the cells were authenticated using short-tandem repeat profiling, and tested negative for mycoplasma contamination. Hypoxic conditions were achieved with a hypoxic chamber flushed with a pre-analyzed gas mixture of 1% $O_2$, 5% $CO_2$, and 94% $N_2$. EBSS (#14155-063) was obtained from Thermo Fisher Scientific. Compounds MG132 (S2619), bafilomycin A1 (S1413) and rapamycin (S1039) were obtained from Selleck Chemical. CHX (BS168A, Biosharp), BMS 303141 (#4609, Tocris

Bioscience), Resveratrol (R5010, Sigma), and Recombinant Murine IFN-γ (Cat#315-05, Peprotech) were obtained commercially.

**Plasmids**. Human full-length p62, NDP52, NBR1, TAX1BP1, Tollip, Optn and BNIP3L cDNA (with fused C-terminal FLAG tag) was subcloned into pcDNA3.1 vector (Invitrogen). Human full-length Atg5 cDNA and Atg7 cDNA were

**Fig. 8 Blockade of TNC sensitizes checkpoint blockade immunotherapy in vitro. a** MDA-MB-231-Atg5KO#4 cells were pre-treated with anti-TNC (10 μg per ml) or anti-PD-L1 (10 μg/ml) for 2 h, then co-cultured with CD3/CD28-activated human T cells. Left, representative dot plots of the cleavage of caspase-3 in tumour cells measured by flow cytometry. Right, percentage of cleaved caspase-3$^+$ tumour cells. **b** MDA-MB-231-Atg5KO#4 cells were pre-treated with anti-TNC (10 μg/ml) for 2 h, then co-cultured with CD3/CD28-activated human T cells in the presence of nivolumab (10 μg/ml). Left, representative dot plots of the cleavage of caspase-3 in tumour cells measured by flow cytometry. Right, percentage of cleaved caspase-3$^+$ tumour cells. **c** MDA-MB-231-Atg7KO#5 cells were pre-treated with anti-TNC (10 μg/ml) or anti-PD-L1 (10 μg/ml) for 2 h, then co-cultured with P53 antigen-specific activated human T cells. Upper, representative dot plots of the cleaved caspase-3 in tumour cells measured by flow cytometry. Bottom, percentage of cleaved caspase-3$^+$ tumour cells. **d** MDA-MB-231-WT cells were pre-treated with 50 μM Resveratrol for 24 h, then co-cultured with CD3/CD28-activated human T cells. Upper, representative dot plots of the cleavage of caspase-3 in tumour cells measured by flow cytometry. Bottom, percentage of cleaved caspase-3$^+$ tumour cells. Error bars represent mean ± SEM, $n = 3$ biological independent samples. The $P$ value was determined by one-way ANOVA with the Dunnett's multiple comparisons test, no adjustments were made for multiple comparisons. NS no significance. All data are representative of three independent experiments.

subcloned into PCDH-CMV-MCS-EF1 vector (System Biosciences). Human full-length TNC cDNA (with fused C-terminal FLAG tag or HA tag) was subcloned into pcDNA3.1 vector. Using the human pcDNA3.1-Flag-TNC vector as a template, several deletion mutants including Flag-TNC P1, Flag-TNC P2, Flag-TNC P3, Flag-TNC P4 were developed. Site-directed mutations in pcDNA3.1-Flag-TNC P2 and HA-Ub (addgene, 18712, deposited by Edward Yeh) were developed by performing a site-directed mutagenesis. All plasmids were generated by using the ClonExpress II One Step Cloning Kit (C112-01) and ClonExpress Multis One Step Cloning Kit (C113-01) from Vazyme, and all mutations were verified by DNA sequencing. Skp2-WT and Skp2-LRR plasmid were kindly provided by Professor Dazhi Xu (Sun Yat-sen University Cancer Center).

**CRISPR-Cas9-mediated gene disruption**. To establish autophagy-deficient cell models, Atg5 Double Nickase Plasmids (h/m), Atg7 Double Nickase Plasmids (h/m) and Beclin1 Double Nickase Plasmid (m) were purchased from Santa Cruz Co. All of these plasmids were transfection-ready purified DNA plasmids. According to the manufacturer's instruction, these CRISPR/Cas9 KO plasmids were transiently transfected into MDA-MB-231, 4T1 or B16F10-OVA cell lines using Lipofectamine 2000 (Invitrogen). After 48 h, puro positive cells were dissociated and seeded at subcloning density. Atg5-knockout, Atg7-knockout and Beclin1-knockout clones were isolated by single-cell dilution cloning from the positive polyclonal sgRNA-transduced populations. The knockout clones include MDA-MB-231-Atg5KO#2, MDA-MB-231-Atg5KO#4, MDA-MB-231-Atg7KO#5, 4T1-Atg5KO#1, 4T1-Beclin1KO#2, B16-OVA-Atg7KO#1 and B16-OVA-Atg7KO#2. All knockout clones were identified by immunoblot and sequencing. Control CRISPR/Cas9 Double Nickase Plasmid (sc-437281) from Santa Cruz Co. was used as a negative control. The detailed information of these CRISPR/Cas9 KO plasmids was provided in Supplementary Table 2.

For CRISPR-Cas9-mediated TNC (m) knockout in 4T1-Atg5KO#1 cells, the specific sgRNA sequence 5′-CCCGGAGCTCATACTGCCCT-3′ (region: chr4: 63,964,709-63,964,728) targeting mouse TNC gene was cloned to LentiCRISPR (pXPR_001) plasmid. The packaging plasmids were co-transfected with pXPR-TNC sgRNA into HEK293T cells, and viral particles were harvested at 48 h post transfection. 4T1-Atg5KO#1 cells were infected with viruses for 24 h in the presence of polybrene (8 μg/ml), and stable cells were subsequently selected by puro for 3 days. The knockout cells were identified by immunoblot and sequencing. LentiCRISPR (pXPR_001) plasmid was used as a negative control.

Representative original sequences of the Atg5, Atg7, Beclin1 and TNC locus targeted by Cas9 nickase were provided in Supplementary Fig. 12.

**Generation of stable cells using lentiviral infection**. For restoration of human Atg5 and Atg7 expression, packaging plasmids were co-transfected with PCDH-Atg5 or PCDH-Atg7 into HEK293T cells, and viral particles were harvested at 48 h post transfection. For stable shRNA-mediated human TNC knockdown, TNC shRNA lentiviral particle (sc-43186-V) and control shRNA Lentiviral Particles(sc-108080) were obtained from Santa Cruz. For generating 4T1 Tet-on-inducible stable clones, shRNA sequence 5′-CAGAUGACCU GGCCUAUAA-3′ targeting mouse TNC gene was cloned to PLKO1-Tet-on vector. Packaging plasmids were co-transfected with PLKO1-Tet-on TNC shRNA into HEK293T cells, and viral particles were harvested at 48 h post transfection. Atg5 knockout or Atg7-knockout cells were infected with viral particles for 24 h in the presence of polybrene (8 μg/ml), and stable cells were subsequently selected by puro for 3 days.

**SILAC, mass spectrometry and data Analysis**. By using a SILAC Protein Quantitation Kit (Thermo Pierce) according to the manufacturer's instructions, MDA-MB-231-Atg5KO4 cell line was labelled with "heavy isotopes" (Lysine $^{13}C_6$ $^{15}N_2$ and arginine$^{13}C_6$ $^{15}N_4$), and MDA-MB- 231-WT cell line was labelled with "light isotopes" (Lysine $^{12}C_6$ and arginine $^{12}C_6$ $^{14}N_4$); MEF-WT cell line was labelled with "heavy isotopes" (Lysine $^{13}C_6$ and arginine$^{13}C_6$$^{15}N_4$), and MEF-Atg5$^{-/-}$ cell line was labelled with "light isotopes" (Lysine $^{12}C_6$ and arginine $^{12}C_6$ $^{14}N_4$). Then the samples were sent to PTM BioLabs Inc for further study. Briefly, the harvested

"heavy" and "light" labelled cells were lysed and mixed 1:1 for the protein content. After washing twice with −20 °C acetone, the proteins pellets were dissolved in 100 mM $NH_4HCO_3$ (pH 8.0) for trypsin digestion. Finally, the sample was then fractionated into fractions by high pH reverse-phase HPLC using Agilent 300 Extend C18 column (5 μm particles, 4.6 mm ID, 250 mm length). The resulting peptides were analyzed by Q Exactive$^{TM}$ Plus hybrid quadrupole-Orbitrap mass spectrometer (Thermo Fisher Scientific) or by Orbitrap Fusion$^{TM}$ Tribrid$^{TM}$ (Thermo Fisher Scientific). The resulting MS/MS data were processed using MaxQuant with integrated Andromeda search engine. Tandem mass spectra were searched against Swissprot Human or Mouse database concatenated with reverse decoy database. False discovery rate thresholds for protein, peptide and modification site were specified at 1%. KEGG database was used to identify enriched pathways by Functional Annotation Tool of GSEA against the background of Homo or mouse sapiens.

**Animal treatment protocol**. Female BALB/c mice and BALB/c SCID mice were obtained from Sun Yat-sen University, Guangzhou and were 6–10 weeks old. All procedures involving mice and experimental protocols were approved by Institutional Animal Care and Use Committee (IACUC) of Sun Yat-sen University Cancer Center. All tumour cells were mixed with matrigel (1:1) injected into the mammary fat fad of mice. To evaluate the effect of autophagy on T-cell-mediated antitumour activity, tumour xenografts were established by $1 \times 10^5$ 4T1-WT, 4T1-Atg5KO#1 and 4T1-Beclin1KO#2 cells in BALB/c mice. To examine the effect of TNC on T-cell-mediated antitumour activity, tumour xenografts were established by $1 \times 10^5$ 4T1-Atg5KO#1-ctrl and 4T1-Atg5KO#1-TNC KO cells in BALB/c mice and BALB/c SCID mice. To detect the antitumour effect of PD1 antibody, tumour xenografts were established by $1 \times 10^5$ 4T1-WT and 4T1-Beclin1KO#2 cells in BALB/c mice. The animals were randomly divided into two groups. The anti-mouse PD-1 antibody (29F.1A12; BioXcell) was intraperitoneally injected on the indicated day at a dose of 100 μg/mouse. To detect the antitumour effect of PD-L1 antibody, tumour xenografts were established by $1 \times 10^5$ 4T1-WT and 4T1-Atg5KO#1 cells in BALB/c mice. The anti-mouse PD-L1 antibody (10F.9G2; BioXcell) was intraperitoneally injected on the indicated day at a dose of 100 μg per mouse. The relevant solvent and control IgG antibody (BioXcell) were administered to control animals. To validate the combined effect of targeting TNC and checkpoint inhibitors in vivo, tumour xenografts were established by $1 \times 10^5$ 4T1-Atg5KO#1 cells stably expressing Tet-on inducible TNC shRNA injected s.c. into BALB/c mice. The animals were randomly divided into four treatment groups: isotype, Dox, anti-PD1, and Dox + anti-PD1. The anti-mouse PD1 antibody was intraperitoneally injected on the indicated day at a dose of 100 μg per mouse. DOX hyclate was given by oral gavage as 2 mg/ml in 5% dextrose in drinking water, ad libitum. Five percent dextrose in drinking water was given as control. Tumour volumes and body weight of mice were observed. Volumes were calculated by the formula: $0.5 \times a \times b^2$ in millimetres, where $a$ is the length and $b$ is the width. After mice were killed, the tumour tissues were excised and weighed.

**T lymphocytes preparations**. For human T lymphocytes preparations, healthy donors were pre-screened for HLA-A2 expression by flow cytometry and only positive individuals subjected to leukapheresis collections. T lymphocytes were then isolated from peripheral blood lymphocytes by depletion of non-T lymphocytes using a Pan T Cell Isolation Kit (Cat# 130-096-535, Miltenyi Biotec). Isolated human T lymphocytes cells were maintained in T-cell culture medium (RPMI 1640, 10% FBS, 2% PSG, 1% MEM Nonessential Amino Acids, 1% Sodium Pyruvate) and Interleukin-2 (Cat# 200-02, Peprotech) in the pre-coated plate with anti-CD3(Cat#300313, BioLegend) and anti-CD28 (Cat#302913, BioLegend). For mouse T lymphocytes preparations, mouse T cells were isolated from spleen of BALB/c mice using the Dynabeads® FlowComp™ Mouse Pan T (CD90.2) Kit (Cat#11465D, Invitrogen) according to the manufacturer's protocol. For OT-I T-cell preparations, OT-1 CD8$^+$ T cells were isolated from spleen of OT-I TCR transgenic mice (kindly provided by Prof. Peng-Hui Zhou from Sun Yat-sen University Cancer Center) using the MojoSort$^{TM}$ Mouse CD8 T Cell Isolation Kit (Cat#480007, Biolegend) according to the manufacturer's protocol. Isolated OT-I CD8$^+$ T cells were maintained in complete RPMI 1640 medium (10% FBS, 20 Mm

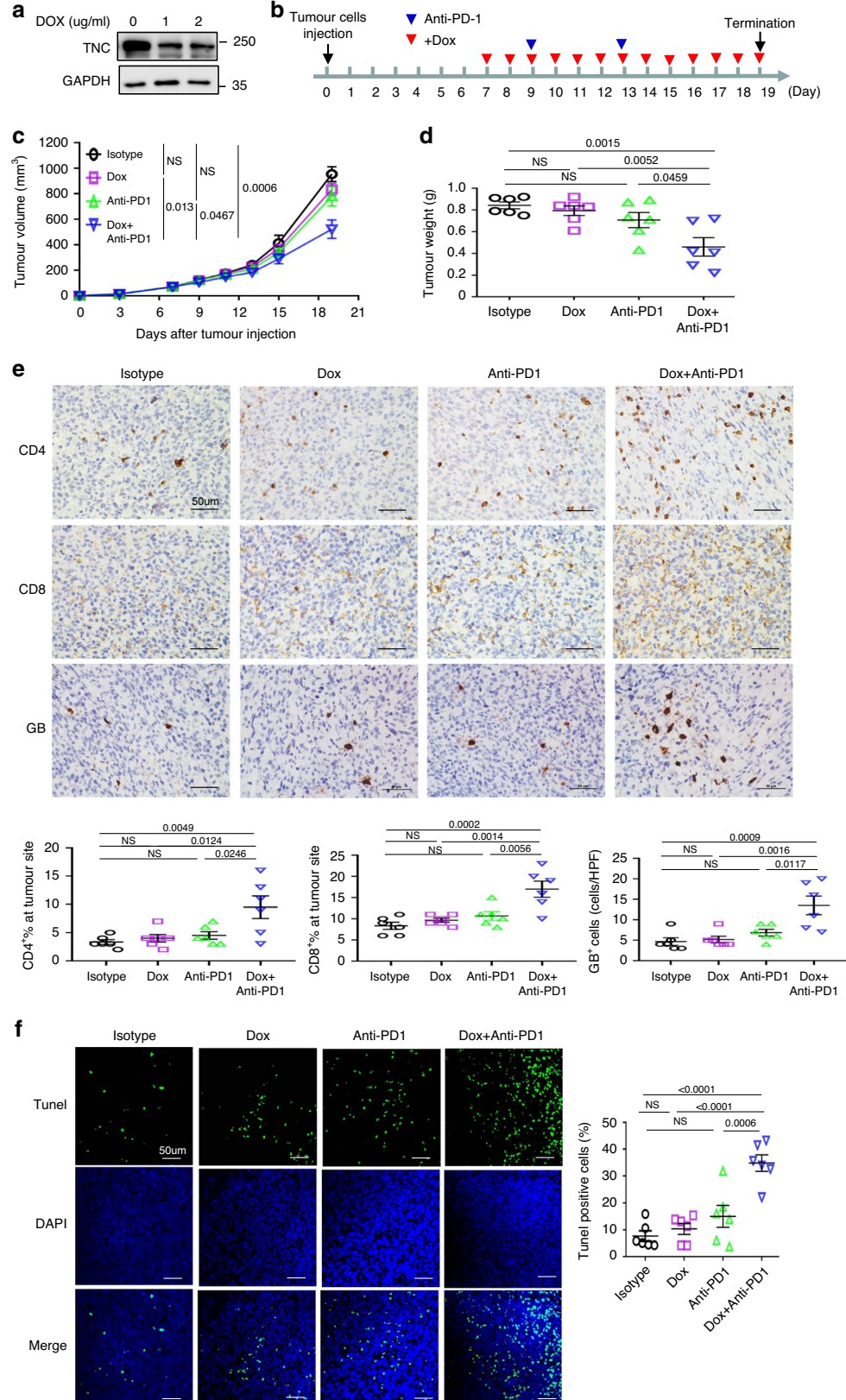

HEPES, 1 mM sodium pyruvated, 0.05 mM 2-mercaptoethanol, 2 mM L-glutamine and 50 U/ml streptomycin and penicillin).

Human P53 peptide-specific T cells were generated using autologous DCs according to a modification of the method reported previously[44,45]. In briefly, PBMCs were isolated using cell separation media (Cat# MD-YL1077-1, MD Pacific) density gradient centrifugation from HLA-A2[+] healthy donors. To generate the mature DCs, PBMCs were suspended in AIM-V medium at $3 \times 10^6$ cells/ml, then

placed in sterile flask to separate plastic adherent and non-adherent population by culture for 1 h at 37 °C. Adherent cells were cultured using AIM-V (Cat#12055091, Gibco) medium with 1000 U/ml GM-CSF (Cat#300-03, PeproTech), 500 U/ml IL-4 (Cat# 200-04, PeproTech) for 6 days to promote DCs differentiation, and the DCs were then matured for 2 days in AIM-V with 1000 U/ml GM-CSF, 500 U/ml IL-4 and 10 ng/ml TNF-α (Cat#300-01, PeproTech). Mature DCs were generally used fresh (without freezing or thawing) for the initial stimulation, but excess DCs were

**Fig. 9 TNC downregulation enhances the antitumour activity of PD1 blockade in vivo. a** 4T1-Atg5KO#1 cells were stably transfected with Tet-on inducible TNC shRNA. Then the cells were treated with DOX for 2 days. **b–d** Tumour growth of 4T1-Atg5KO#1 cells stably expressing Tet-on inducible TNC shRNA in BALB/c mice following treatment with DOX and PD1 antibody. The treatment protocol was summarized by the arrows (**b**). Tumour volumes (**c**) and tumour weights from experiment on autopsy on day 19 (**d**) were calculated. **e** Representative images of IHC staining of CD4, CD8 and granzyme B (GB) expression in xenograft tumour sections were shown for mice with treatment in (**c**) (upper). HPF, ×400 magnification. Scar bar, 50 μm. Quantitative IHC analysis of CD4, CD8 and granzyme B expression (bottom). **f** Representative images of TUNEL staining (green) and DAPI-stained nuclei (blue) in xenograft tumour sections were shown for mice with treatment in (**c**) (left). Scar bar, 50 μm. Quantification of positive TUNEL cells (right). Error bars represent mean ± SEM, $n = 6$ mice per group. The P value in **c–f** was determined by one-way analysis of ANOVA with Tukey's multiple comparisons test, no adjustments were made for multiple comparisons. NS no significance. The data are representative of two independent experiments.

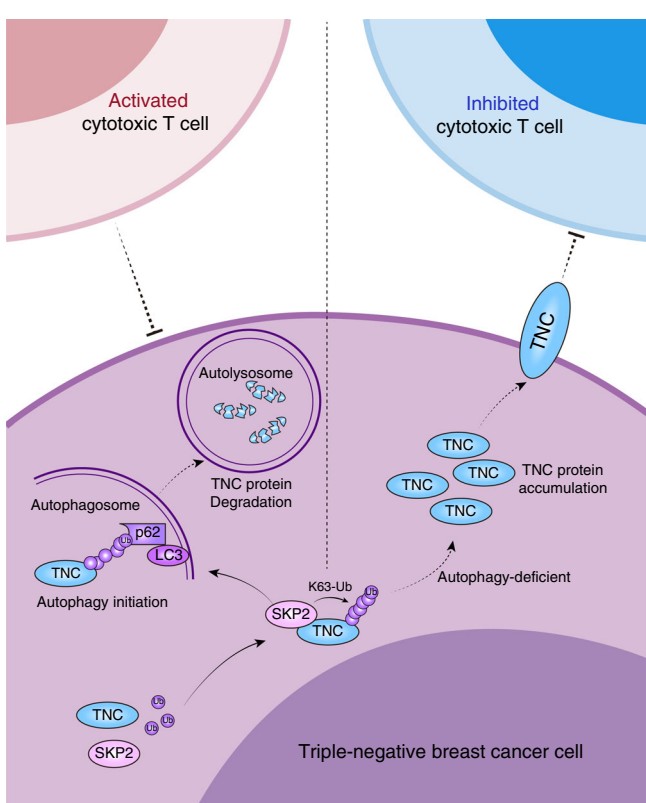

**Fig. 10 Proposed model of regulation and immunosuppression of TNC in TNBC.** Skp2 is able to catalyze the formation of the Lys63-linked polyubiquitin chains targeting TNC, which facilitats the recognition of TNC by the autophagy receptor p62, followed by further degradation by the autophagy-lysosome system. In autophagy-deficient TNBC cells, TNC is accumulated due to an abnormal autophagy-lysosome degradation pathway, resulting in TNBC cells resistance to T-cell-mediated immune attack.

frozen and frequently used in the second stimulation. To stimulate p53-specific T cells, mature DCs were pulsed with 40 μg/ml of p53$_{264-272}$ peptide (LLGRNSFEV, synthesized by Genscript Inc.) for 4 h at 37 °C/5% $CO_2$. Then, p53 loading DCs were co-cultured with autologous purified T cells at a ratio of 1:10 in X-VIVO medium (Cat#04-418Q, Lonza) containing 20 U/ml IL-2 (Cat#200-02, PeproTech), 5 ng per ml IL-7 (Cat#200-07, PeproTech) in 24 well plates. Two stimulations with P53-peptide-pulsed DC were performed 7 days apart. Half of the old medium was replaced every 2–3 days with fresh medium. Fourteen days after the first P53 loading-DCs stimulation, the frequency of P53$_{264-272}$ Tetramer⁺CD8⁺ T cells was detected by HLAA*02:01/P53$_{264-272}$ tetramer-LLGRNSFEV-PE (Cat#TS-M081-1, MBL International Corporation). HLAA*02:01/NY-ESO-1$_{157-165}$ Tetramer-SLLMWI TQC-PE (Cat#TB-M011-1, MBL International Corporation) was used as a control staining.

**T-cell killing assay**. The prepared tumour cells and activated T cells were seeded into U-shaped 96-well microtiter plates at a ratio of 1:10–1:30 in triplicates. Caspase-3 cleavage assay and lactat dehydrogenase-release (LDH) assay were performed to measuring T-cell-mediated cytotoxicity. For activated caspase-3 assay, FITC active caspase-3 apoptosis kit (Cat#550480) was purchased from BD

Biosciences. After incubation for 4–10 h, the cells were harvested, and stained with Anti-APC-CD3 antibody to excluded CD3⁻ tumour cells. After washing twice, the cell were fixed and permeabilized, then stained with FITC-conjugated activated caspase-3 antibody. For LDH assay, LDH Cytotoxicity Assay Kit (Cat#C0016) was purchased from Beyotime Biotechnology. To get the total tumour lysis value, the release reagent were added into the indicated well 1 h before harvest. After incubation for 12–24 h, supernatants were collected from each well after centrifuging. LDH release was measured in the culture medium according to the protocol suggested by the manufacture. The Absorbtion on 490 and 600 nm for each well was measured by SpectraMax Plus 384 (Molecular Devices). The cytotoxic lysis rates were calculated using the following formula: Cytotoxic lysis (%) = (OD$_{Tumour with T cells}$ − OD$_{Tumour only}$ − OD$_{T cells only}$)/(OD$_{Total Tumour Lysis}$ − OD$_{Tumour only}$) × 100%. Standardized OD value = OD$_{490nm}$-OD$_{600nm}$. For T-cell killing assay, human PD-L1 antibody (clone 29E.2A3, Cat#329709, BioLegend), mouse PD-L1 antibody (clone 10 F.9G2, Cat#BE0101, BXcell) and TNC antibody (Clone 578, Cat#MAB2138, R&D Systems) were purchased commercially. All of these antibodies have been reported to own the function of neutralizition to block a biological response[46,47].

**Flow cytometry analysis**. For cell surface staining, cell suspensions were washed twice in PBS and stained with indicated fluorescent labelled antibodies for 30 min on ice and washed with PBS. For TILs isolation, tumours were cut, minced, followed by incubation with Type IV Collagenase(Sigma) and Type I Deoxyribonuclease (Sigma) for 60 min at 37 °C with gentle shaking. After passing through a 70-μm filter, lymphocytes were purified from the interface of mouse Ficoll gradient centrifugation (MultiSciences). For intracellular staining, the cells were sorted for fixation and permeabilization using the Cytofix/CytoPerm buf kit (Cat# 554714, BD Bioscience). For detecting the proliferation of T-cell, human HLA-A2⁺ T lymphocytes were labelled with CFSE, and activated in vitro with anti-CD3 pre-coated U-96-plates in present of 100 U/ml IL2. When needed, irradiated (100 Gy) breast cancer cells were added in co-culture system at the indicated ratio in triples. All flow cytometry analysis were conducted on CytoFlex (Beckman) and the data were analyzed using FlowJo software according to the manufacturers' instructions. Compensation beads were used to evaluate spectral overlap, compensation was automatically calculated. All antibodies used for flow cytometry analysis are listed in Supplementary Table 3.

**Immunoblot and Immunoprecipitation**. For immunoblot, cells were harvested and lysed in 1× SDS sample buffer or 1× RIPA lysis buffer (Cell Signaling Technology) adding 1 mM phenylmethanesulfonyl fluoride immediately before use. A volume of 25–50 μg of total proteins was separated by SDS-PAGE transferred to PVDF membrane. Quantification of western blots was performed using ImageJ software. For immunoprecipitation, cells were collected and lysed in Pierce IP Lysis Buffer (Thermo Scientific) supplemented with Complete Protease Inhibitor Cocktail (Roche). After preclearing with protein A/G agarose (Roche) beads for 1 h at 4 °C, whole-cell lysates were used for immunoprecipitation with the indicated antibodies. Generally, 1–2 μg of commercial antibody was added to 1 mg of cell lysate, and the mixture was incubated at 4 °C overnight. After adding protein A/G agarose beads, the incubation was continued for 1 h. Antibodies used in immunoblot and immunoprecipitation are listed in Supplementary Table 4.

**SiRNA transfection**. The cells were seeded into six-well plates the day before transfection. Transfection of siRNA was performed with lipofectamine RNAimax (Invitrogen) according to the manufacturer's instruction. Oligonucleotide sequence of siRNAs are provided in Supplementary Table 5.

**Immunohistochemical (IHC) staining**. For human TNBC breast specimens analysis, 160 paraffin blocks of human TNBC breast lesions (as judged by review of the hematoxylin and eosin-stained sections) were selected for this study. These specimens were obtained following the guidelines approved by the Sun Yat-sen University Cancer. For 4T1 tumour xenografts, the tumour mass was isolated from mice and immersed with formalin and embedded into paraffin block. Sections were submerged into EDTA citrate buffer (pH 6.0 or pH 8.0), and microwaved for antigenic retrieval. Then the slides were incubated with the primary antibody at 4 °C overnight. Normal mouse/rabbit IgG as negative controls were used to ensure

specificity. Then the slides were treated by HRP polymer conjugated secondary antibody for 30 min and developed with diamino-benzidine solution (ZSGB-Bio). Nuclei were counterstained with hematoxylin. Image acquisition was performed using a Nikon camera and software. The IHC staining results were reviewed independently by two pathologists blinded to the clinicopathological information. TNC staining scores were assigned as follows: score 1 weak staining in <50% or moderate staining in <20%; score 2, weak staining in ≥50%, moderate staining in 20–50% or strong staining in <20%; score 3, moderate staining in ≥50% or strong staining ≥20%. Score 1 was considered low expression for TNC staining, Scores 2 and 3 were considered high expression for TNC staining[48]. For evaluation of LC3B staining, LC3B dot-like IHC staining was scored from 0 to 3: score 0, no dots visible or barely dots visible in <5% of the tumour cells, score 1, detectable dots in 5–25% of the tumour cells; score 2, detectable dots in 25–75% of the tumour cells; score 3, dots visible in >75% of the tumour cells. Diffuse cytoplasmic staining was assessed as absent or presenting only faint background staining (score 0), weak (score 1) or strong (score 2). Scores 1–3 were considered positive for LC3B staining[33]. The percentage of CD8 + T-cell-present area or CD4+ T-cell-present area within the tumour cell nest was recorded[36]. Staining for granzyme B was scored as the number of positive cells per HPF (400- fold magnification magnification). Antibodies used in IHC are listed in Supplementary Table 6.

**Statistical analysis**. Statistical analyses were conducted using GraphPad Prism 8.0.1. (GraphPad, La Jolla, CA, USA) and SPSS 20 software. Survival curves were plotted by the Kaplan–Meier method in SPSS and assessed using the log-rank test, and univariate Cox proportional hazards regression was carried out to identify HR (hazard ratios) and 95% CI (confidence intervals). Multivariate analysis was used to determine independent prognostic factors using a Cox proportional hazards regression model. The relationship between TNC expression and LC3B was assessed using the Spearman correlation analysis. The results are presented as the mean ± SEM was analyzed by a unpaired Student's $t$ test, or one-way ANOVA with Dunnett's multiple comparisons test or one-way ANOVA with Tukey's multiple comparisons test, or Wilcoxon matched-pairs signed rank test using GraphPad Prism. All the statistical tests were two-sided, $P < 0.05$ was considered statistically significant.

**Reporting summary**. Further information on research design is available in the Nature Research Reporting Summary linked to this article.

## Data availability

The correlation between Atg5/Beclin1 gene expression and the abundance of TILs were performed using the Tumour Immune Estimation Resource (TIMER 1.0) (https://cistrome.shinyapps.io/timer). Kaplan–Meier analysis of RFS, OS, DMFS, and PPS based on TNC mRNA levels were performed using the KM-plotter breast cancer database (http://kmplot.com/analysis). Twenty-six advanced melanoma patients data were retrieved from Gene Expression Omnibus (GSE78220). The mass spectrometry proteomics data of the indicated MDA-MB-231 cell lines have been deposited in the ProteomeXchange Consortium via the PRIDE[49] partner repository with the dataset identifier PXD019946; The mass spectrometry proteomics data of the indicated MEF cell lines have been deposited in the ProteomeXchange Consortium via the PRIDE[49] partner repository with the dataset identifier PXD019947. All the other data that support the findings of this study are available from the corresponding author upon reasonable request. The source data underlying Figs. 2a, d, f, h, k, 3b, d, e, 4a–d, f, h, 5a–f, 6a, c–e, 9a and Supplementary Figs. 2a–f, i, j, l, n, p, 3a–b, 4d, e, h, i, j, 5b–g, 6a, b, and 11a are provided as a Source Data file. Source data are provided with this paper.

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

## Acknowledgements

This study was supported by the Natural Science Foundation of China (81772835, 81772624, 81630079, 81972481, 81972855, 81803006, 81802789), the Science and Technology Project of Guangzhou (201707010086, 201803010007), the National Key R&D Program of China (2017YFC0908501), the Natural Science Foundation of Guangdong Province (2017A030313481, 2019A1515011209), and the Science and Technology Project of Guangdong Province (2017A020215032).

## Author contributions

R.D., Z.L.L., H.L.Z., Y.H., J.H.H., Y.H.C., J.M., Y.Y. L.H.Z. and D.Y. performed the experiments and analyzed the data. X.D.P., and G.K.F. provided experimental materials. N.N.Z. and X.L. performed bioinformatics analyses. P.S., J.T. and Y.W. prepared the clinical and pathological data of TNBC patients. R.D., X.F.Z., J.T. and Z.L.L. wrote the manuscript. R.D., X.F.Z. and J.T. designed and supervised this project.

## Competing interests

The authors declare no competing interests.
