## [Peer Review File · Nature Communications]

Reviewers' comments:

Reviewer #1 (Remarks to the Author): Expert in autophagy and immunology

Li et al in this paper entitled "Autophagy Deficiency Promotes Triple-Negative Breast Cancer Resistance to T Cell-Mediated Cytotoxicity by Blocking Tenascin-C Degradation" report a potentially new and interesting mechanism of tumor immune tolerance in the triple negative breast cancer. They found that autophagy down regulated Tenascin-C, which directly inhibited cytotoxicity of T cells in vitro. They further demonstrated that such the autophagy/Tenascin-C axis was important for restricting tumor growth in vivo and presented human association data suggesting clinical significance. The finding is potentially interesting. However, some of the data are not convincing and quality are poor.

Specific critiques:

Fig1

Based on data in panel a, the author drew the conclusion that "indicating that autophagy might initiate the tumor immune response, especially in TNBC". Such conclusion cannot be drawn simply based on association data. Also the mRNA levels of Atg5 and Atg6 genes are not associated with autophagy activities. The conclusion needs to be toned down.

Panel j, the error bars and statistical analysis were missing. Also, the efficiency of silencing for each gene was not shown. For panels k and i, the LDH assay cannot be used to distinguish T cell death and tumor death in the co-culture system. T cell death can be greatly increased in the co-culture when compared to T cell culture alone. Also for k, it is not clear whether they have successfully obtained the antigen specific T cells and purity of such T cells was not shown. Tumor cells that were not loaded with peptides should have been used as negative controls.

Fig2.

The original flow cytometric plots need to be shown for panel c.

Panel I: granzyme B is known to be highly expressed by Treg as well. The authors need to clarify the cell sources of granzyme B.

Panel j. The merged picture is not consistent with TUNEL staining alone.

Fig3

panel c left part, TNC was not labeled. Panel d, to be consistent, cytolytic activities should be demonstrated using 4T1 cells. Panel e. why is there a band of TNC in TNC KO lane?

H flow cytometric plots need to be shown. Also the level of Treg and granzyme B expression in Treg should be shown.

Fig7. Author should clarify how the specificity of IHC staining was determined. Panel d, the data are confusing. Both LC3 and TNC protein should accumulate when autophagy is blocked. The authors should address why they saw a reverse correlation with the two proteins.

Fig8. Antigen specific cytotoxic assays should have been used. It is not clear wither the non-specific cytotoxic assays employed here were representative of what happened in vivo. The sources of PD-L1 and TNC mAbs should be provided and whether these antibodies are functional should be demonstrated. The expression level of PD-L1 on these tumor cell lines should be determined by flow cytometry.

Reviewer #2 (Remarks to the Author):Expert in autophagy and immunology

In this manuscript, the authors examined the molecular basis of TNBC resistance to anti-tumor T cell response. They showed that that autophagy promoted the tumor immune response, and that

autophagy defects inhibited T cell-mediated killing in TNBC tumors. In addition they identified Tenascin-C as a strong candidate for autophagy-deficiency-mediated immunosuppression. The authors have analyzed the consequence of tenascin-C targeting in TNBC and showed that such targeting sensitized T cell-mediated tumor killing and improved the antitumor effects of immune checkpoint inhibitors in autophagy-deficient TNBC tumors. It is suggested that Tenascin-C could be a candidate therapeutic agent for developing combination immunotherapy for TNBC.

This is a potentially interesting study. The various experiments are well done and the authors have drawn several conclusions.

-Reports in the literature indicate that inhibition of autophagy resulted in the regulation of NK cell infiltration (Mgurditchian T et al). Targeting autophagy inhibits melanoma growth by enhancing NK cells infiltration in a CCL5-dependent manner. Proc Natl Acad Sci U S A. 2017 Did the authors check this point? Why the authors focused only on CTLs ?

-The ability of the Tenascin C to bind and activate EGFR, activating a variety of relevant cell signaling pathways has been reported. Do the authors know if the regulation of immunosuppression interferes with this receptor signaling.

Specific comments:

1. Fig 1J: I do not understand the figure. A better explanation is clearly needed

2. Authors show that PD1 treatment failed to control the growth of the 4T1-Beclin1 KO tumors compared to 4T1-WT. Is it similar with 4T1-ATG5 KO tumors?

3. Fig2i: I do not understand what the authors are trying to show. Does the green staining really represent the released granzyme B ? (or granzyme B contained in T cells that are more recruited to the tumor after anti-PD1 treatment)

4. Sup Fig 4C: could the authors evaluate the % inhibition of TNC expression (e.g. by densitometry)?

5. Fig 3D: since this is a very important point, authors should confirm that TNC knockdown in ATG5 or ATG7 KO cells restore their susceptibility to T cell-mediated killing using LDH assay as in fig 1 (and not only by evaluating caspase 3 activation)

6. I don't think that the data displayed in sup Fig 4g are consistent with the authors statement ("more T cells underwent growth arrest in the presence of the autophagy-incompetent MDA-MB-231 cells as measured by carboxyfluorescein succinimidyl ester [CFSE] dilution")

7. Fig 3h. Authors mentioned in the text that "the TNC knockout tumors had fewer CD4+ and CD8+ TIL populations and fewer activated cytotoxic T cells (granzyme B+) in their TILs than the WT tumors" but the data presented in the figure show the exact opposite. Moreover, this is not an intracellular cytokine staining as mentioned in the legend.

8. Does the slight increase in TNC mRNA expression in ATG5 KO statistically significant? (a p-value is needed)

9. Fig4e: Do the authors tried to transfect their cells with a fluorescence-tagged LC3 and performed a staining with TNC to show that TNC is incorporated in autophagosomes?

10. In fig5g, I found that the decrease of cytoplasmic TNC-p62+ punctate in response to EBSS treatment and after Skp2 knockdown is difficult to visualize and to evaluate. A better quantification is needed

11. The discussion has to be improved , more concise and relevant.

Reviewer #3 (Remarks to the Author): Expert in TNBC

This manuscript demonstrated that defects in autophagy inhibited T-cell mediated cell death in TNBC through TNC after ubiquitination by Skp2. This may have implications for potential enhancement of immunotherapy in TNBC.

Major concerns:

1) Key controls for Crispr knockouts were absent. Multiple clones, multiple cell lines or add backs should be completed to ensure that off target effects are not misleading the authors.

2) Given the heterogeneity and subtypes of TNBC, the authors should test whether the known subtypes of TNBC have differences in RFS based on TNC levels. Indeed, this is likely the single largest flaw with the manuscript - TNBC encompasses a wide variety of tumors and this is not a mechanism that will be broadly applicable to the entire class of tumors. Finding the subset of patients where this mechanism is valid would substantially benefit the manuscript.

Minor:

Figure 9E is not appreciated by color blind reviewers - no staining is observed.

The level of detail in the methods is appropriate as were statistical tests.

We would like to take this opportunity to thank you for your thoughtful critiques and constructive comments that helped us to improve our manuscript. Based on your kind advices, we have extensively revised the original manuscript with some new experiments. Also, the text of manuscript has been checked for formatting. We have corrected the mistakes and some information has been added in the corresponding text. Response point to point as listed below.

Point-by-point response

Reviewer #1 (Remarks to the Author): Expert in autophagy and immunology

Li et al in this paper entitled “Autophagy Deficiency Promotes Triple-Negative Breast Cancer Resistance to T Cell-Mediated Cytotoxicity by Blocking Tenascin-C Degradation” report a potentially new and interesting mechanism of tumor immune tolerance in the triple negative breast cancer. They found that autophagy down regulated Tenasin-C, which directly inhibited cytotoxicity of T cells in vitro. They further demonstrated that such the autophagy/Tenasin-C axis was important for restricting tumor growth in vivo and presented human association data suggesting clinical significance. The finding is potentially interesting. However, some of the data are not convincing and quality are poor.

Specific critiques:

Fig1

Based on data in panel a, the author drew the conclusion that “indicating that autophagy might initiate the tumor immune response, especially in TNBC”. Such conclusion cannot be drawn simply based on association data. Also the mRNA levels of Atg5 and Atg6 genes are not associated with autophagy activities. The conclusion needs to be toned down.

Response: In the revised manuscript, we have modified the conclusion as “indicating the expression of these two autophagy-related genes might be involved in the tumor immune response, especially in TNBC” (Please refer to highlighted sentence at **Page 5** in the revised manuscript).

Panel j, the error bars and statistical analysis were missing. Also, the efficiency of silencing for each gene was not shown.

Response: To better understand this figure, we have repeated this experiment again, and presented the new data in another way including the error bars and statistical analysis (**Fig.1f and supplementary Fig.2m** in the revised manuscript). Also the efficiency of silencing for each gene was shown in **supplementary Fig.2l** of the revised manuscript. We got the similar result as before. The results showed that the silencing of the expression of almost all autophagy-related genes decreased the susceptibility of the tumor cells to apoptosis induced by T-lymphocytes

For panels k and i, the LDH assay cannot be used to distinguish T cell death and tumor death in the co-culture system. T cell death can be greatly increased in the co-culture when compared to T cell culture alone. Also for k, it is not clear whether they have successfully obtained the antigen specific T cells and purity of such T cells was not shown. Tumor cells that were not loaded with peptides should have been used as negative controls.

Response: The reviewer has raised a good point. It is true that LDH assay is not the best method to evaluate T-cell-mediated cytotoxicity. Because T cell death can be greatly increased in the co-culture when compared to T cell culture alone, and it cannot be used to distinguish T cell death and tumor death in the co-culture system. Therefore, we deleted this result of LDH, and have done a new experiment to evaluate the P53 antigen specific T cell-mediated cytotoxicity using flow cytometry-based CTL assay by detection of the cleavage of caspase 3 in tumor cells, and presented the new data in **Fig.1g and supplementary Fig.2n** of the revised manuscript. Peptide 264–272 from naturally processed p53 has proven to be a potential T epitope because of its strong affinity to HLA-A2, and MDA-MB-231 cells display high p53

concentrations in the nucleus due to a p53 gene mutation in codon 280¹. In the experiment, P53-antigen loading-DCs were cocultured with autologous nonadherent T-lymphocytes from the same HLA-A2⁺ healthy donor. Also, MDA-MB-231 cells that were not loaded with P53 peptides have been used as negative controls. The result showed that knockout of Atg5 reduced the P53 antigen specific T cell-induced tumor killing, restoration of Atg5 recovered the autophagy-incompetent cells' susceptibility to killing by cytotoxic T cells (**Fig.1g** , **supplementary Fig.2n**, Please refer to highlighted sentence at **Page 6** in the revised manuscript).

To better elucidate the specific killing effect of T cells against the autophagy-deficient tumor cells, we also have done a new experiment to evaluate an antigen-specific T cell-mediated killing against B16F10-OVA cells, and presented the new data in **Fig.1h** and **supplementary Fig.2o** of the revised manuscript. We depleted Atg7 in the ovalbumin (OVA) positive B16F10 cells (**Supplementary Fig. 2n** in the revised manuscript), then the cells were co-cultured with activated CD⁺8 T cells isolated from OT-I TCR transgenic mice, which provided an antigen-specific T cell-mediated killing to B16F10-OVA cells. The result also showed that compared to their autophagy-competent counterparts, the autophagy-deficient B16F10-OVA Atg7 KO cells were more resistant to the tumor antigen-specific T cell-mediated killing (**Fig.1h**, Please refer to highlighted sentence at **Page 6** in the revised manuscript).

Fig2.

The original flow cytometric plots need to be shown for panel c.

Response: To further confirm the in vivo effect of autophagy on T cell-mediated antitumor activity, we have repeated this experiment again, and presented the new data in **Fig. 2a-2b** of the revised manuscript. The representative original flow cytometric plots have been shown in **Fig. 2b**. We got the similar result as before. The result showed that the the autophagy-deficient 4T1-Atg5 KO tumors grew faster than the autophagy- competent 4T1 control cells in the immunocompetent BALB/c mice (**Fig. 2a** in the revised manuscript). And 4T1-Atg5 KO tumors had not only a decreased total CD4⁺ and CD8⁺ TIL population but also fewer activated cytotoxic

CD4 T and CD8 T cells in their TILs ($\text{IFN}\gamma^+\text{CD4}^+$ and $\text{IFN}\gamma^+\text{CD8}^+$) than those in the autophagy-competent 4T1 control cells (**Fig. 2b** in the revised manuscript).

Panel I: granzyme B is known to be highly expressed by Treg as well. The authors need to clarify the cell sources of granzyme B.

Response: Actually, granzyme B is most commonly found in the granules of natural killer cells (NK cells) and cytotoxic T cells. Also, granzyme B is highly expressed in 5-30% of $\text{CD4}^+\text{Foxp3}^+$ Tregs in tumors. To clarify the cell sources of granzyme B, we originally want to perform the immunofluorescence staining of both CD8 and granzyme B in the same frozen tissue slices and quantify the proportion of co-location of CD8 and granzyme B. But we we performed this animal experiment one year ago and the frozen tissue slices were kept in the -80 degree refrigerator for too long. We found the ice crystals in the frozen tissue slices were particularly serious which had pierced cells and destroyed tissue morphology. Therefore, we deleted this immunofluorescence picture of granzyme B. Additionally, we also have done a new experiment to compare the antitumor effect of PD-L1 antibody between 4T1-WT and 4T1-Atg5 KO tumors, and presented the new data in **Fig. 2g-2k, Supplementary Fig. 3d** of the revised manuscript. We got the similar result that PD-L1 treatment failed to control the growth of the 4T1- Atg5 KO tumors compared to 4T1-WT in normal, immunocompetent mice (**Fig. 2g-2i** in the revised manuscript). Actually, the PD-L1 treatment stimulated more activated cytotoxic CD4 T and CD8 T cells in their TILs ($\text{IFN}\gamma^+\text{CD4}^+$ and $\text{IFN}\gamma^+\text{CD8}^+$) and resulted in an increased number of TUNEL positive cells in the 4T1-WT tumors (**Fig. 2j, 2k, Supplementary Fig. 3d**, Please refer to highlighted sentence at **Page 10** in the revised manuscript).

Panel j. The merged picture is not consistent with TUNEL staining alone.

Response: We have corrected the mistake. The merged picture is now consistent with TUNEL staining alone (**Fig. 2f** in the revised manuscript).

Fig3

panel c left part, TNC was not labeled.

Response: TNC has been labeled in **Fig. 3c** of the revised manuscript. In addition, we have detected the expression of TNC in autophagy-deficient cell models with multiple cell lines and added backs, and presented the new data in **Fig. 3c** of the revised manuscript. we confirmed that the protein levels of TNC were substantially elevated in the autophagy-deficient MDA-MB-231 and MEF cells, but blunted by restoration of autophagy (**Fig. 3c**, Please refer to highlighted sentence at **Page 13** in the revised manuscript).

Panel d, to be consistent, cytolytic activities should be demonstrated using 4T1 cells.

Response: To elucidated whether TNC protected autophagy-incompetent TNBC cells from immune surveillance, the MDA-MB-231 Atg5 KO and MDA-MB-231 Atg7 KO cells were infected with lentiviral vectors encoding either a scrambled short hairpin RNA or a TNC-specific shRNA. The result showed that the knockdown of TNC in MDA-MB-231 Atg5 KO and Atg7 KO cells restored the autophagy-incompetent cells' susceptibility to killing by cytotoxic T cells using Caspase-3 cleavage assay and LDH assay (**Fig. 3d, Supplementary Fig.4e**, Please refer to highlighted sentence at **Page 13** in the revised manuscript). We also examined the tumorigenesis of 4T1-Atg5 KO tumor cells after depleting TNC using specific sgRNA in vivo (**Fig. 3e** in the revised manuscript). The result showed that the TNC knockout significantly reduced the tumor burden as confirmed by the growth curve of the xenograft tumor volume and the tumor weight in syngeneic BALB/c mice (**Fig. 3f, 3g** in the revised manuscript). However, we did not observe any significant changes in the tumor growth in the severe combined immunodeficient (SCID) mice (**Supplementary Fig. 4g, 4h** in the revised manuscript), suggesting that the differential tumorigenicity was attributed to immune surveillance. Indeed, the TNC knockout tumors had more CD4⁺ and CD8⁺ TIL populations and more activated cytotoxic T cells (IFN γ ⁺CD4⁺ and IFN γ ⁺CD8⁺) in their TILs than the control tumors (**Fig. 3h, Supplementary Fig. 4i**, Please refer to highlighted sentence at **Page 14** in the revised manuscript).

Panel e. why is there a band of TNC in TNC KO lane?

Response: For CRISPR-Cas9-mediated TNC (m) knockout in 4T1-Atg5 KO cells, the specific sgRNA sequence AGGGCAGTATGAGCTCCGGG targeting mouse TNC gene was cloned to LentiCRISPR (pXPR_001) plasmid. The packaging plasmids were co-transfected with pXPR-TNC sgRNA into HEK293T cells, and viral particles were harvested at 48 hr post-transfection. 4T1-Atg5 KO cells were infected with viruses for 24 hr in the presence of polybrene. The positive cells were selected under puromycin for 3 days. We did not select monoclones from the positive cells because viral infection was very efficient. Although there was a very weak band of TNC in TNC KO lane, the knockout efficiency of TNC maybe more than 90% (**Fig. 3e** in the revised manuscript).

Panel h flow cytometric plots need to be shown. Also the level of Treg and granzyme B expression in Treg should be shown.

Response : To further confirm the role of TNC in immune surveillance in autophagy-deficient 4T1-Atg5 KO tumors, we have repeated this experiment again, and presented the new data in **Fig. 3e-3h** of the revised manuscript. In the new experiment, we evaluated the function of activated cytotoxic T cells by detecting IFN γ . The representative original flow cytometric plots have been shown in **Supplementary Fig. 4i** of the revised manuscript. We got the similar result as before. The result showed that the TNC knockout significantly reduced the tumor burden as confirmed by the growth curve of the xenograft tumor volume and the tumor weight in syngeneic BALB/c mice (**Fig. 3f, 3g** in the revised manuscript). Indeed, the TNC knockout tumors had more CD4⁺ and CD8⁺ TIL populations and more activated cytotoxic T cells (IFN γ ⁺CD4⁺ and IFN γ ⁺CD8⁺) in their TILs than the control tumors (**Fig. 3h, Supplementary Fig. 4i**, Please refer to highlighted sentence at **Page 14** in the revised manuscript).

Fig7.

Author should clarify how the specificity of IHC staining was determined.

Response: For human TNBC breast specimens, primary antibodies for anti-TNC

(GTX12298, Genetex), anti-LC3B (NB100-2220, Novus), anti-CD8 (ab17147, Abcam) were used. The datasheets of the three antibodies have noted that these antibodies can be applied for IHC staining, and provided the sample figure of IHC staining. Here, we provided normal mouse/rabbit IgG as negative controls of IHC staining to ensure specificity (Please refer to highlighted sentence at **Page 44**). The result showed that there was no any positive brown cells when the TNBC breast tissue were incubated with normal mouse/rabbit IgG.

Figure. A validation for the specificity of the antibodies in human TNBC breast specimens.

Panel d, the data are confusing. Both LC3 and TNC protein should accumulate when autophagy is blocked. The authors should address why they saw a reverse correlation with the two proteins.

Response: The previous result in our study showed that TNC was degraded by the selective autophagy-lysosome pathway. To further validate our findings in human cancer patient samples, we analyzed the correlation between TNC and LC3B in 159 TNBC primary breast cancer tissues by IHC. We observed a clearly negative correlation between the expression levels of TNC and LC3B in these specimens. The frequency of cases with a low expression of TNC among the patients with the positive expression of LC3B (44/64 cases, 68.8%) was significantly larger than that among those cases with a negative expression of LC3B (20/64, 31.2%) (**Fig.7e** in the revised manuscript). LC3B is a well-established marker for autophagy, and the evaluation of LC3B staining was determined by previous report². To a large extent, the expression of LC3B dot-like immunohistochemical staining represents the autophagic capability of cells. If it's positive for LC3B staining, the autophagic capability of these cells was relatively high. As a result, TNC was degraded by autophagic lysosome pathway and the expression of TNC IHC staining was low. If it's negative for LC3B staining, the autophagic capability of these cells was relatively low. As a result, the degradation of TNC via autophagic lysosome pathway was blocked and the expression of TNC IHC staining was high. A previous study also showed a negative correlation between HK2 and MAP1LC3B in a tissue microarray (TMA) of liver cancer samples, for HK2 was degraded by the selective autophagy-lysosome pathway³.

Fig8.

Antigen specific cytotoxic assays should have been used. It is not clear wither the non-specific cytotoxic assays employed here were representative of what happened in vivo.

Response: The reviewer has raised a good point. We have done this experiment, and presented the new data in **Fig. 8c** and **Supplementary Fig. 9b** of the revised manuscript. We used antigen specific T cell-mediated cytotoxicity assays to measure the synergistic effect. The result showed that the blockage of TNC resensitized anti-PD-L1-treated MDA-MB-231 Atg7 KO cells to P53 antigen-specific T cell-induced tumor killing (**Fig. 8c** in the revised manuscript). And the effect of

mouse OT1-CD8⁺T cell-mediated killing in B16F10-OVA Atg7 KO cells also could be enhanced when anti-TNC and anti-PD-L1 were used synergically (Please refer to highlighted sentence at **Page 26, Supplementary Fig. 9b** in the revised manuscript).

The sources of PD-L1 and TNC mAbs should be provided and whether these antibodies are functional should be demonstrated.

Response: For T-cell killing assay, human PD-L1 antibody (clone 29E.2A3, Cat#329709, BioLegend), mouse PD-L1 antibody (clone 10F.9G2, Cat#BE0101, Bxcell) and Human/Mouse Tenascin C antibody (Clone 578, Cat#MAB2138, R&D Systems) were purchased commercially. All of these antibodies have been reported to own the function of neutralization to block a biological response^{4,5}. The information has been added to the revised manuscript in **Methods (T-cell killing assay**, Please refer to highlighted sentence at **Page 41** in the revised manuscript).

The expression level of PD-L1 on these tumor cell lines should be determined by flow cytometry.

Response: In the revised manuscript, we have done this experiment, and presented the new data in **Supplementary Fig. 9a**. The result showed that the autophagy-deficient tumor cells expressed PD-L1 on the membrane (**Supplementary Fig. 9a**, Please refer to highlighted sentence at **Page 26** in the revised manuscript).

Reviewer #2 (Remarks to the Author):Expert in autophagy and immunology

In this manuscript, the authors examined the molecular basis of TNBC resistance to anti-tumor T cell response. They showed that that autophagy promoted the tumor immune response, and that autophagy defects inhibited T cell-mediated killing in TNBC tumors. In addition they identified Tenascin-C as a strong candidate for autophagy-deficiency-mediated immunosuppression. The authors have analyzed the consequence of tenascin-C targeting in TNBC and showed that such targeting sensitized T cell-mediated tumor killing and improved the antitumor effects of immune checkpoint inhibitors in autophagy-deficient TNBC tumors. It is suggested that Tenascin-C could be a candidate therapeutic agent for developing combination immunotherapy for TNBC. This is a potentially interesting study. The various experiments are well done and the authors have drawn several conclusions.

-Reports in the literature indicate that inhibition of autophagy resulted in the regulation of NK cell infiltration (Mgurditchian T et al). Targeting autophagy inhibits melanoma growth by enhancing NK cells infiltration in a CCL5-dependent manner. Proc Natl Acad Sci U S A. 2017 Did the authors check this point? Why the authors focused only on CTLs ?

Response: The interactions between tumor cell autophagy and the immune system are complicated. Several studies have demonstrated the potential of autophagy to induce potent immune responses. Chemotherapy-induced autophagy causes the release of ATP from tumor cells in mice, thereby leading to the recruitment of immune cells including DC, CD4⁺ or CD8⁺ T cells, then stimulating antitumor immune responses⁶. PTEN loss in melanoma promotes resistance to T cell-mediated killing by inhibiting autophagy⁷. MAGE-A proteins are reportedly associated with the suppression of autophagy in melanoma, potentially disrupting the development of optimal anticancer immunity and driving primary resistance to CTLA-4 blockade². In contrast, autophagy has also been reported to limit immune-mediated cytotoxicity. The hypoxia-induced resistance of lung tumors to cytolytic T lymphocyte (CTL)-mediated lysis is associated with autophagy induction in target cells⁸. Targeting autophagy

inhibits melanoma growth by enhancing NK cells infiltration in a CCL5-dependent manner⁹. Thus, additional studies to understand the interactions between tumor cell autophagy and the immune system are needed¹⁰.

To investigate the potential contribution of autophagy to the tumor immune response in breast cancer patients, we first evaluated the correlation between autophagy-related gene expression and abundance of T cell immune infiltrates (TIL) using the public resource Tumor Immune Estimation Resource (TIMER)¹¹. Only in basal tumors but not Luminal and Her2 tumors, the expression levels of Beclin 1 and Atg5 were found to be positively related to the abundance of B cells, CD4+ T cells, CD8+ T cells, neutrophils, and dendritic cells ($P < 0.05$) (**Figure 1a** in the revised manuscript), indicating the expression of these two autophagy-related genes might be involved in the tumor immune response, especially in TNBC. Since the effectiveness of many immune checkpoint inhibitors in the treatment of TNBC is currently under evaluation in ongoing clinical trials, we further elucidated the role of autophagy in the T-lymphocyte immune system in TNBC. Thus, we did not check the interactions between tumor cell autophagy and NK immune cells in TNBC tumors in this study. Actually, the infiltration of functional cytotoxic immune cells, including NK and cytotoxic T lymphocytes, is a major factor in achieving successful immunotherapies⁹. We will explore the interactions between tumor cell autophagy and NK cell-mediated antitumor immune responses in TNBC in the future study.

-The ability of the Tenascin C to bind and activate EGFR, activating a variety of relevant cell signaling pathways has been reported. Do the authors know if the regulation of immunosuppression interferes with this receptor signaling.

Response: TNC monomer comprises a N-terminal tenascin assembly domain followed by the EGF-like repeat region, a region of up to 17 fibronectin type III (FNIII) repeats the fibronectin type III (FNIII)-like region and a C-terminal fibrinogen-like domain¹². It is known that TNC binds to several receptors, such as integrins, Annexin II, EGFR, Contactin, etc. Among these receptors, integrins are the

major class¹³. In fibroblasts, TNC is reported to activate EGFR signaling and subsequently activate extracellular signal-regulated, mitogen-activated protein kinase through binding of the EGF-like repeats of tenascin-C to the EGFR¹⁴. In theory, the MAP kinase signaling pathway in T lymphocytes might be activated by the EGF-like repeats of TNC through their direct activation of EGFR. However, TNC is well known for its function in overcoming immune surveillance, and is reported to negatively regulate ERK phosphorylation in T lymphocytes. Delayed ERK phosphorylation and reduced amoeba-like shape formation in Jurkat cells contacting U-118MG cells was associated with abundance tenascin-C expression¹⁵. Specific silencing of TNC in prostate cancer stem-like cells substantially increased the ERK2 phosphorylation levels in T lymphocytes¹⁶. TNC produced from stem-like brain tumor initiating cells inhibited T cell proliferation associated with reduced p-ERK and p-mTOR¹⁷. Mechanistically, TNC can engage the integrin β receptor of T cells through its fibronectin type III (FNIII) repeats, then block GTPase Rho activation to inhibit the reorganization of the actin/myosin and microtubule cytoskeleton^{16, 18, 19}. TNC also inhibits integrin $\alpha 5\beta 1$ -dependent T lymphocyte adhesion to fibronectin through the binding of its fnIII 1-5 repeats to fibronectin²⁰. Therefore, the regulation of immunosuppression of fibronectin type III (FNIII) repeats of TNC interferes with its EGF-like repeats to EGFR receptor signaling in T lymphocytes.

Specific comments:

1. Fig 1J: I do not understand the figure. A better explanation is clearly needed

Response: To better understand this figure, we have repeated this experiment again, and presented the new data in another way including the error bars and statistical analysis (**Fig.1f and supplementary Fig.2m** in the revised manuscript). Also the efficiency of silencing for each gene was shown in **supplementary Fig.2l** of the revised manuscript. We got the similar result as before. The results showed that the silencing of the expression of almost all autophagy-related genes decreased the susceptibility of the tumor cells to apoptosis induced by T-lymphocytes

2. Authors show that PD1 treatment failed to control the growth of the 4T1-Beclin1 KO tumors compared to 4T1-WT. Is it similar with 4T1-ATG5 KO tumors?

Response: The reviewer has raised a good point. We have done a new experiment to compare the antitumor effect of PD-L1 antibody between 4T1-WT and 4T1-Atg5 KO tumors, and presented the new data in **Fig. 2g-2k, Supplementary Fig. 3d** of the revised manuscript. We got the similar result that PD-L1 treatment failed to control the growth of the 4T1- Atg5 KO tumors compared to 4T1-WT in normal, immunocompetent mice (**Fig. 2g-2i** in the revised manuscript). Actually, the PDL1 treatment stimulated more activated cytotoxic CD4 T and CD8 T cells in their TILs ($\text{IFN}\gamma^+\text{CD4}^+$ and $\text{IFN}\gamma^+\text{CD8}^+$) and resulted in an increased number of TUNEL positive cells in the 4T1-WT tumors (**Fig. 2j, 2k, Supplementary Fig. 3d**, Please refer to highlighted sentence at **Page 10** in the revised manuscript).

3. Fig2i: I do not understand what the authors are trying to show. Does the green staining really represent the released granzyme B ? (or granzyme B contained in T cells that are more recruited to the tumor after anti-PD1 treatment)

Response: Actually, granzyme B is most commonly found in the granules of natural killer cells (NK cells) and cytotoxic T cells. Also, granzyme B is highly expressed in 5-30% of $\text{CD4}^+\text{Foxp3}^+$ Tregs in tumors. To clarify the cell sources of granzyme B, we originally want to perform the immunofluorescence staining of both CD8 and granzyme B in the same frozen tissue slices and quantify the proportion of co-location of CD8 and granzyme B. But we we performed this animal experiment one year ago and the frozen tissue slices were kept in the -80 degree refrigerator for too long. We found the ice crystals in the frozen tissue slices were particularly serious which had pierced cells and destroyed tissue morphology. Therefore, we deleted this immunofluorescence picture of granzyme B. Additionally, we have also done a new experiment to compare the antitumor effect of PD-L1 antibody between 4T1-WT and 4T1-Atg5 KO tumors, and presented the new data in **Fig. 2g-2k, Supplementary Fig.**

3d of the revised manuscript. We got the similar result that PDL1 treatment failed to control the growth of the 4T1- Atg5 KO tumors compared to 4T1-WT in normal, immunocompetent mice (**Fig. 2g-2i** in the revised manuscript). Actually, the PD-L1 antibody treatment stimulated more activated cytotoxic CD4⁺ T and CD8⁺ T cells in their TILs (IFN γ ⁺CD4⁺ and IFN γ ⁺ CD8⁺) and resulted in an increased number of TUNEL positive cells in the 4T1-WT tumors (**Fig. 2j, 2k, Supplementary Fig. 3d**, Please refer to highlighted sentence at **Page 10** in the revised manuscript).

4. Sup Fig 4C: could the authors evaluate the % inhibition of TNC expression (e.g. by densitometry)?

Response: To clearly show the efficiency of knockdown of TNC, we have repeated this experiment again, and presented the new data in **supplementary Fig.4c** of the revised manuscript. We also quantified the bands of TNC and GAPDH by Image J software. The result showed that the expression of TNC decreased nearly 60~70 % after TNC knockdown (**supplementary Fig.4c** in the revised manuscript).

5. Fig 3D: since this is a very important point, authors should confirm that TNC knockdown in ATG5 or ATG7 KO cells restore their susceptibility to T cell-mediated killing using LDH assay as in fig 1 (and not only by evaluating caspase 3 activation)

Response: The reviewer has raised a good point. We have done a new experiment and presented the new data in **Fig. 3d, Supplementary Fig.4e** of the revised manuscript. To elucidated whether TNC protected autophagy-incompetent TNBC cells from immune surveillance, the MDA-MB-231 Atg5 KO and MDA-MB-231 Atg7 KO cells were infected with lentiviral vectors encoding either a scrambled short hairpin RNA or a TNC-specific shRNA. The result showed that the knockdown of TNC in MDA-MB-231 Atg5 KO and Atg7 KO cells restored the autophagy-incompetent cells' susceptibility to killing by cytotoxic T cells using Caspase-3 cleavage assay (**Fig. 3d** in the revised manuscript). We also confirmed that TNC knockdown in MDA-MB-231 Atg5 KO and Atg7 KO cells restore their susceptibility to T

cell-mediated killing using LDH assay (**Supplementary Fig.4e**, Please refer to highlighted sentence at **Page 13** in the revised manuscript).

6. I don't think that the data displayed in sup Fig 4g are consistent with the authors statement ("more T cells underwent growth arrest in the presence of the autophagy-incompetent MDA-MB-231 cells as measured by carboxyfluorescein succinimidyl ester [CFSE] dilution")

Response: We have repeated this experiment again, and presented the new data in **Supplementary Fig.4f** of the revised manuscript. By using carboxyfluorescein succinimidyl ester [CFSE] dilution, the flow cytometry analysis showed that more CD4⁺T and CD8⁺T cells underwent proliferation in the presence of MDA-MB-231 Atg5 KO-TNC shRNA cells when that compared to MDA-MB-231 Atg5 KO-ctrl shRNA cells (**Supplementary Fig.4f**, Please refer to highlighted sentence at **Page 13** in the revised manuscript).

7. Fig 3h. Authors mentioned in the text that "the TNC knockout tumors had fewer CD4⁺ and CD8⁺ TIL populations and fewer activated cytotoxic T cells (granzyme B⁺) in their TILs than the WT tumors" but the data presented in the figure show the exact opposite. Moreover, this is not an intracellular cytokine staining as mentioned in the legend.

Response : To further confirm the role of TNC in immune surveillance in autophagy-deficient 4T1-Atg5 KO tumors, we have repeated this experiment again, and presented the new data in **Fig. 3e-3h** of the revised manuscript. In the new data, we evaluated the function of activated cytotoxic T cells by detecting IFN γ . The representative original flow cytometric plots have been shown in **Supplementary Fig. 4i** of the revised manuscript. We got the similar result as before. The result showed that the TNC knockout significantly reduced the tumor burden as confirmed by the growth curve of the xenograft tumor volume and the tumor weight in syngeneic BALB/c mice (**Fig. 3f, 3g** in the revised manuscript). Indeed, the TNC knockout

tumors had more CD4⁺ and CD8⁺ TIL populations and more activated cytotoxic T cells (IFN γ ⁺CD4⁺ and IFN γ ⁺ CD8⁺) in their TILs than the control tumors (**Fig. 3h**, **Supplementary Fig. 4i**, Please refer to highlighted sentence at **Page 14** in the revised manuscript).

8. Does the slight increase in TNC mRNA expression in ATG5 KO statistically significant? (a p-value is needed)

Response: To determine whether the increase in the TNC protein levels in the autophagy-deficient cells occurred at the transcription level, we have repeated this experiment again, and presented the new data in **Supplementary Fig. 5a** of the revised manuscript. The results showed that TNC had similar levels of mRNA in the autophagy-competent and the autophagy-incompetent MDA-MB-231 cells (fold change was not more than 2 times) , although the slight increase/decrease in TNC mRNA expression had statistical difference ($P < 0.05$) (Please refer to highlighted sentence at **Page 16**, **Fig. 5a** in the revised manuscript). In addition, we have detected the protein expression of TNC in autophagy-deficient cell models with multiple cell lines and added back. The result showed that that the protein level of TNC were substantially elevated in the autophagy-deficient MDA-MB-231 and MEF cells (fold change was more than 6 times), but blunted by restoration of autophagy (**Fig. 3c**, Please refer to highlighted sentence at **Page 13** in the revised manuscript). Therefore, we reasoned that TNC might be tightly regulated by the protein degradation pathway.

9. Fig4e: Do the authors tried to transfect their cells with a fluorescence-tagged LC3 and performed a staining with TNC to show that TNC is incorporated in autophagosomes?

Response: The reviewer has raised a good point. Confocal microscopy result revealed that the cytoplasmic punctate formation of TNC-p62 was significantly increased by EBSS- or hypoxia-induced autophagy (**Fig. 4e** in the revised manuscript). To determine whether TNC was incorporated in autophagosomes, we have done a new

experiment, and presented the new data in **Supplementary Fig. 5h** of the revised manuscript. The result showed that the ratio of punctation with fluorescence-tagged LC3 that localized with TNC was also significantly increased by EBSS-induced autophagy, which indicated that TNC was incorporated in autophagosomes (**Supplementary Fig. 5h**, Please refer to highlighted sentence at **Page 16** in the revised manuscript) .

10. In fig5g, I found that the decrease of cytoplasmic TNC-p62+ punctate in response to EBSS treatment and after Skp2 knockdown is difficult to visualize and to evaluate. A better quantification is needed

Response: We have repeated this experiment again, and presented the new data in **Fig.5g** of the revised manuscript. We also quantified the TNC-p62+ punctat in another way. We got the similar result as before. The result also showed that cytoplasmic punctate formation of TNC-p62 by EBSS-induced autophagy reduced following the Skp2 knockdown (**Fig. 5g**, Please refer to highlighted sentence at **Page 20** in the revised manuscript).

11. The discussion has to be improved , more concise and relevant.

Response: We have modified the discussion carefully (Please refer to highlighted sentence at **Page 31-34** in the revised manuscript). We mainly discussed this study from three aspects: (1) Defective autophagy is a key immunosuppressive factor in TNBC tumors. As autophagic potential may be impaired in TNBC tumors, this finding may at least partially explain why most TNBC patients are still resistant to immune checkpoint inhibitors. (2) TNC is a key regulator of autophagy-deficiency-mediated immunosuppression. Abnormalities in selective autophagic degradation lead to the accumulation of TNC. (3) TNC is a candidate therapeutic agent for developing combination immunotherapy for TNBC, and which subgroups of TNBC maybe benefit from the combination strategy with inhibition of TNC and immune checkpoint inhibitors.

Reviewer #3 (Remarks to the Author): Expert in TNBC

This manuscript demonstrated that defects in autophagy inhibited T-cell mediated cell death in TNBC through TNC after ubiquitination by Skp2. This may have implications for potential enhancement of immunotherapy in TNBC.

Major concerns:

1) Key controls for Crispr knockouts were absent. Multiple clones, multiple cell lines or add backs should be completed to ensure that off target effects are not misleading the authors.

Response: The reviewer has raised a good point. To elucidate the role of autophagy in the T-lymphocyte immune system, we have established autophagy-deficient cell models by multiple clones, multiple cell lines or add backs, and presented the new data in the revised manuscript. These autophagy-deficient cell and animal models include : (1) MDA-MB-231 WT, MDA-MB-231 Atg5 KO#2, MDA-MB-231 Atg5 KO#2+Atg5 cDNA, MDA-MB-231 Atg5 KO#4, and MDA-MB-231 Atg5 KO#4 +Atg5 cDNA (**Fig 1b-1c, Fig.1g, Supplementary Fig.2a-2b, 2e-2f**); (2) MDA-MB-231 WT, MDA-MB-231 Atg7 KO, and MDA-MB-231 Atg7 KO+Atg7 cDNA (**Fig 1d-1e, Supplementary Fig.2c-2d, 2g-2h**); (3) 4T1-WT and 4T1-Atg5 KO xenografts in immunocompetent BALB/c mice (**Fig 2a,2b**); (4) 4T1 WT and 4T1 Beclin1 KO xenografts in immunocompetent BALB/c mice (**Supplementary Fig 3a-3c**); (5) Compare the antitumor effect of a PD1 antibody between 4T1-WT and 4T1-Beclin1 KO xenografts in immunocompetent BALB/c mice (**Fig 2c-2f**); (6) Compare the antitumor effect of PDL1 antibody between 4T1-WT and 4T1-Atg5 KO xenografts in immunocompetent BALB/c mice (**Fig 2g-2k**); (7) B16-OVA WT, B16-OVA Atg7 KO#1, and B16-OVA Atg7 KO#2 (**Fig 1h, Supplementary Fig.2o**); (8) MEF WT and MEF Atg5^{-/-} (**Supplementary Fig 2i-2k**). In addition, we also silenced the expression of autophagy-related genes in the MDA-MB-231 cells, including Atg3, Atg5, Beclin 1, Atg7, Atg10, Atg13, Atg14, vsp34, uvrag (**Fig.1f, Supplementary Fig2i-2m**). All these data demonstrated that defects in autophagy

reduced T-cell mediated cytotoxicity in TNBC *in vitro* and *in vivo*, and this decrease was markedly reversed by recovery of autophagy.

2) Given the heterogeneity and subtypes of TNBC, the authors should test whether the known subtypes of TNBC have differences in RFS based on TNC levels. Indeed, this is likely the single largest flaw with the manuscript - TNBC encompasses a wide variety of tumors and this is not a mechanism that will be broadly applicable to the entire class of tumors. Finding the subset of patients where this mechanism is valid would substantially benefit the manuscript.

Response: The reviewer has raised a good point. To test whether the known subtypes of TNBC have differences in RFS based on TNC levels, we performed the Kaplan-Meier analyses using the KM-plotter breast cancer database (<http://kmplot.com/analysis/>), and presented the new data in **Fig. 7b** of the revised manuscript. Basal patients were stratified according the six known subtypes , including basal-like1, basal-like2, immunomodulatory, mesenchymal, mesenchymal stem-like, and luminal androgen receptor subtypes. The result showed that high TNC expression level was associated with poor RFS in basal-like 1, basal-like 2, immunomodulatory, mesenchymal and mesenchymal stem-like subtypes, especially in basal-like and immunomodulatory subtypes which had significance ($P < 0.05$). But patients with luminal androgen receptor subtype showed the opposite trend (**Fig. 7b**, Please refer to highlighted sentence at **Page 23** in the revised manuscript). That means except TNBC patients with luminal androgen receptor subtype, the other TNBC patients with basal-like 1, basal-like 2, immunomodulatory, and mesenchymal, mesenchymal stem-like subtypes , especially with basal-like and immunomodulatory subtypes, maybe benefit from the combination strategy with inhibition of TNC and immune checkpoint inhibitors. (Please refer to highlighted sentence at **Page34** in the revised manuscript) .

Minor:

Figure 9E is not appreciated by color blind reviewers - no staining is observed.

Response: We have repeated the IHC staining of CD4, CD8 and granzyme B in paraffin-embedded tissue sections from 4T1-Atg5 KO tumors, and presented the new data in **Fig.9e** of the revised manuscript. We got the similar result as before. The cells stained brown were positive cells. The result showed that tumors with the combined treatment not only had increased CD4⁺ and CD8⁺ TIL populations but also displayed a substantially increased granzyme B release compared to tumors with each single agent treatment (**Fig.9e** in the revised manuscript).

The level of detail in the methods is appropriate as were statistical tests.

Response: We have modified the method of **Statistical Analysis** in the revised manuscript. Statistical analyses were conducted using GraphPad Prism software and SPSS software. The results of quantitative experiments were reported as mean ± SEM of three independent experiments. Survival curves were plotted by the Kaplan-Meier method. Univariate and multivariate Cox proportional hazards regression were carried out to identify the protein marker as an independent predictor of survival. The relationship between TNC expression and LC3B was assessed using Pearson's chi-square test. All other comparisons were analyzed by unpaired two-tailed Student's t test. A *P* value of <0.05 was considered to indicate statistical significance(Please refer to highlighted sentence at **Page46-47** in the revised manuscript).

References

1. Gnjatic S, Cai Z, Viguier M, Chouaib S, Guillet JG, Choppin J. Accumulation of the p53 protein allows recognition by human CTL of a wild-type p53 epitope presented by breast carcinomas and melanomas. *Journal of immunology* **160**, 328-333 (1998).
2. Shukla SA, *et al.* Cancer-Germline Antigen Expression Discriminates Clinical Outcome to CTLA-4 Blockade. *Cell* **173**, 624-633 e628 (2018).
3. Jiao L, *et al.* Regulation of glycolytic metabolism by autophagy in liver cancer involves selective autophagic degradation of HK2 (hexokinase 2). *Autophagy* **14**, 671-684 (2018).
4. Husmann K, Faissner A, Schachner M. Tenascin promotes cerebellar granule cell migration and neurite outgrowth by different domains in the fibronectin type III repeats. *The Journal of cell biology* **116**, 1475-1486 (1992).
5. Jiao S, *et al.* PARP Inhibitor Upregulates PD-L1 Expression and Enhances Cancer-Associated Immunosuppression. *Clinical cancer research : an official journal of the American Association for Cancer Research* **23**, 3711-3720 (2017).
6. Michaud M, *et al.* Autophagy-dependent anticancer immune responses induced by chemotherapeutic agents in mice. *Science* **334**, 1573-1577 (2011).
7. Peng W, *et al.* Loss of PTEN Promotes Resistance to T Cell-Mediated Immunotherapy. *Cancer discovery* **6**, 202-216 (2016).
8. Noman MZ, *et al.* Blocking hypoxia-induced autophagy in tumors restores cytotoxic T-cell activity and promotes regression. *Cancer research* **71**, 5976-5986 (2011).
9. Mgrditchian T, *et al.* Targeting autophagy inhibits melanoma growth by enhancing NK cells infiltration in a CCL5-dependent manner. *Proceedings of the National Academy of Sciences of the United States of America* **114**, E9271-E9279 (2017).
10. Amaravadi RK. Cancer. Autophagy in tumor immunity. *Science* **334**, 1501-1502 (2011).
11. Li T, *et al.* TIMER: A Web Server for Comprehensive Analysis of Tumor-Infiltrating Immune Cells. *Cancer research* **77**, e108-e110 (2017).
12. Guttery DS, Shaw JA, Lloyd K, Pringle JH, Walker RA. Expression of tenascin-C and its isoforms in the breast. *Cancer metastasis reviews* **29**, 595-606 (2010).
13. Orend G, Chiquet-Ehrismann R. Tenascin-C induced signaling in cancer. *Cancer letters* **244**, 143-163 (2006).

14. Swindle CS, *et al.* Epidermal growth factor (EGF)-like repeats of human tenascin-C as ligands for EGF receptor. *The Journal of cell biology* **154**, 459-468 (2001).
15. Huang JY, *et al.* Extracellular matrix of glioblastoma inhibits polarization and transmigration of T cells: the role of tenascin-C in immune suppression. *Journal of immunology* **185**, 1450-1459 (2010).
16. Jachetti E, *et al.* Tenascin-C Protects Cancer Stem-like Cells from Immune Surveillance by Arresting T-cell Activation. *Cancer research* **75**, 2095-2108 (2015).
17. Mirzaei R, *et al.* Brain tumor-initiating cells export tenascin-C associated with exosomes to suppress T cell activity. *Oncoimmunology* **7**, e1478647 (2018).
18. Wenk MB, Midwood KS, Schwarzbauer JE. Tenascin-C Suppresses Rho Activation. *The Journal of cell biology* **150**, 913-920 (2000).
19. Woodside DG, Wooten DK, McIntyre BW. Adenosine Diphosphate (ADP)-Ribosylation of the Guanosine Triphosphatase (GTPase) Rho in Resting Peripheral Blood Human T Lymphocytes Results in Pseudopodial Extension and the Inhibition of T Cell Activation. *The Journal of experimental medicine* **188**, 1211-1221 (1998).
20. Hauzenberger D, Olivier P, Gundersen D, Ruegg C. Tenascin-C inhibits beta1 integrin-dependent T lymphocyte adhesion to fibronectin through the binding of its fnIII 1-5 repeats to fibronectin. *European journal of immunology* **29**, 1435-1447 (1999).

Reviewers' comments:

Reviewer #1 (Remarks to the Author):

Fig.1G, this experimental setup is unconventional. If p53 is expressed by MDA-MB-231 cells already, what is the need to add p53 peptides? Also the author did not explain why they loaded DC with p53 peptides? If the purpose was to activate p53-specific T cells, they should show the frequency of such T cells in PBMC from their donors. Usually the frequency is too low to be meaningful.

Fig.1H and suppl Fig.2O. The description of crispr/cas9 approach is not adequate. The following information should be provided. 1st, the region of the genes that are targeted; 2nd the sequences of guide RNA should be given; 3rd, the genomic analysis confirming the successful targeting must be provided; 4th, the full western plots should be shown with molecular size markers clearly labeled; 5th, the authors have to indicate how many clones have been tested to verify the results.

Fig2. The authors should clarify the source of TUNEL signals, i.e. from tumor cells or immune cells. The flow cytometry should be performed to clarify the cellular source of granzyme B.

Fig3E. The authors should clarify how they made the crispr lines instead of clones throughout the manuscript, particularly in the method section. In addition, they need to clarify what control cell lines they used. Cas9 protein can be quite immunogenetic. Therefore, they need to show whether control cell lines express cas9 protein and whether the level of cas9 protein is similar to experimental cell lines. In addition, they need to show whether the growth of vector control cell lines is similar to parental cell lines.

Fig3H. The authors did not address my concerns.

Fig7. The IHC should be validated using positive and negative control samples. For examples, antibodies can be tested with cell lines that overexpress and under-express TNC as positive and negative controls respectively.

Fig8. It is not clear how P53 antigen-specific human T lymphocytes were generated. The authors need to clarify how these cells are generated and the frequency of p53-specific T cells.

Reviewer #2 (Remarks to the Author):

I am satisfied with the changes made.

Reviewer #3 (Remarks to the Author):

The authors have addressed my concerns, except for making Figure 9 color blind friendly. I cannot see a difference in panel E. Is this a color blind issue or does this suggest that the CD4 / CD8 / granzyme response is not significant?

Quantitation of the result across many different fields of view and not just the small selected areas would help.

We would like to take this opportunity to thank you for your thoughtful critiques and constructive comments that helped us to improve our manuscript. Based on your kind advices, we have extensively revised the original manuscript with some new experiments. Also, the text of manuscript has been checked for formatting. We have corrected the mistakes and some information has been added in the corresponding text. Response point to point as listed below.

Point-by-point response

Reviewer #1 (Remarks to the Author):

Specific critiques:

Fig.1G

This experimental setup is unconventional. If p53 is expressed by MDA-MB-231 cells already, what is the need to add p53 peptides? Also the author did not explain why they loaded DC with p53 peptides? If the purpose was to activate p53-specific T cells, they should show the frequency of such T cells in PBMC from their donors. Usually the frequency is too low to be meaningful.

Response : The reviewer has raised a good point. As p53 is expressed by MDA-MB-231 cells already, we did not add p53 peptides to load MDA-MB-231 cells to do the specific killing effect of T cells. We are sorry for the misunderstanding caused by our incorrect annotation in Fig.1g. We have corrected this mistake and added some new data in this revised manuscript. We also have modified the detailed protocol in the method section of “T lymphocytes preparations” , and provided the frequency of P53₂₆₄₋₂₇₂ Tetramer⁺CD8⁺ T cells (Please refer to highlighted sentence at **Page 6, Page 40-41** in the revised manuscript).

The purpose of the study is to investigate whether autophagy failure contributes to the limitation of T-lymphocyte attack on triple-negative breast cancer (TNBC) cells. We screened a panel of TNBC cell lines and found that only MDA-MB-231 cell line expressed HLA-A2, which is important for tumor cell recognition by HLA-A2⁺ T cells (Supplementary Fig. 1). To elucidate the role of autophagy of MDA-MB-231 cells in inducing T cell immune response, we have established autophagy-deficient

MDA-MB-231 cell models by multiple clones, and add backs. Then the tumor cells were co-cultured with CD3/CD28-activated human HLA-A2⁺ T lymphocytes. All these data demonstrated that autophagy defects in MDA-MB-231 cells reduced T-cell mediated cytotoxicity by non-specific T cell killing assay *in vitro*, and this decrease was markedly reversed by recovery of autophagy (Fig. 1b-1f, Supplementary Fig. 2a-2h).

Then we further measured the antigen specific T cell-mediated cytotoxicity on the autophagy-deficient MDA-MB-231 cells. Peptide 264–272 from naturally processed p53 has proven to be a potential T epitope because of its strong affinity to HLA-A2¹, and MDA-MB-231 cells is reported to display high p53 concentrations in the nucleus due to a p53 gene mutation in codon 280². Using competition assays, Gnjatic et al. demonstrated that tumor lysis by D5/L9V, a CTL line directed against HLA-A2-restricted peptide 264–272 from wild-type p53, was due to recognition of endogenously produced p53 peptide 264–272 associated with the HLA-A2.1 molecule on the surface of tumor cells (including MDA-MB-231, MCF-7, and melanoma M8 cell lines)². Our result also showed the high levels of p53 protein in autophagy-deficient MDA-MB-231 cell lines, which was similar to the autophagy-competent MDA-MB-231 cell lines (**Supplementary Fig.2n** in the revised manuscript). Considering that p53 is highly expressed by MDA-MB-231 cells, we generated the P53₂₆₄₋₂₇₂ peptide-specific T cells by stimulated by autologous DC loading with P53₂₆₄₋₂₇₂.

In the experiment, the DCs loaded with P53₂₆₄₋₂₇₂-antigen were cocultured with autologous T lymphocytes from HLA-A2⁺ healthy donor to induce P53 peptide-specific T cells. T cells stimulated with no peptide-pulsed DCs were used as negative control. The result showed that the frequency of P53₂₆₄₋₂₇₂ tetramer⁺ CD8⁺ T cells increased from 1.95% to 10.2% after stimulation with with P53₂₆₄₋₂₇₂ peptide-pulsed DCs (**Supplementary Fig.2o** in the revised manuscript). The cytotoxicity of P53 peptide-pulsed DC-treated T cells targeting MDA-MB-231 cells were higher than that of negative control (**Fig.1g** in the revised manuscript). These data suggest that T cells stimulated with P53₂₆₄₋₂₇₂ peptide-pulsed DCs could kill

MDA-MB-231 cells specifically by recognition of endogenous p53 epitope antigen presented by tumor cells. As expected, we observed that the cytotoxicity of P53-specific T cells to the MDA-MB-231 Atg5-KO cells was reduced, but the cytotoxicity was recovered when restoration of Atg5 (**Fig.1g** in the revised manuscript). Altogether, these data confirm that autophagy failure contributes to the limitation of T-lymphocyte attack on TNBC cells.

Additionally, we have used another antigen specific T cell-mediated cytotoxicity model. We depleted Atg7 in ovalbumin (OVA) positive melanoma B16F10 cells (**Supplementary Fig. 2p** in the revised manuscript). Then the cells were co-cultured with activated CD8⁺ T cells isolated from OT-I TCR transgenic mice, which providing an antigen-specific T cell-mediated killing to B16F10-OVA cells. The data also showed that compared to their autophagy-competent counterparts, the autophagy-deficient B16F10-OVA Atg7 KO cells were more resistant to the antigen-specific T cell-mediated killing (**Fig.1h**).

Fig. 1 Autophagy deficiency in tumor cells reduces T cell-mediated tumor killing *in vitro*.

(g) The indicated MDA-MB-231 cells were cocultured with bulk populations of p53₂₆₄₋₂₇₂ peptide-specific human T cells. Upper, representative dot plots of the cleavage of caspase-3 in tumor cells measured by flow cytometry. Bottom, percentage of cleaved caspase-3⁺ tumor cells. (h)

The indicated B16F10-OVA cells were treated with IFN γ for 24h, then cocultured with OT-1 CD8⁺ T cells isolated from spleen of OT-I TCR transgenic mice. Left, representative dot plots of the cleavage of caspase-3 in tumor cells measured by flow cytometry. Right, percentage of cleaved caspase-3⁺ tumor cells. Error bars represent mean \pm SEM of triplicates. All data are representative of three independent experiments. NS, not significance, ** $P < 0.01$, * $P < 0.05$.

Supplementary Fig. 2 Autophagy deficiency reduces T cell-mediated tumor killing *in vitro*.

(n) The expression of P53 in the indicated MDA-MB-231 cell lines. (o) Autologous purified T cells were cocultured with p53 peptide-pulsed DCs followed by stained with p53₂₆₄₋₂₇₂ tetramer and anti-CD8. The numbers in the upper right quadrants (Q2) of dot plots indicate the frequency of p53₂₆₄₋₂₇₂ tetramer⁺CD8⁺ T cells. (p) Effect of Atg7 knockout in B16F10-OVA cells using CRISPR- Cas9 technology. All data are representative of three independent experiments.

Fig.1H and suppl Fig.2O.

The description of crispr/cas9 approach is not adequate. The following information should be provided. 1st, the region of the genes that are targeted; 2nd the sequences of guide RNA should be given; 3rd, the genomic analysis confirming the successful targeting must be provided; 4th, the full western plots should be shown with molecular size markers clearly labeled; 5th, the authors have to indicate how many clones have been tested to verify the results.

Response: The reviewer has raised a good point. We have modified the method section of “CRISPR–Cas9-mediated Gene Disruption” (Please refer to highlighted sentence at **Page 36-38** in the revised manuscript). In this section, we clarified how the CRISPR–Cas9-mediated knockout cell lines were established and what control cell lines we used.

To establish autophagy-deficient cell model in mouse B16F10-OVA cell line, mouse Atg7 Double Nickase Plasmid was purchased from Santa Cruz Co. This plasmid was transfection-ready purified DNA plasmid. It was transiently transfected into B16F10-OVA cells using Lipofectamine 2000 (Invitrogen). After 48h, puro positive cells were dissociated and seeded at subcloning density. **Atg7-knockout clones (Atg7 KO#1 and Atg7 KO#2)** were isolated by single-cell dilution cloning and used for further analysis. The knockout cells were identified by immunoblot and sequencing. Control CRISPR/Cas9 Double Nickase Plasmid (sc-437281) from Santa Cruz Co. was used as a negative control. The detailed information of CRISPR/Cas9 KO plasmids was provided as follows:

Targeted genes	Product Cat#	Product Name	Sequences of guide RNA	Targeted region of the genes
Mouse Atg7	sc-42880 5-NIC	ATG7 Double Nickase Plasmid (m)	Plamid A: GAAGTTGAAC GAGTACCGCC	chr6:114,673,079-114,673,098
			Plamid B: GTGCCAGAAG CCAACGTCCA	chr6:114,673,045-114,673,064

The representative information of the genomic analysis confirming the successful targeting was provided as follows (bracketed numbers refer to number of deleted base pairs and inserted nucleotides):

clones	Sequences (5'-3')
B16F10-OVA Atg7 KO#1	TTTAATAGTGCCCTGGACGTTGG [+7] CTTCTGGCACGAA CTGACC [-5,+6] GAAGT TGAACGAGTACCGCC
B16F10-OVA Atg7 KO#2	TTTAATAGTGCCCTGGACGT[-8,+38]GGCACGAACTGACC CAGAAGAAGTTGA[+14] ACGAGTACCGCC

The knockout cells were also identified by the full western plots with molecular size markers (**Supplementary Fig. 2p** in the revised manuscript)

Supplementary Fig. 2 Autophagy deficiency reduces T cell-mediated tumor killing *in vitro*.
(p) Effect of Atg7 knockout in B16F10-OVA cells using CRISPR- Cas9 technology.

Fig2. The authors should clarify the source of TUNEL signals, i.e. from tumor cells or immune cells. The flow cytometry should be performed to clarify the cellular source of granzyme B.

Response: Actually, granzyme B is most commonly found in the granules of natural killer cells (NK cells) and cytotoxic T cells. Granzyme B is also highly expressed in 5-30% of CD4⁺ Foxp3⁺ Tregs in tumors.

In Fig.2c-2f, we compared the antitumor effect of a PD1 antibody between 4T1-WT and 4T1-Beclin1 KO tumors. The result showed that the systemic PD1 treatment obviously limited the tumor volume of the 4T1-WT xenografts when such tumors were grown in normal, immunocompetent mice, and the tumor growth inhibition was nearly 65% according to the tumor weight (Fig. 2d). However, the PD1 treatment slightly limited the growth of the 4T1-Beclin1 KO tumors, and had no obvious statistical difference (Fig. 2e). Also PD1 treatment resulted in an increased number of TUNEL positive cells in the 4T1-WT tumors (Fig. 2f). To clarify the cell sources of granzyme B in 4T1-WT tumors treated with anti-PD1, we originally want

to perform the immunofluorescence staining of both CD8 and granzyme B in the same OCT-embedded frozen tissue slices and quantify the proportion of co-location of CD8 and granzyme B. But we performed this animal experiment one year ago and the frozen tissue slices were kept in the -80 degree refrigerator for too long. We found the ice crystals in the frozen tissue slices were particularly serious which had pierced cells and destroyed tissue morphology. Therefore, we deleted this immunofluorescence picture of granzyme B in the experiment of 4T1-WT tumors treated with anti-PD1 in the # NCOMMS-19-00894B manuscript (the first revision).

To further evaluate the effect of autophagy on T cell-mediated antitumor activity in vivo, we then did a new experiment to compare the antitumor effect of PD-L1 antibody between 4T1-WT and 4T1-Atg5 KO tumors, and presented the new data in **Fig. 2g-2k** in the # NCOMMS-19-00894B manuscript (the first revision). IFN γ is one of T-cell modulation of activation markers. In this experiment, we evaluated the activated cytotoxic T cells by detecting the frequency of IFN γ ⁺ CD4⁺T and IFN γ ⁺ CD8⁺T cells instead of granzyme B, just like other studies reported^{3, 4}. Since we focused on the function of cytotoxic T cells, we did not detect the level of Treg and granzyme B expression in Treg in the new experiment. In addition, we got the similar result when compared the antitumor effect of a PDL1 antibody between 4T1-WT and 4T1-Atg5 KO tumors (Fig. 2g). The result showed that anti-PDL1 treatment failed to control the growth of the 4T1- Atg5 KO tumors compared to 4T1-WT tumors in normal, immunocompetent mice (Fig. 2h, 2i). Actually, the anti-PDL1 treatment stimulated more activated cytotoxic CD4 T and CD8 T cells in their TILs (IFN γ ⁺CD4⁺ and IFN γ ⁺ CD8⁺) and resulted in an increased number of TUNEL positive cells in 4T1-WT tumors (Fig. 2j, 2k). To clarify the source of TUNEL signals in Fig2k, we did new experiments to perform double immunofluorescent staining for TUNEL and EpCAM (a marker for epithelial cell), and for TUNEL and CD45 (a marker for leukocyte). We presented the new data in the second revision. The double immunofluorescent staining showed that the apoptotic cells were mainly from EpCam⁺ tumor cells, not from CD45⁺ immune cells (**Supplementary Fig. 3d**, Please refer to highlighted sentence at **Page 10, Page 47** in the revised manuscript).

Fig. 2 Loss of autophagy promotes resistance to T cell-mediated anti-tumor activity *in vivo*.

(c) Tumor growth of 4T1 breast tumor xenografts in BALB/c mice following treatment with mouse anti-PD1 antibody. The treatment protocol is summarized by the arrows. (d,e) Tumor growth of 4T1-WT cells (d) or 4T1-Beclin 1 KO cells (e) in BALB/c mice with anti-PD-1 antibody treatment (n=6 per group). Tumor volumes (left) and tumor weights upon autopsy on day 18 (right) were calculated. (f) Representative images of TUNEL staining of 4T1-WT OCT-embedded frozen tumor sections after treatment with mouse anti-PD1 in (d) (left). The apoptotic cells with DNA fragmentation were stained positively as green nuclei. Scar bar, 50um. Quantification of positive TUNEL cells on the right. (g) Tumor growth of 4T1 breast tumor xenografts in BALB/c mice following treatment with mouse anti-PDL1 antibody. The treatment protocol is summarized by the arrows. (h-i) Tumor growth of 4T1 WT cells (h) or 4T1-Atg5 KO cells (i) in BALB/c mice with mouse anti-PDL1 antibody treatment (n=8 per group). Tumor volumes (left) and tumor weights upon autopsy on day 14 (right) were calculated. (j) FACS analysis of IFN γ ⁺ in CD45⁺CD4⁺ and CD45⁺CD8⁺ T cell populations from the isolated TILs in (h)(right). Representative dot plots from a representative mouse for each group (left). (k) Representative images of TUNEL staining of 4T1-WT formalin-fixed paraffin-embedded tumor sections after treatment with mouse anti-PDL1 in (h) (left). Scar bar, 50um. Quantification of positive TUNEL cells (right). Error bars represent mean \pm SEM. All data are representative of two independent experiments. NS, not significance, * $P < 0.05$, ** $P < 0.01$.

Supplementary Fig. 3 Autophagy deficiency reduces T cell-mediated tumor killing *in vivo*.

(d) Immunofluorescent double staining for TUNEL and EpCAM in the same section from anti-PDL1-treated 4T1-WT groups. (e) Immunofluorescent double staining for TUNEL and CD45 in the same section from anti-PDL1-treated 4T1-WT groups. Scar bar, 10um.

Fig3E. The authors should clarify how they made the crispr lines instead of clones throughout the manuscript, particularly in the method section. In addition, they need to clarify what control cell lines they used.

Response: The reviewer has raised a good point. We have modified the method section of “CRISPR–Cas9-mediated Gene Disruption” carefully (Please refer to highlighted sentence at **Page 36-38** in the revised manuscript). In this section, we clarified how the CRISPR–Cas9-mediated knockout cell lines were established and what control cell lines we used.

To establish autophagy-deficient cell models, Atg5 Double Nickase Plasmids (h/m), Atg7 Double Nickase Plasmids (h/m) and Beclin1 Double Nickase Plasmid (m) were purchased from Santa Cruz Co. All of these plasmids were transfection-ready purified DNA plasmids. According to the manufacturer's instruction, these CRISPR/Cas9 KO plasmids were transiently transfected into MDA-MB-231, 4T1 or B16F10-OVA cell lines using Lipofectamine 2000 (Invitrogen). After 48h, puro positive cells were dissociated and seeded at subcloning density. Atg5-knockout, Atg7-knockout and Beclin1-knockout clones were isolated by single-cell dilution cloning from the positive polyclonal sgRNA-transduced populations. All knockout clones were identified by immunoblot and sequencing. Control CRISPR/Cas9 Double Nickase Plasmid (sc-437281) from Santa Cruz Co. was used as a negative control. The detailed information of these CRISPR/Cas9 KO plasmids was provided as follows:

Targeted genes	Product Cat#	Product Name	Sequences of guide RNA	Targeted region of the genes
Human Atg5	sc-41684 7-NIC	ATG5 Double Nickase Plasmid (h)	Plamid A: GAATATCCTG CAGAAGAAAA	chr6:106,696,05 9-106,696,078
			Plamid B: TCCGATTGAT GGCCCAAAC	chr6:106,696,08 7-106,696,106
Mouse Atg5	sc-41914 9-NIC	ATG5 Double Nickase Plasmid (m)	Plamid A: AGTGAAAAAG CACTTTCAGA	chr10:44,289,90 7-44,289,926
			Plamid B: AACGTCAAAT AGCTGACTCT	chr10:44,289,87 5-44,289,894
Human Atg7	sc-40099 7-NIC	ATG7 Double Nickase Plasmid (h)	Plamid A: TGCCCCTTTTA GTAGTGCCT	chr3:11,340,214 -11,340,233
			Plamid B: AACTGCAGTT TAGAGAGTCC	chr3:11,340,194 -11,340,213
Mouse Atg7	sc-42880 5-NIC	ATG7 Double Nickase Plasmid (m)	Plamid A: GAAGTTGAAC GAGTACCGCC	chr6:114,673,07 9-114,673,098
			Plamid B: GTGCCAGAAG CCAACGTCCA	chr6:114,673,04 5-114,673,064
Mouse Beclin1	sc-42503 3-NIC	BECN1 Double Nickase Plasmid (m)	Plamid A: GGACACGAGC TTCAAGATCC	chr11:101,301,7 58-101,301,777
			Plamid B: TCAGAGGCTG GCTACAGCGC	chr11:101,301,7 82-101,301,801

The representative information of the genomic analysis confirming the successful targeting was provided as follows (bracketed numbers refer to number of deleted base

pairs and inserted nucleotides):

clones	Sequences (5'-3')
MDA-MB-231 Atg5 KO#2	AAATCCATTTTCTTCTGCAGGATA[-37]AATCTGTCTGTAA TGATATA
MDA-MB-231 Atg5 KO#4	AAATCCATTTTCTTCTGCAG[+34]GATATTCCATGAGTTTC CGATTGA TGGCCCA AA ACTGGTCAAATC
MDA-MB-231 Atg7 KO#5	GGGATCCTGGACTCTCTAAA[-24,+2]GCCTTGGATGTTGGG TTTT
B16F10-OVA Atg7 KO#1	TTTAATAGTGCCCTGGACGTTGG [+7] CTTCTGGCACGAA CTGACC [-5,+6] GAAGT TGAACGAGTACCGCC
B16F10-OVA Atg7 KO#2	TTTAATAGTGCCCTGGACGT[-8,+38]GGCACGAACTGACC CAGAAGAAGTTGA[+14] ACGAGTACCGCC
4T1-Atg5 KO#1	GACGTTGGTAACTGACAAAG[-12,+1]TTCAGAAGGTTATG AGACAA
4T1-Beclin1 KO#2	CCGGTCCAGGATCTTGAAGC[-25]ACAGCGCTGGCACACG AAGCT

For CRISPR-Cas9-mediated TNC (m) knockout in 4T1-Atg5 KO#1 cells, the specific sgRNA sequence AGGGCAGTATGAGCTCCGGG (region: chr4: 63,964,709-63,964,728) targeting mouse TNC gene was cloned to LentiCRISPR (pXPR_001) plasmid. The packaging plasmids were co-transfected with pXPR-TNC sgRNA into HEK293T cells, and viral particles were harvested at 48 hr post-transfection. 4T1 Atg5 KO#1 cells were infected with viruses for 24 hr in the presence of polybrene (8 µg/ml), and stable cells were subsequently selected by puro for 3 days. The knockout cells were identified by immunoblot and sequencing. LentiCRISPR (pXPR_001) plasmid was used as a negative control. We did not select monoclonal cells from the positive cells because viral infection was very efficient. Although there was a very weak band of TNC in TNC KO lane, the knockout efficiency of TNC maybe more than 90% (**Fig. 3e** in the revised manuscript). The detailed information of the genomic analysis confirming the successful targeting was provided as follows (bracketed numbers refer to number of deleted base pairs and inserted nucleotides, the mutations in lower case, N/N indicates positive colonies out of total sequenced) :

Cell line	Sequences (5'-3')
4T1-Atg5 KO#1	TGGTCTTGTAGGTCCACCCG[-1]AGCTCATACTGCCCTTGGGC (17/24)
TNC KO	TGGTCTTGTAGGTCCACCCGGG[+1]AGCTCATACTGCCCTTGG (2/24)
	TGGTCTTGcAcGTCCACCCG[-1]AGCTCAgACTGtCaTTGG (1/24)
	TGGTCTcGTAGGatCACgCGatG[-1]TCAaACaGCgCTTgTGC (1/24)
	TGGTCTCG[-48] GGTTATCCAGTCCTGTGGAT (1/24)
	TGGTCTTGTAGGTCCA[-11] TACTGCCCTTGGGCTGTGAT (1/24)
	TGGTCT[-47] CAGGTTATCCAGTCCTGTGGAT (1/24)

Fig3E. Cas9 protein can be quite immunogenetic. Therefore, they need to show whether control cell lines express cas9 protein and whether the level of cas9 protein is similar to experimental cell lines.

Response: The reviewer has raised a good point. We have done a new experiment to evaluate the Cas 9 protein expression among 4T1-Atg5 KO#1 parental, vector control and TNC-KO cell lines by WB. We presented the new data in **supplementary Fig.4g** of the revised manuscript. The result showed that the level of cas9 protein in vector control cell line was similar to TNC-KO cell line. The parental 4T1-Atg5 KO#1 cell line was isolated by single-cell dilution cloning from the puro-positive polyclonal mouse ATG5 sgRNA-transduced 4T1 populations. Mouse Atg5 Double Nickase Plasmid (sc-419149-NIC, Santa Cruz Co.) was a transfection-ready purified DNA plasmid. When the plasmid was transiently transfected into 4T1 cells using Lipofectamine 2000, it would cut the cell's genome at the desired location, but it did not integrate with the host cell DNA in general. And the plasmid would lost during cell passage. Therefore, there was no cas9 protein expression in parental 4T1-Atg5 KO#1 cell line (**supplementary Fig.4g**, Please refer to highlighted sentence at **Page 14** in the revised manuscript).

Supplementary Fig 4. TNC is involved in autophagy deficiency-mediated immunosuppression. (g) The expression of Cas9 in the indicated 4T1-Atg5 KO#1 cell lines.

Fig3E. In addition, they need to show whether the growth of vector control cell lines is similar to parental cell lines.

Response: The reviewer has raised a good point. We have done a new experiment to evaluate the growth rate among 4T1-Atg5 KO#1 parental, vector control and TNC knock-out cell lines by MTT assay. We presented the new data in **supplementary Fig.4h** of the revised manuscript. The result showed that the growth of vector control cell line was similar to parental cell line. Also reduction of TNC expression in autophagy-deficient 4T1 cells did not influence their proliferation in vitro (**supplementary Fig.4h**, Please refer to highlighted sentence at **Page 14** in the revised manuscript).

Supplementary Fig 4. TNC is involved in autophagy deficiency-mediated immunosuppression. (h) Growth curves of the indicated 4T1-Atg5 KO#1 cell lines over a 5-day period by MTT assay (n=3). Error bars represent mean \pm SEM.

Fig3H. The authors did not address my concerns.

Panel h flow cytometric plots need to be shown. Also the level of Treg and granzyme B expression in Treg should be shown.

Response: Actually, granzyme B is most commonly found in the granules of natural killer cells (NK cells) and cytotoxic T cells. Granzyme B is also highly expressed in 5-30% of CD4⁺ Foxp3⁺ Tregs in tumors.

In this study, we identified TNC as a key regulator of autophagy-deficiency-mediated immunosuppression. TNC is well known for its function in arresting T-cell proliferation and activation to overcome immune surveillance. Jachetti

et al. reported that specific silencing of TNC substantially dampened the effects of prostate cancer stem-like cells on T-cell modulation of activation markers, and fully rescued the capacity of CD4⁺T and CD8⁺T cells to produce IFN γ . While the frequency of IFN γ ⁺ CD4⁺T and IFN γ ⁺ CD8⁺T cells was dramatically reduced when soluble TNC was added to the cultures of human or mouse T cells³.

To further confirm the role of TNC in immune surveillance in autophagy-deficient 4T1-Atg5 KO tumors, we have repeated this experiment again, and presented the new data in **Fig. 3e-3h** in the # NCOMMS-19-00894B manuscript (the first revision). In this experiment, we evaluated the activated cytotoxic T cells by detecting the frequency of IFN γ ⁺ CD4⁺T and IFN γ ⁺ CD8⁺T cells instead of granzyme B, just like Jachetti et al reported³. Since we focused on the function of cytotoxic T cells, we did not detect the level of Treg and granzyme B expression in Treg in the new experiment. We got the similar result as before in the new experiment. The result showed that the TNC knockout significantly reduced the tumor burden as confirmed by the growth curve of the xenograft tumor volume and the tumor weight in syngeneic BALB/c mice (**Fig. 3e-3g** in the revised manuscript). Indeed, the TNC knockout tumors had more CD4⁺ and CD8⁺ TIL populations and more activated cytotoxic T cells (IFN γ ⁺CD4⁺ and IFN γ ⁺CD8⁺) in their TILs than the control tumors (**Fig. 3h**, Please refer to highlighted sentence at **Page 14-15** in the revised manuscript).

Fig. 3 TNC is overexpressed in autophagy-deficient TNBC cells and inhibits T-cell priming.

(e) The effect of TNC knockout in 4T1-Atg5 KO cells using CRISPR-Cas9 technology. (f, g) Tumour growth of indicated mouse 4T1-Atg5 KO cells in BALB/c mice. Tumour volumes were calculated (n=5) (f), and tumor weights from experiment on autopsy on day 27 (g). (h) FACS analysis of CD45⁺CD4⁺, CD45⁺CD8⁺, and IFN γ ⁺ in CD45⁺CD4⁺ and CD45⁺CD8⁺ T cell populations from the isolated TILs in (f) (right). Representative dot plots from a representative mouse for each group (left). Error bars represent mean \pm SEM. All data are representative of two independent experiments. ** $P < 0.01$, * $P < 0.05$.

Fig7. The IHC should be validated using positive and negative control samples. For examples, antibodies can be tested with cell lines that overexpress and under-express TNC as positive and negative controls respectively.

Response: The reviewer has raised a good point. In Fig.7, primary antibodies for anti-TNC (GTX12298, Genetex), anti-LC3B (NB100-2220, Novus), anti-CD8 (ab17147, Abcam) were used for human TNBC breast specimens. We bought these three antibodies commercially. The datasheets of the three antibodies have noted that these antibodies can be applied for IHC staining, and provided the sample figure of IHC staining. Here, we have done new experiments to further confirmed the antibody specificity for IHC, and presented the new data (**Supplementary Fig. 8a-8c** Please refer to highlighted sentence at **Page 45** in the revised manuscript). Normal mouse IgG and normal rabbit IgG were used as negative controls. The result showed that there was no any positive brown cells in IHC staining when the TNBC breast tissues were incubated with normal mouse/rabbit IgG (**Supplementary Fig. 8a**). We also provided Immunocytochemistry (ICC) staining to further ensure specificity. The result showed that the positive brown staining decreased obviously when endogenous TNC was knockdown (**Supplementary Fig. 8b** in the revised manuscript), and increased clearly when exogenous TNC was overexpressed (**Supplementary Fig. 8c** Please refer to highlighted sentence at **Page 23, Page 45** in the revised manuscript).

Supplementary Fig. 8 A validation for the specificity of the antibodies. (a) TNBC tissue sections were stained with anti-TNC, anti-LC3B, anti-CD8 and normal rabbit/mouse IgG for Immunohistochemistry (IHC) assay. (b) MDA-MB-231 Atg7 KO#5 cells stably expressed TNC shRNA were fixed and stained with anti-TNC and normal rabbit IgG for Immunocytochemistry (ICC) assay. (c) HeLa cells were transiently transfected with HA-tagged TNC plasmids for 48 h. The cells were fixed and stained with anti-TNC and normal rabbit IgG for ICC assay.

Fig8. It is not clear how P53 antigen-specific human T lymphocytes were generated. The authors need to clarify how these cells are generated and the frequency of p53-specific T cells.

Response : The reviewer has raised a good point. As p53 is expressed by MDA-MB-231 cells already, we did not add p53 peptides to load MDA-MB-231 cells to do the specific killing effect of T cells. We are sorry for the misunderstanding caused by our incorrect annotation in Fig.8c. We have corrected this mistake and added some new data in this revised manuscript. We also have modified the detailed protocol in the method section of “T lymphocytes preparations” , and provided the frequency of P53₂₆₄₋₂₇₂ Tetramer⁺CD8⁺ T cells (Please refer to highlighted sentence at **Page 26, Page 40-41** in the revised manuscript).

In the Fig.8c, we also used P53 antigen-specific T cell-mediated cytotoxicity assay to measure the synergistic effect. In the experiment, the DCs loaded with P53₂₆₄₋₂₇₂-antigen were cocultured with autologous T lymphocytes from HLA-A2⁺ healthy donor to induce P53 peptide-specific T cells. T cells stimulated with no peptide-pulsed DCs were used as negative control. The result showed that P53₂₆₄₋₂₇₂ Tetramer positive CD8⁺ T cells increased from 1.96 % to 9.97% after stimulation with P53₂₆₄₋₂₇₂-pulsed DCs (**Supplementary Fig.10b** in the revised manuscript). Also the cytotoxicity of P53 peptide-pulsed DC-treated T cells targeting MDA-MB-231 cells were higher than that of negative control (**Fig. 8c** in the revised manuscript). These data suggest that T cells stimulated with P53₂₆₄₋₂₇₂ peptide-pulsed DCs could kill MDA-MB-231 cells specifically by recognition of endogenous p53 epitope antigen presented by tumor cells. As expected, the blockage of TNC resensitized anti-PD-L1-treated MDA-MB-231 Atg7 KO cells to antigen-specific T cell-induced tumor killing (**Fig. 8c** in the revised manuscript).

Additionally, we have used another antigen specific T cell-mediated cytotoxicity model. We found the effect of mouse OT1-CD8⁺T cell-mediated killing in B16F10-OVA Atg7 KO cells also could be enhanced when anti-TNC and anti-PD-L1 were used synergically (Supplementary Fig. 10c).

Fig. 8 Blockade of TNC sensitizes checkpoint blockade immunotherapy *in vitro*. (c) MDA-MB-231 Atg7 KO cells were pre-treated with anti-TNC (10 ug/ml) or anti-PD-L1 (10ug/ml) for 2 hours, then co-cultured with P53 antigen-specific activated human T cells. Upper, representative dot plots of the cleaved caspase-3 in tumor cells measured by flow cytometry. Bottom, percentage of cleaved caspase-3⁺ tumor cells. Error bars represent mean \pm SEM of triplicates. All data are representative of three independent experiments. NS, not significance, ** $P < 0.01$.

Supplementary Fig. 10 Blockade of TNC sensitizes checkpoint blockade immunotherapy *in vitro*. (b) Autologous purified T cells were cocultured with p53 peptide-pulsed DCs, followed by stained with p53₂₆₄₋₂₇₂ tetramer and anti-CD8. The numbers in the upper right quadrants (Q2) of dot plots indicate the frequency of p53₂₆₄₋₂₇₂ tetramer⁺CD8⁺ T cells. (c) After treatment with IFN γ for 24h, B16F10-OVA Atg7 KO cells were pre-treated with anti-TNC (10 ug/ml) or anti-PD-L1 (10ug/ml) for 2 hours, then co-cultured with mouse OT1-CD8 T cells. Left, representative dot plots of the cleavage of caspase-3 in tumor cells measured by flow cytometry. Right, percentage of the cleaved caspase-3 in tumor cells. Error bars represent mean \pm SEM of triplicates. All data are representative of three independent experiments. NS, not significance, ** $P < 0.01$, * $P < 0.05$.

Reviewer #3 (Remarks to the Author):

The authors have addressed my concerns, except for making Figure 9 color blind friendly. I cannot see a difference in panel E. Is this a color blind issue or does this suggest that the CD4/CD8/granzyme response is not significant? Quantitation of the result across many different fields of view and not just the small selected areas would help.

Response: The reviewer has raised a good point. We provided more clearly representative images of IHC staining of CD4, CD8 and granzyme B, and quantified CD4, CD8 and granzyme B expression across many different fields of view in the revised manuscript. The result showed that tumors with the combined treatment not only had increased CD4⁺ and CD8⁺ TIL populations but also displayed a substantially increased granzyme B release compared to tumors with each single agent treatment (Fig.9e in the revised manuscript).

Fig. 9 Downregulation of TNC enhances the antitumor activity of PD1 Blockade in mice bearing autophagy-incompetent 4T1 tumors.

(e) Representative images of IHC staining of CD4, CD8 and granzyme B (GB) expression in

4T1-Atg5KO#1-Tet-on-TNC shRNA xenograft tumor sections after treatment in (c)(left). HPF, 400x magnification. Scar bar, 50um. Quantitative IHC analysis of CD4, CD8 and granzyme B expression(right). Error bars represent mean \pm SEM. NS, not significance, * $P < 0.05$, ** $P < 0.01$.

References

1. Gnjjatic S, Bressac-de Paillerets B, Guillet JG, Choppin J. Mapping and ranking of potential cytotoxic T epitopes in the p53 protein: effect of mutations and polymorphism on peptide binding to purified and refolded HLA molecules. *European journal of immunology* **25**, 1638-1642 (1995).
2. Gnjjatic S, Cai Z, Viguier M, Chouaib S, Guillet JG, Choppin J. Accumulation of the p53 protein allows recognition by human CTL of a wild-type p53 epitope presented by breast carcinomas and melanomas. *Journal of immunology* **160**, 328-333 (1998).
3. Jachetti E, *et al.* Tenascin-C Protects Cancer Stem-like Cells from Immune Surveillance by Arresting T-cell Activation. *Cancer research* **75**, 2095-2108 (2015).
4. Michaud M, *et al.* Autophagy-dependent anticancer immune responses induced by chemotherapeutic agents in mice. *Science* **334**, 1573-1577 (2011).

Reviewers' comments:

Reviewer #1 (Remarks to the Author):

1. Their data that "the frequency of P53264-272 tetramer+ CD8+ T cells increased from 1.95% to 10.2% after stimulation with P53264-272peptide-pulsed DCs" are not convincing. There is no control tetramer staining. It is not clear whether their signals were real. The frequency of tetramer+ CD8 T cells is too high compared to what have been reported (<https://cancerres.aacrjournals.org/content/62/12/3521.long>) particularly in normal PBMC.

2. they have provided additional details about the Crispr/cas9 strategies. It is clear they used clones in their studies. They have to clarify that in their text. If they have used clones, they have to clarified how many clones they have used to control for clonal variation. In addition, the exact clone characteristics, particularly the original sequencing results showing exact sequence insertion and deletion, need to be put in the supplemental materials. The entire Western blot indicating the size of the target protein and the possible truncated protein (due to Crispr/cas9-triggered deletion or truncation) in these clones need to be shown. It is not possible that the entire protein is absent due to small deletion or insertion. Assuming the "null" allele is a huge problem in the current literature involving Crispr/Cas9 technologies.

Dear reviewers,

We would like to take this opportunity to thank you for your thoughtful critiques and constructive comments that helped us to improve our manuscript. Based on your kind advices, we have extensively revised the manuscript. Hopefully, the improvements are acceptable. Response point to point as listed below.

Reviewer #1 (Remarks to the Author):

1. Their data that “the frequency of P53₂₆₄₋₂₇₂ tetramer⁺ CD8⁺ T cells increased from 1.95% to 10.2% after stimulation with P53₂₆₄₋₂₇₂peptide-pulsed DCs” are not convincing. There is no control tetramer staining. It is not clear whether their signals were real. The frequency of tetramer⁺ CD8 T cells is too high compared to what have been reported (<https://cancerres.aacrjournals.org/content/62/12/3521.long>) particularly in normal PBMC.

Response : Thank you for the helpful suggestion. We are sorry for not providing the control tetramer staining figures in the last second revision. We have corrected this mistake and added new data in this revised manuscript. The frequency of P53₂₆₄₋₂₇₂ Tetramer⁺CD8⁺ T cells was detected by HLA*02:01/P53₂₆₄₋₂₇₂ tetramer-LLGRNSFEV-PE (Cat#TS-M081-1, MBL International Corporation). HLA*02:01/ NY-ESO-1₁₅₇₋₁₆₅ Tetramer- SLLMWITQC- PE (Cat#TB-M011-1, MBL International Corporation) was used as a control staining. (Please refer to highlighted sentence at **Page 40** in the third revised manuscript).

Peptide-pulsed dendritic cells (DCs) have been demonstrated to increase the frequency of peptide-specific autologous T cells in the previous studies. For instance, Seth M Pollack *et al.* reported that NY-ESO-1 tetramer positive CD8⁺ T cells were not observed in the starting leukapheresis product (detection thresh hold <0.01%). After two stimulation of NY-ESO-1₁₅₇₋₁₆₅ peptide-pulsed DCs cycles, the largest positive sample contained NY-ESO-1 tetramer positive T cells at a frequency of 8.05% (Figure 1 and Table 2 of this paper)¹. Beatriz M. Carreno *et al.* reported that the frequency of TMEM48 F169L-specific T cells in CD8⁺ populations isolated directly from PBMC

(Patient MEL21) was 0.4%, and increased to 2.6% after ex-vivo expansion using TMEM48 F169L peptide-pulsed autologous DC; The frequency of SEC24A P469L-specific T cells in CD8⁺ populations isolated directly from PBMC (Patient MEL38) was 0.1%, and increased to 3.3% after ex-vivo expansion using SEC24A P469L peptide-pulsed autologous DC (Fig.S4B of this paper)².

In our study, to measure the antigen specific T cell-mediated cytotoxicity on the autophagy-deficient MDA-MB-231 cells, human P53 peptide-specific T cells were generated using autologous dendritic cells (DCs) according to a modification of the method reported previously^{1, 3}. In this experiment, the DCs loaded with P53₂₆₄₋₂₇₂-antigen were cocultured with autologous T lymphocytes from HLA-A2⁺ healthy donor to induce P53 peptide-specific T cells. Autologous T cells stimulated with no peptide-pulsed DCs were used as control T cell. The result showed that the frequency of P53₂₆₄₋₂₇₂ tetramer⁺ CD8⁺T cells increased from 1.95% to 10.2% after stimulation with P53₂₆₄₋₂₇₂ peptide-pulsed DCs. As a control staining, NY-ESO-1₁₅₇₋₁₆₅ Tetramer⁺CD8⁺T cells didn't change obviously. We are sorry for not providing the control tetramer staining figures in the last second revision.

Actually, it has been reported that the mean frequency of p53₂₆₄₋₂₇₂ tetramer⁺ CD3⁺ CD8⁺ T cells in CD3⁺ CD8⁺ T cells isolated directly from PMBC of HLA-A2⁺

healthy individuals is around 0.02%⁴. According to the mean frequency reported, we modified the gate strategy for p53₂₆₄₋₂₇₂ tetramer analysis in the third revision. After an adjustment, the frequency of P53₂₆₄₋₂₇₂ tetramer⁺CD8⁺T cells increased from 0.12% to 2.2% after stimulation with P53₂₆₄₋₂₇₂ peptide-pulsed DCs. As a control staining, NY-ESO-1₁₅₇₋₁₆₅ Tetramer⁺CD8⁺T cells didn't change obviously. **We presented the new data in Supplementary Fig.2o of the third revised manuscript.**

Supplementary Fig.2 Autophagy deficiency reduces T cell-mediated tumor killing *in vitro*.

(o) Autologous purified T cells were cocultured with p53 peptide-pulsed DCs, followed by stained with p53₂₆₄₋₂₇₂ tetramer and anti-CD8. NY-ESO-1₁₅₇₋₁₆₅ tetramer was used as a control tetramer staining. The numbers in the upper right quadrants (Q2) of dot plots indicate the frequency of p53₂₆₄₋₂₇₂ tetramer⁺CD8⁺ T or NY-ESO-1₁₅₇₋₁₆₅ tetramer⁺CD8⁺ T cells. Unstimulated T cells were from PBMC of the same HLA-A2⁺ healthy donor.

In addition, in the Fig.8c, we also used P53 antigen-specific T cell-mediated cytotoxicity assay to measure the synergistic effect. In the experiment, the DCs loaded with P53₂₆₄₋₂₇₂-antigen were cocultured with autologous T lymphocytes from HLA-A2⁺ healthy donor to induce P53 peptide-specific T cells. T cells stimulated with no peptide-pulsed DCs were used as control T cells. The result showed that the frequency of P53₂₆₄₋₂₇₂ tetramer⁺ CD8⁺T cells increased from 1.96% to 9.97% after stimulation with P53₂₆₄₋₂₇₂ peptide-pulsed DCs. As a control staining, NY-ESO-1₁₅₇₋₁₆₅ Tetramer⁺CD8⁺T cells didn't change obviously. We are sorry for not

providing the control tetramer staining figures in the last second revision.

According to the reported mean frequency of p53₂₆₄₋₂₇₂ tetramer⁺ from PMBC of HLA-A2⁺ healthy individuals⁴, we also modified this gate strategy for p53₂₆₄₋₂₇₂ tetramer analysis in the third revision. After an adjustment, the frequency of P53₂₆₄₋₂₇₂ tetramer⁺CD8⁺T cells increased from 0.17% to 1.14% after stimulation with P53₂₆₄₋₂₇₂ peptide-pulsed DCs. As a control staining, NY-ESO-1₁₅₇₋₁₆₅ Tetramer⁺CD8⁺T cells didn't change obviously. **We presented the new data in supplementary Fig.10b of the third revised manuscript.**

Supplementary Fig. 10 Blockade of TNC sensitizes checkpoint blockade immunotherapy *in vitro*. (b) Autologous purified T cells were cocultured with p53 peptide-pulsed DCs, followed by stained with p53₂₆₄₋₂₇₂ tetramer and anti-CD8. NY-ESO-1₁₅₇₋₁₆₅ tetramer was used as a control tetramer staining. The numbers in the upper right quadrants (Q2) of dot plots indicate the frequency of p53₂₆₄₋₂₇₂ tetramer⁺CD8⁺ T or NY-ESO-1₁₅₇₋₁₆₅ tetramer⁺CD8⁺ T cells. Unstimulated T cells were from PBMC of the same HLA-A2⁺ healthy donor.

Moreover, we have used OVA antigen-specific T cell-mediated cytotoxicity model to elucidate the role of autophagy in the T-lymphocyte immune system and to measure the synergistic effect (**Fig.1h, Supplementary Fig. 2p, Supplementary Fig. 10c**). We depleted Atg7 in ovalbumin (OVA) positive melanoma B16F10 cells. Then the cells were co-cultured with activated CD8⁺ T cells isolated from OT-I TCR transgenic mice, which providing an antigen-specific T cell-mediated killing to B16F10-OVA cells. We obtained the similar results from OVA -specific T cell-mediated cytotoxicity model and P53 antigen-specific T cell-mediated cytotoxicity model.

2. they have provided additional details about the Crispr/cas9 strategies. It is clear they used clones in their studies. They have to clarify that in their text. If they have used clones, they have to clarified how many clones they have used to control for clonal variation. In addition, the exact clone characteristics, particularly the original sequencing results showing exact sequence insertion and deletion, need to be put in the supplemental materials. The entire Western blot indicating the size of the target protein and the possible truncated protein (due to Crispr/cas9-triggered deletion or truncation) in these clones need to be shown. It is not possible that the entire protein is absent due to small deletion or insertion. Assuming the “null” allele is a huge problem in the current literature involving Crispr/Cas9 technologies.

Response: Thank you for the helpful suggestion. To elucidate the role of autophagy in the T-lymphocyte immune system, we have established autophagy-deficient cell models by multiple clones, multiple cell lines or add backs. We have clarified the

Crispr/cas9 strategies in the method section of **“CRISPR-Cas9-mediated Gene Disruption”** (Please refer to the sentence at **Page 36-37**). To establish autophagy-deficient cell models, human Atg5 Double Nickase Plasmids (sc-416847-NIC), mouse Atg5 Double Nickase Plasmids (sc-419149-NIC), human Atg7 Double Nickase Plasmids (sc-400997-NIC), mouse Atg7 Double Nickase Plasmids (sc-428805-NIC), and mouse Beclin1 Double Nickase Plasmid (sc-425033-NIC) were purchased from Santa Cruz Co. All of these plasmids were transfection-ready purified DNA plasmids. All of these plasmids were transfected to the indicated cell lines according to the manufacturer’s instruction, and the knockout clones were isolated by single-cell dilution cloning. **The knockout clones include MDA-MB-231-Atg5KO#2, MDA-MB-231-Atg5KO#4, MDA-MB-231-Atg7KO#5, 4T1-Atg5KO#1, 4T1-Beclin1KO#2, B16-OVA-Atg7 KO#1, and B16-OVA-Atg7 KO#2.** Control CRISPR/Cas9 Double Nickase Plasmid (sc-437281) from Santa Cruz Co. was used as a negative control. For CRISPR-Cas9-mediated TNC (m) knockout in 4T1-Atg5KO#1 cell, the specific sgRNA sequence 5’-CCCGGAGCTCATACTGCCCT-3’ (region: chr4: 63,964, 709-63,964,728) targeting mouse TNC gene was cloned to LentiCRISPR (pXPR_001) plasmid. The packaging plasmids were co-transfected with pXPR-TNC sgRNA into HEK293T cells, and viral particles were harvested at 48 hr post-transfection. 4T1 Atg5 KO#1 cells were infected with viruses for 24 hr in the presence of polybrene (8 µg/ml), and stable cells were subsequently selected by puro for 3 days. LentiCRISPR (pXPR_001) plasmid was used as a negative control. We did not select monoclonal cells from the positive cells because viral infection was very efficient. Although there was a very weak band of TNC in TNC KO lane, the knockout efficiency of TNC was at least more than 80% (**Fig. 3e**).

All the knockout cell lines were identified by sequencing. We presented the new data of representative original sequences of the Atg5, Atg7, Beclin1 and TNC locus targeted by Cas9n in Supplementary Fig.12 of the third revised manuscript.

a Human Atg5 locus

Target B PAM

5' .. TATAACGAAATCCATTTTCTCTGCAGGATATTCATGAGTTCCGATTGATGGCCAAAAC**TGG**TCAAATCTGTC..3'
 |||
 3' .. ATATTGCTTTA**CGT**AAAAGAAGAC**GTCTATAAG**GTACTCAAAGGCTAACTACCGGGTTTGACCAAGTTAGACAG..5'

PAM Target A

5' .. TATAACGAAATCCATTTTCTCTGCAGGATATTCATGAGTTCCGATTGATGGCCAAAAC**TGG**TCAAATCTGTC..3' (MDA-MB-231-WT)
 5' .. TATAACGAAATCCATTTTCTCTGCAGGATA-----AATCTGTC..3' (MDA-MB-231-Atg5KO#2, Allele1)
 5' .. TATAACGAAATCCATTTTCTCTGCAGGATATTCATGAGTTCCGATTGATGGCCAAAAC**TGG**TCAAATCTGTC..3' (MDA-MB-231-Atg5KO#2, Allele2)
 catga

5' .. TATAACGAAATCCATTTTCTCTGCAGGATATTCATGAGTTCCGATTGATGGCCAAAAC**TGG**TCAAATCTGTC..3' (MDA-MB-231-Atg5KO#4, Allele1)
 gatattccatgagtttccgattgatcttctgca

5' .. TATAACGAAATCCATTTTCTCTGCAGGATATTCATGAGTTCCGATTGATGGCCAAAAC**TGG**TCAAATCTGTC..3' (MDA-MB-231-Atg5KO#4, Allele2)
 ctctgcaggatattccatgagtttccgattgat

b Human Atg7 locus

Target A PAM

5' .. CGGCAGCTACGGGGATCCTGGACTCTCTAAACTGCAGTTTGCCCTTTTAGTAGTGCCTTGGATGTTGGGTTTTG..3'
 |||
 3' .. GCGCTGATGCCCC**TAGGAC**TGAGAGATTGAC**GTCAA**CGGGGAAATCATCACGGAACCTACAACCCAAAAC..5'

PAM Target B

5' .. CGGCAGCTACGGGGATCCTGGACTCTCTAAACTGCAGTTTGCCCTTTTAGTAGTGCCTTGGATGTTGGGTTTTG..3' (MDA-MB-231-WT)
 5' .. CGGCAGCTACGGGGATCCTGGACTCTCTAAACTGCAGTTTGCCCTTTTAGTAGTGCCTTGGATGTTGGGTTTTG..3' (MDA-MB-231-Atg7KO#5, Allele1)
 5' .. CGGCAGCTACGGGGATCCTGGACTCTCTAAACTGCAGTTT---tta---TAGTGCCTTGGATGTTGGGTTTTG..3' (MDA-MB-231-Atg7KO#5, Allele2)
 ta

c Mouse Atg7 locus

Target A PAM

5' .. ATAGTGCCCTGGACGTTGGCTTCTGGCAGAACTGACCCAGAAGAAGTTGAACGAGTACCGCTGGACGAGGC..3'
 |||
 3' .. TATCAC**GGGAC**CTGCAACCGAAGACCGTCTGACTGGTCTTCTCAACTTGCTCATGCGGACCTGCTCCG..5'

PAM Target B

5' .. ATAGTGCCCTGGACGTTGGCTTCTGGCAGAACTGACCCAGAAGAAGTTGAACGAGTACCGCTGGACGAGGC..3' (B16F10-ova-WT)
 5' .. ATAGTGCCCTGGACGTTGGCTTCTGGCAGAACTGACCCAGAAGAAGTTGAACGAGTACCGCTGGACGAGGC..3' (B16F10-ova-Atg7KO#1, Allele1)
 acgttgg tggca

5' .. ATAGTGCCCTGGACGTTGGCTTCTGGCAGAACTGACCCAGAAGAAGTTGAACGAGTACCGCTGGACGAGGC..3' (B16F10-ova-Atg7KO#1, Allele2)
 gacccaagaagaagttgacccaagaagaagttgaaogagaa ccagaagaagttga

5' .. ATAGTGCCCTGGACGTTGGCTTCTGGCAGAACTGACCCAGAAGAAGTTGAACGAGTACCGCTGGACGAGGC..3' (B16F10-ova-Atg7KO#2, Allele1)
 agaagagttgaaogaagaagcaogactgacc ttgaaagagttacc

5' .. ATAGTGCCCTGGACGTTGGCTTCTGGCAGAACTGACCCAGAAGAAGTTGAACGAGTACCGCTGGACGAGGC..3' (B16F10-ova-Atg7KO#2, Allele2)
 agaagaagttgaaogaagaagcaogactgacc

d Mouse Atg5 locus

Target B PAM

5' .. GCTTTTGCCAAGAGTCAGCTATTTGACGTTGGTAACTGACAAAGTGA AAAAGCACTTTCAGAGGTTATGAGACA..3'
 |||
 3' .. CGAAAAC**GCTCTCAGT**CGATAAACTGCACCATTTGACTTTTTCGTTGAAAGTCTTCCAATCTCTGT..5'

PAM Target A

5' .. GCTTTTGCCAAGAGTCAGCTATTTGACGTTGGTAACTGACAAAGTGA AAAAGCACTTTCAGAGGTTATGAGACA..3' (4T1-WT)
 5' .. GCTTTTGCCAAGAGTCAGCTATTTGACGTTGGTAACTGACAAAGTGA AAAAGCACTTTCAGAGGTTATGAGACA..3' (4T1-Atg5KO#1, Allele1)
 5' .. GCTTTTGCCAA-----AGGTTATGAGACA..3' (4T1-Atg5KO#1, Allele2)
 g

e Mouse Beclin1 locus

Target B PAM

5' .. ACCCGGTCCAGGATCTTGAAGCTCGTGTCCAGTTTCAGAGGCTGGCTACAGCGCTGGCACACGAAGCTCACC..3'
 |||
 3' .. TGGGCCA**GGTCTAGAACT**TCGAGCAGGTCAAAGTCTCCGACCGATGTCGCGACCGTGTGCTTCGAGTGG..5'

PAM Target A

5' .. ACCCGGTCCAGGATCTTGAAGCTCGTGTCCAGTTTCAGAGGCTGGCTACAGCGCTGGCACACGAAGCTCACC..3' (4T1-WT)
 5' .. ACCCGGTCCAGGATCTTGAAGC-----ACAGCGCTGGCACACGAAGCTCACC..3' (4T1-Beclin1KO#2, Allele1)
 5' .. ACCCGGTCCAGGATCTTGAAGCTCGTGTCCAGTTTCAG-----CGTGGCACACGAAGCTCACC..3' (4T1-Beclin1KO#2, Allele2)

f Mouse TNC locus

Target PAM

5' .. ACTCCCATGGTCTTGTAGGTCCACCGGAGCTCATACTGCCCTTGGGCTGTGATTTTGCTCAGGTTATCCAGT..3'
 |||
 5' .. ACTCCCATGGTCTTGTAGGTCCACCGGAGCTCATACTGCCCTTGGGCTGTGATTTTGCTCAGGTTATCCAGT..3' (4T1-Atg5KO#1 WT)
 5' .. ACTCCCATGGTCTTGTAGGTCCACCGGAGCTCATACTGCCCTTGGGCTGTGATTTTGCTCAGGTTATCCAGT..3' (4T1-Atg5KO#1 TNCKO, 17/22)
 5' .. ACTCCCATGGTCTTGTAGGTCCACCGGAGCTCATACTGCCCTTGGGCTGTGATTTTGCTCAGGTTATCCAGT..3' (4T1-Atg5KO#1 TNCKO, 2/22)

5' .. ACTCCCATGGTCT-----CAGGTTATCCAGT..3' (4T1-Atg5KO#1 TNCKO, 1/22)
 5' .. ACTCCCATGGTCTTGTAGGTCCA---g-----TACTGCCCTTGGGCTGTGATTTTGCTCAGGTTATCCAGT..3' (4T1-Atg5KO#1 TNCKO, 1/22)
 5' .. ACTCCCATGGTCT-----cg-----GGTATCCAGT..3' (4T1-Atg5KO#1 TNCKO, 1/22)

Supplementary Fig. 12 Representative sequences of the Atg5, Atg7, Beclin1 and TNC locus targeted by Cas9n in the indicated cell lines. (a-e) DNA sequencing analysis of targeting ATG5, ATG7, and Beclin1 in the knockout clones include MDA-MB-231-Atg5KO#2(a), MDA-MB-231-Atg5KO#4(a), MDA-MB-231-Atg7KO#5(b), B16-OVA-Atg7KO#1(c), B16-OVA-Atg7KO#2(c), 4T1-Atg5KO#1(d), and 4T1-Beclin1KO#2 (e). (f) DNA sequencing analysis of TNC in 4T1-ATG5KO#1 cell. The PAM sequences are underlined and highlighted in green; the targeting sequences are underlined and highlighted in blue; deletions (-) in red, insertions (^) and mutations, lower case in red; N/N indicates positive colonies out of total sequenced.

We have detected the knockout clones with the entire Western blot indicating the size of the target protein and the possible truncated protein. We presented the new data in Supplementary Fig. 2a, Supplementary Fig. 2d, Supplementary Fig. 2p, and Supplementary Fig.3a in the third revised manuscript. The result showed that the targeted Atg5, Atg7, and Beclin1 proteins by CRISPR-Cas9-mediated gene disruption were nearly completely absent. More importantly, the formation of the lipidated form of LC3 (LC3B-II), the most important marker for autophagy, was completely blocked in the knockout clones. These mean the autophagy-deficient cell models were successfully established when the autophagy-related genes Atg5, Atg7, and Beclin1 were knockout. Actually, Arne H. Smits *et al.* reported that they observed residual protein expression, including truncated protein, for about one third of the quantified targets in CRISPR-Cas9-generated KO lines they studied, at variable levels from low to original⁵. However, we did not observe the obvious truncated protein in the knockout clones. There could be a variety of possible reasons. In general, the Cas9-sgRNA complex is targeted to a specific sequence in the coding region of a gene and mediate DNA double-stranded break (DSBs)⁶. Especially, CRISPR Double Nickase Plasmid products, which contains two Cas9-nicking enzymes directed by a pair of sgRNAs targeting opposite strands of a target locus, offer improved specificity while maintaining a high level of knockout efficiency⁷. The DSBs is repaired by non-homologous end joining (NHEJ)-mediated DNA insertion or deletion mutations that can lead to frameshifts and a premature termination codon (PTC) in the expressed transcript, resulting in nonsense-mediated decay (NMD) of the mRNA and aberrant peptide products that may be easy to degrade^{5, 8, 9}. Also, frameshift mutations can lead

to full disruption of the protein despite no or weak NMD⁵. In addition, we checked the source information of the antibodies we used for immunoblot at Cell Signaling Technology website (<https://www.cellsignal.com/>). The detailed source information of Atg7 Antibody (Cat #2631, Cell Signaling Technology) is not provided at the website. According to the source information of Atg5 antibody (Cat#12994, Cell Signaling Technology) and Beclin1 antibody (Cat#3738, Cell Signaling Technology) provided at the website, the recognition epitopes of the two antibodies are after the cleavage sites induced by Cas9/RNA. As one of the reasons, the two antibodies may not be recognized after frameshifts.

4T1

Figures. Effect of Atg5, Atg7, and Beclin1 knockout in the indicated clones using CRISPR-Cas9 technology.

We hope that these responses are adequate and would like to express our gratitude to you for your time and considerations in reviewing our manuscript.

Best Regards,

Dr. Rong Deng

References

1. Pollack SM, *et al.* Tetramer guided, cell sorter assisted production of clinical grade autologous NY-ESO-1 specific CD8(+) T cells. *J Immunother Cancer* **2**, 36 (2014).
2. Carreno BM, *et al.* Cancer immunotherapy. A dendritic cell vaccine increases the breadth and diversity of melanoma neoantigen-specific T cells. *Science* **348**, 803-808 (2015).
3. Eura M, *et al.* A wild-type sequence p53 peptide presented by HLA-A24 induces cytotoxic T lymphocytes that recognize squamous cell carcinomas of the head and neck. *Clinical cancer research : an official journal of the American Association for Cancer Research* **6**, 979-986 (2000).
4. Hoffmann TK, *et al.* Frequencies of tetramer+ T cells specific for the wild-type sequence p53(264-272) peptide in the circulation of patients with head and neck cancer. *Cancer research* **62**, 3521-3529 (2002).
5. Smits AH, *et al.* Biological plasticity rescues target activity in CRISPR knock outs. *Nature methods* **16**, 1087-1093 (2019).
6. Mali P, *et al.* RNA-guided human genome engineering via Cas9. *Science* **339**, 823-826 (2013).

7. Ran FA, *et al.* Double nicking by RNA-guided CRISPR Cas9 for enhanced genome editing specificity. *Cell* **154**, 1380-1389 (2013).
8. Lieber MR. The mechanism of double-strand DNA break repair by the nonhomologous DNA end-joining pathway. *Annual review of biochemistry* **79**, 181-211 (2010).
9. Lykke-Andersen J, Bennett EJ. Protecting the proteome: Eukaryotic cotranslational quality control pathways. *The Journal of cell biology* **204**, 467-476 (2014).

REVIEWERS' COMMENTS:

Reviewer #1 (Remarks to the Author):

The authors have satisfactorily addressed this reviewer's concern.